# Mapping the interaction surface between Ca$_V$β and actin and its role in calcium channel clearance

Francisco Castilla[1,2,8,10], Victor Lugo[1,2,10], Erick Miranda-Laferte[1], Nadine Jordan[1], Pitter F. Huesgen [3,9], Beatrix Santiago-Schübel [3,4 ✉], Mercedes Alfonso-Prieto [5,6 ✉] & Patricia Hidalgo [1,7 ✉]

Defective ion channel turnover and clearance of damaged proteins are associated with aging and neurodegeneration. The L-type Ca$_V$1.2 voltage-gated calcium channel mediates depolarization-induced calcium signals in heart and brain. Here, we determined the interaction surface between actin and two calcium channel subunits, Ca$_V$β$_2$ and Ca$_V$β$_4$, using cross-linking mass spectrometry and protein-protein docking, and uncovered a role in replenishing conduction-defective Ca$_V$1.2 channels. Computational and in vitro mutagenesis identified hotspots in Ca$_V$β that decreased the affinity for actin but not for Ca$_V$1.2. When coexpressed with Ca$_V$1.2, none of the tested actin-association-deficient Ca$_V$β mutants altered the single-channel properties or the total number of channels at the cell surface. However, coexpression with the Ca$_V$β$_2$ hotspot mutant downregulated current amplitudes, and with a concomitant reduction in the number of functionally available channels, indicating that current inhibition resulted from a build-up of conduction silent channels. Our findings established Ca$_V$β$_2$–actin interaction as a key player for clearing the plasma membrane of corrupted Ca$_V$1.2 proteins to ensure the maintenance of a functional pool of channels and proper calcium signal transduction. The Ca$_V$β–actin molecular model introduces a potentially druggable protein-protein interface to intervene Ca$_V$-mediated signaling processes.

Voltage-gated calcium (Ca$_V$) channels convert electrical signals into increased intracellular calcium concentrations in neuronal, heart muscle, and endocrine cells, thereby triggering a variety of physiological responses[1]. The functional core of the Ca$_V$ complex is formed by the pore-forming subunit (Ca$_V$α$_1$) and the β-subunit (Ca$_V$β). The latter regulates multiple calcium signaling pathways by directly controlling calcium permeation and intracellular Ca$_V$α$_1$ trafficking[2–6]. Dysfunction of Ca$_V$ subunits is associated with several cardiovascular, neurological, and psychiatric pathological conditions that are collectively known as calcium channelopathies[1,7–16].

[1]Institute of Biological Information Processing (IBI-1)—Molecular and Cellular Physiology, Forschungszentrum Jülich, Jülich, Germany. [2]Graduate Program, Faculty of Mathematics and Natural Sciences, Heinrich-Heine University, Düsseldorf, Germany. [3]Central Institute of Engineering, Electronics and Analytics (ZEA-3), Forschungszentrum Jülich, Jülich, Germany. [4]Institute of Biological Information Processing (IBI-7)—Structural Biochemistry, Forschungszentrum Jülich, Jülich, Germany. [5]Institute of Neuroscience and Medicine (INM-9)—Computational Biomedicine, Forschungszentrum Jülich, Jülich, Germany. [6]Cécile and Vogt Institute for Brain Research, Medical Faculty, Heinrich-Heine University, Düsseldorf, Germany. [7]Institute of Biochemistry, Heinrich-Heine University, Düsseldorf, Germany. [8]Present address: Biologics Analytical Research and Development, AbbVie Deutschland GmbH & Co. KG, Ludwigshafen, Germany. [9]Present address: Institute of Biology II, University of Freiburg, Freiburg, Germany. [10]These authors contributed equally: Francisco Castilla, Victor Lugo. ✉e-mail: b.santiago-schuebel@fz-juelich.de; m.alfonso-prieto@fz-juelich.de; pa.hidalgo@fz-juelich.de

The $Ca_V\beta$ family comprises four subclasses ($Ca_V\beta_1$–$Ca_V\beta_4$), each with several alternative splice variants[2,4,6,17–19]. $Ca_V\beta$ is one of the smallest members of the membrane-associated guanylate kinase (MAGUK) family. It has a modular architecture, with two highly conserved protein-protein interaction (PPI) domains (a Src homology 3 (SH3) domain and a guanylate kinase (GK) domain) that form the structural and functional core of the protein that recapitulates most of the channel modulatory functions[20–22]. In contrast, the N- and C-terminal regions and the HOOK linker joining the two conserved domains are variable in sequence and length[17].

High-resolution crystal structures are available for the $Ca_V\beta_2$, $Ca_V\beta_3$, and $Ca_V\beta_4$ subunits, both alone and in complex with a highly conserved consensus sequence among $Ca_V\alpha_1$ subunits: a segment of 18 amino acids located in the intracellular I–II loop and known as the $\alpha_1$ interaction domain (AID)[23–26]. $Ca_V\beta$ has an elongated shape, with the SH3 and GK domains located on opposite sides of the protein's core. No crystallographic data are available for the flexible regions. The AID forms an α-helix that fits into a conserved hydrophobic groove (α-binding pocket) formed exclusively by GK domain residues.

Beyond its canonical role in potentiating $Ca_V$-mediated calcium currents as part of the $Ca_V$ complex, $Ca_V\beta$ also interacts with diverse protein partners to fulfil a variety of cellular functions, including with filamentous actin[2,4,18,27–33]. The association between $Ca_V\beta$-actin has been demonstrated by in vitro binding assays and in vivo via fluorescence lifetime imaging microscopy and single-molecule localization microscopy[32]. However, in the absence of a molecular model for the $Ca_V\beta$–actin interaction, its function has been explored by disrupting the interaction using inhibitors of actin polymerization[32,33]. Based on this strategy, actin association with two $Ca_V\beta$ isoforms, $Ca_V\beta_2$ and $Ca_V\beta_4$ (which are highly expressed in the heart and brain, respectively) was proposed to upregulate L-type calcium currents in HL-1 cardiomyocytes and to increase the readily releasable pool of synaptic vesicles independent of channel function in primary hippocampal neurons, respectively[32,33].

In this work, we identified the actin interaction surface of $Ca_V\beta_2$ and $Ca_V\beta_4$ by implementing a strategy that combined chemical cross-linking mass spectrometry (XL-MS) with computational biology, followed by experimental validation of the resulting $Ca_V\beta$–actin models (Fig. 1). We identified several hotspot residues and generated mutants with significantly reduced in vitro affinity for actin, but not for the AID, in the background of $Ca_V\beta_2$ and the $Ca_V\beta_4$ truncated variant ($Ca_V\beta_4$ R482X). This $Ca_V\beta_4$ variant lacking the 39 C-terminal amino acids has been linked to juvenile myoclonic epilepsy but its etiology is unclear[13,34].

Electrophysiological recordings in HEK293 cells showed that, as compared with the wild-type $Ca_V\beta4$, neither $Ca_V\beta4$ R482X nor the actin-association-deficient $Ca_V\beta4$ R482X hotspot mutant affect macroscopic ionic currents mediated by $Ca_V1.2$ or $Ca_V2.2$.

However, cells coexpressing $Ca_V1.2$ and an actin-association-deficient $Ca_V\beta_2$ mutant have diminished ionic currents compared with those coexpressing $Ca_V1.2$ and the wild-type $Ca_V\beta_2$. Through gating current measurements combined with stationary noise analysis, we show the $Ca_V\beta_2$ mutant mediates current downregulation by reducing the number of functionally available channels.

Here, we propose a model in which the $Ca_V\beta_2$–actin interaction is required for the normal endocytic turnover of conduction-defective L-type $Ca_V$ proteins at the cell surface and, thus, for proper calcium signaling and protein homeostasis.

## Results

### Identification of cross-linked peptides by mass spectrometry

We mapped the contact surface between recombinant $Ca_V\beta_2$ and in vitro polymerized actin using XL-MS. Given that recombinant full-length $Ca_V\beta$ proteins are less stable under the lower salt conditions used for actin polymerization than their corresponding core regions

counterparts, we employed the core region of $Ca_V\beta_2$ ($Ca_V\beta_2$-core, Fig. 2A)[22,32]. The $Ca_V\beta_2$-core eluted as a monodisperse peak following size exclusion chromatography, with no indication of protein aggregation (Fig. 2B).

For chemical cross-linking, we employed two MS-cleavable cross-linkers, disuccinimidyl sulfoxide (DSSO) and disuccinimidyl dibutyric urea (DSBU), with spacer arms of 10.3 Å and 12.5 Å, respectively, that mainly react with lysine[35,36]. The cross-linking reaction was optimized for cross-linker concentration and incubation time, and compared to control reactions, in which either no cross-linker was added to the reaction or to the individual proteins and each protein alone was incubated with the cross-linkers (Fig. 2C and Supplementary Fig. 1). These experiments established 2.5 and 5.0 molar excess of DSSO and DSBU, respectively, and an incubation time of 60 minute as the optimal conditions for detecting cross-linking specific products between $Ca_V\beta_2$ and actin (Fig. 2C). The stability of the $Ca_V\beta_2$–actin complex was assessed using a F-actin cosedimentation assay, which is widely used to analyze the association of proteins with actin filaments[37,38]. The fraction of $Ca_V\beta_2$-core bound to actin show statistically non-significant difference after a 30- or 90- minute incubation time with actin, indicating that the complex formation is stable at least for 90 minutes under our experimental conditions (Supplementary Fig. 2).

To identify the sites of cross-linking, the reaction was performed in independent replicates and cross-linked proteins were resolved by SDS-PAGE (Supplementary Fig. 1). Although the efficiency of the cross-linking reaction for $Ca_V\beta_2$-core (48.2 kDa) and actin (43 kDa) was low, sufficient intermolecular cross-linked products were obtained with both cross-linkers. Products had an apparent molecular mass of about 110 kDa, which is fully absent in control reactions, suggesting that intermolecular chemical cross-links had formed between the $Ca_V\beta_2$-core and actin (Fig. 2C). The bands containing those cross-linked products were excised from the gel followed by in-gel trypsinization. The resulting peptides were analyzed by nano LC-MS/MS and identified from acquired mass spectrometry data using three software packages, MaxLynx, Merox and MetaMorpheus, as described in the Methods[39–41].

Within the five DSSO replicates, MetaMorpheus, MaxLynx and Merox detected a total of 893, 249 and 67 cross-linked-peptide-to-spectrum matches (CSMs), which included 179, 52 and 14 inter-linked CSMs, respectively (Supplementary Table 1). The inter-linked CSMs resulted in 56 intermolecular crosslinks (inter-XLs), of which eight inter-XLs were identified by all three software packages, 15 by MaxLynx and MetaMorpheus, one by MetaMorpheus and Merox, and 32 by MetaMorpheus only (Supplementary Fig. 3A).

Using the four DSBU replicates, MetaMorpheus, MaxLynx and Merox detected a total of 370 CSMs (including 19 inter-link CSMs), 331 CSMs (14 inter-link CSMs) and 103 CSMs (seven inter-link CSMs), respectively (Supplementary Table 1). 15 inter-XLs were identified, with four inter-XLs found by all three software packages, six by MaxLynx and MetaMorpheus, two by MetaMorpheus and Merox, one by MaxLynx and Merox, and two by MetaMorpheus only (Supplementary Fig. 3B).

Taking together the inter-XLs found with the two cross-linkers and removing redundancy, a total of 22 unique inter-XLs were identified (Fig. 2D and Supplementary Table 2). Comparing interlinks obtained with DSBU and DSSO resulted in seven common inter cross-links, 14 inter-XLs identified only with DSSO and one inter-XL only with DSBU (Supplementary Fig. 3C). Of the 22 unique inter-XLs, 16 were identified by at least two distinct software packages (Supplementary Table 2).

In the primary sequence the residues involved in inter-XLs are distributed within two distinct regions in actin (K52 and K63 in one; K293, K317, K328, and K330 in the other), and are mostly located within the GK domain in $Ca_V\beta_2$ (K274, K347, K354, K358 and K362) (Fig. 2D). When the aforementioned residues are mapped on the three-dimensional structure, they are distributed within two discrete

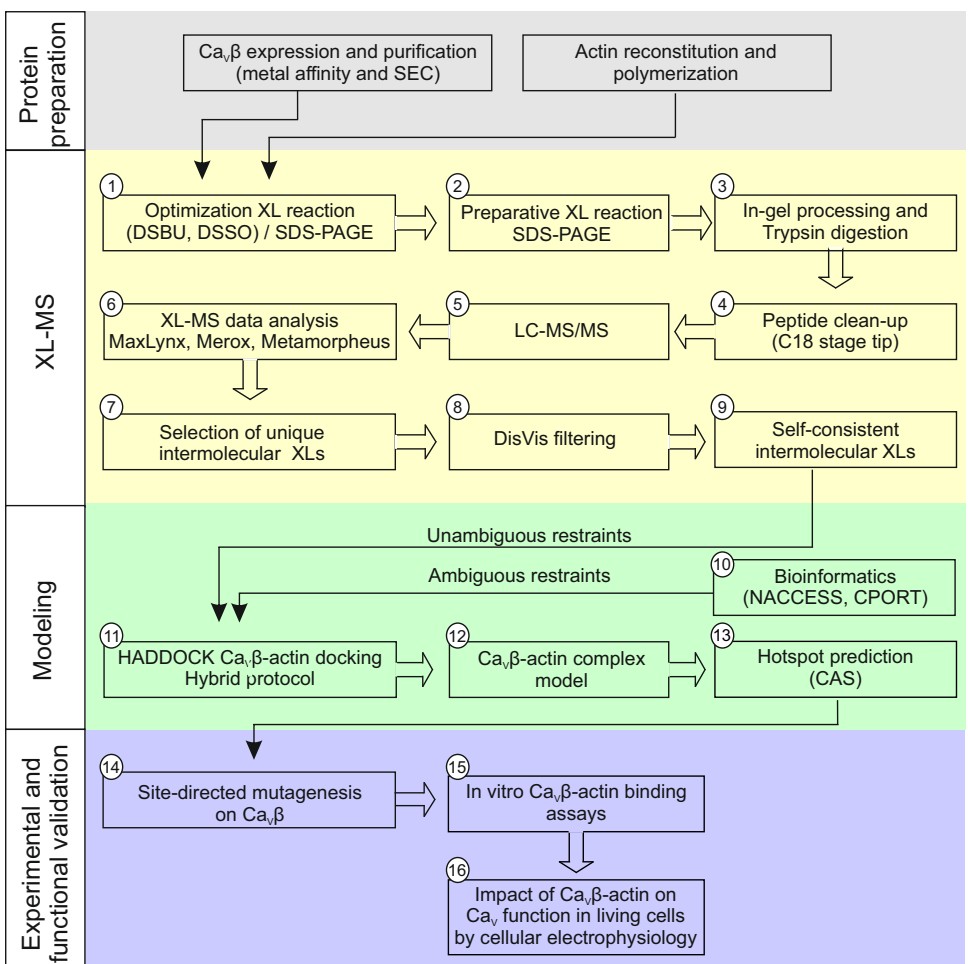

**Fig. 1 | Workflow diagram illustrating the strategy used to identify the inter-action surface between Ca$_V$β and actin by combining cross-linking mass spec-trometry (XL-MS) with protein-protein docking and experimental validation.** The overall strategy can be divided in four phases; a pre-phase, named protein preparation, aimed to generate the proteins under study, and XL-MS, modeling and experimental and functional validation phases. The optimization of the XL-reaction conditions was done by testing different amounts of the XL (DSBU and DSSO) at different reaction times, as shown in Supplementary Fig. 1, and visualized by SDS-PAGE (1). Replicas of the optimal XL-reaction conditions were carried out for further analysis (2). The band containing the specific cross-linked product between Ca$_V$β and actin was excised and in gel trypsinized (3). The mix of peptides were subjected to C18 stage tip cleaning (4) and LC-MS/MS (5). XL-MS data analysis was performed using three different search engines (6). Unique intermolecular cross-links that were identified by at least two search engines (7) were considered for filtering using DisVis (8) to obtain a set of self-consistent XL-MS derived distance restraints (9). XL-MS restraints together with bioinformatics-predicted restraints from NACCESS and CPORT programs, considered as unambiguous and ambiguous restraints, respectively (10), served as input files for the HADDOCK hybrid docking protocol (11) to generate Ca$_V$β−actin models (12). The top four HADDOCK models of the selected docking cluster were used to predict hotspots via Computational Alanine Scanning (CAS) with six predictors (Anchor, BeAtMuSiC, Bude Alanine Scan, Mutabind2, Robetta and SAAMBE-3D) (13). Hotspot mutations selected based on the average ΔΔG over the six predictors were introduced by site-directed mutagenesis into Ca$_V$β (14). The impact of the Ca$_V$β hotspot mutants on its binding affinity for actin were assessed by in vitro assays (15), and the mutants with decreased affinity were tested for its effect on calcium channel function in living cells by electrophysiology (16).

patches of an actin monomer, but form a contiguous surface on two adjacent monomers within the actin protofilament (Fig. 2E). The latter agrees with Ca$_V$β binding to actin filaments[32].

We employed DisVis analysis[42,43] to evaluate the information content of the distance restraints between Ca$_V$β and actin and considered a total of 16 unique inter-XLs with monomeric actin or of 32 XLs with dimeric actin (since the 16 unique inter-XLs can be a priori assigned to either monomer of the two adjacent actin subunits). After several iterations of DisVis filtering, a set of 11 self-consistent inter-XLs were identified, with a lower average violated fraction for dimeric actin (Fig. 2F and Supplementary Table 3).

### Structural model of the interaction between Ca$_V$β$_2$ and actin

Although DisVis analysis hints at association of Ca$_V$β with dimeric actin, it does not fully discard binding to monomeric actin. Therefore, docking-based modeling of the Ca$_V$β$_2$−actin interaction was

performed for both dimeric and monomeric actin. Calculations were carried out on the HADDOCK 2.4 webserver using the XL-MS distance restraints resulting from DisVis analysis and interface information predicted with bioinformatics[44,45].

For the Ca$_V$β$_2$−actin dimer complex, all generated models were clustered using an interface ligand (i-l)-RMSD cutoff of 7.5 Å, resulting in one cluster containing 198 models (Supplementary Fig. 4A and Supplementary Tables 4A and 5A). The similarity across those models was assessed with two RMSD-based metrics, the ligand (l)-RMSD and the interface (i)-RMSD, taking Ca$_V$β$_2$ as ligand and actin as receptor, as described[46,47]. The l-RMSD reports the RMSD of the backbone atoms of the Ca$_V$β$_2$ ligand after optimally aligning the actin receptor and the i-RMSD provides the backbone RMSD of the interface residues (using a 10 Å distance cutoff) of both actin and Ca$_V$β$_2$. We obtained values of 4.3 ± 4.1 and 1.0 ± 0.4 Å for l-RMSD and i-RMSD, respectively (Supplementary Fig. 4B, C and Supplementary Table 5A), indicating a high

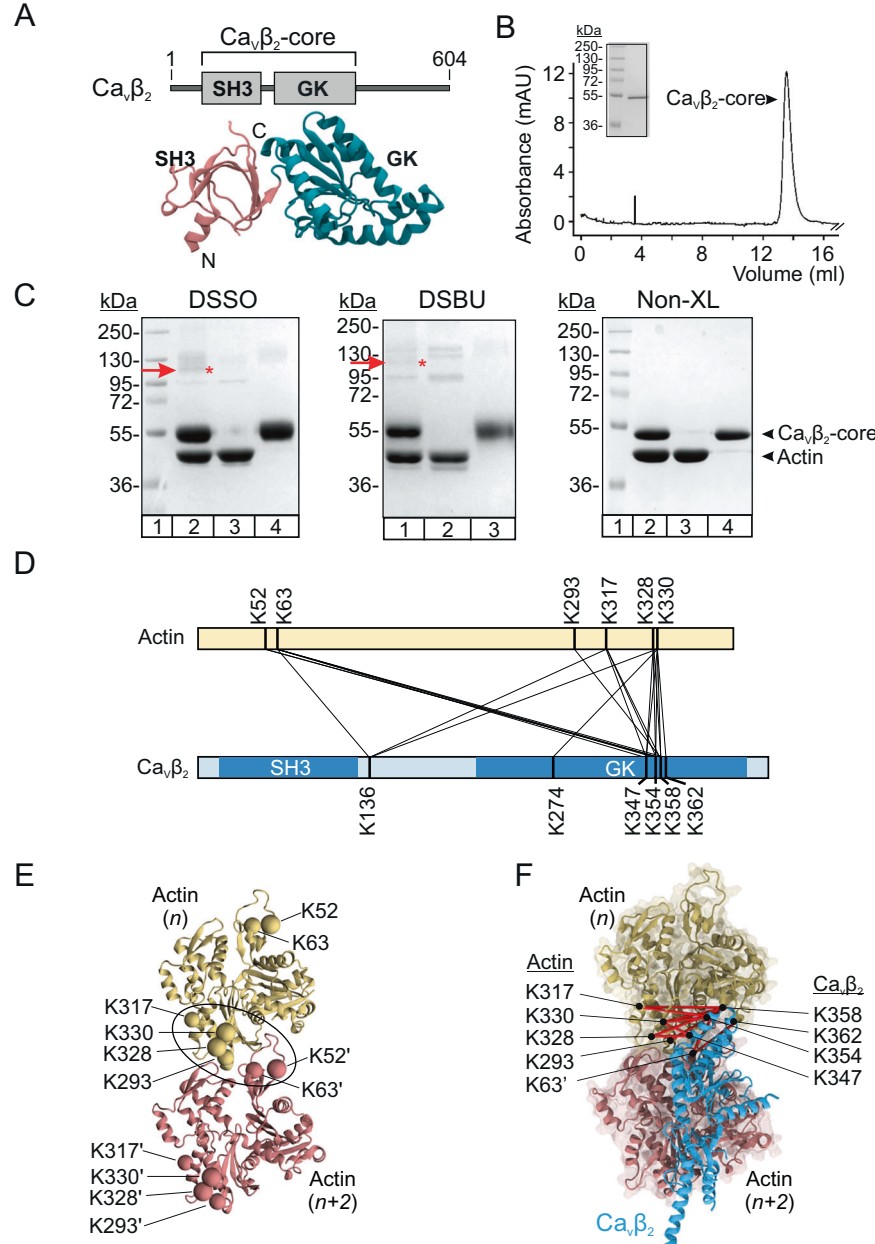

**Fig. 2 | Chemical XL-MS and computational analysis define a set of distance constraints for modeling the Ca$_V$β–actin complex. A** Domain architecture (top) and crystal structure (bottom) of rat Ca$_V$β$_2$ (UniProtKB Q8VGC3-2; PDB 5V2P). Src homology 3 (SH3) and guanylate kinase (GK) domains forming the protein core are labeled. Numbers represent the first and last residues. **B** Size exclusion chromatography profile of the Ca$_V$β$_2$-core; inset, purified protein resolved by SDS-PAGE. Numbers denote the size of molecular weight (MW) standards. The experiments were repeated at least 3 times. **C** Representative SDS-PAGE images of optimal cross-linking reactions for the Ca$_V$β$_2$-core and actin using DSSO, DSBU, and control reaction with only vehicle, no cross-linker (Non-XL) from 5, 4 and 3 independent replicas, respectively. DSSO: lane 1, MW standards; lane 2, Ca$_V$β$_2$-core–actin cross-linking; lane 3, actin alone incubated with DSSO; and lane 4, Ca$_V$β$_2$-core alone incubated with DSSO. DSBU: lane 1, Ca$_V$β$_2$-core–actin cross-linking; lane 2, actin alone incubated with DSBU; and lane 3, Ca$_V$β$_2$-core alone incubated with DSBU. MW standards lane was omitted and the full image is shown

in Supplementary Fig. 1. The red arrow and asterisk denote the band that was excised from the gel for tryptic digestion and LC-MS/MS analysis. Non-XL control: lane 1, MW standards; lane 2, Ca$_V$β$_2$-core and actin together incubated with DMSO; lane 3, actin alone incubated with DMSO; and lane 4, Ca$_V$β$_2$-core alone incubated with DMSO. **D** Linear representation depicting the location and residue number of unique intermolecular cross-links between Ca$_V$β$_2$-core and actin (UniProtKB P68135, PDB 5OOE). **E** Three-dimensional ribbon representation of two adjacent actin monomers in one protofilament, named *n* and *n + 2*, shown in yellow and pink, respectively. Actin amino acids participating in the unique intermolecular cross-links (shown as spheres, with those in monomer *n + 2* labeled with a prime (ʹ) symbol) form a contiguous patch across two adjacent actin subunits (black contour). **F** Structural mapping of inter-cross-links (red lines) that satisfied data consistency upon DisVis filtering for one Ca$_V$β$_2$ molecule and two adjacent actin monomers. Actin is shown in space-filling and ribbon mode, but for clarity Ca$_V$β$_2$ is displayed in ribbon mode only.

similarity of the PPI interface across docking models, despite a shift in the position and orientation of Ca$_V$β$_2$ relative to actin, according to the CAPRI classification[46,47].

Next, we tested the cluster models for their consistency with the inter-XLs between Ca$_V$β$_2$ and actin (Supplementary Fig. 4D). No

violations of the maximum-bound distance of 30 Å between the Cα-Cα of two cross-linked residues were observed in all 198 models, except for one Ca$_V$β–actin cross-link pair (K358–K317), where for a fraction of complexes the distance was up to 35 Å (Supplementary Fig. 4D). Previous studies have considered a maximum Cα-Cα distance of 35 Å to

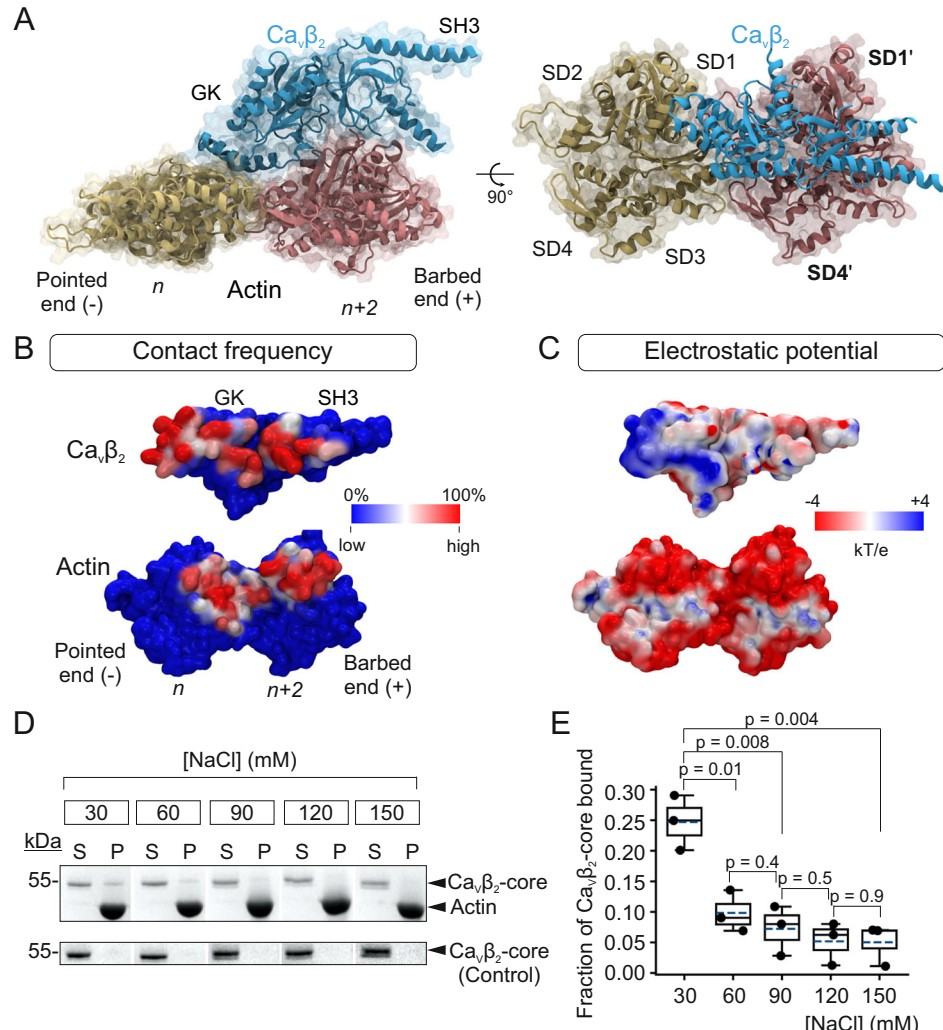

**Fig. 3 | Model of Ca$_V$β$_2$ bound to actin shows an extended contact surface and electrostatic complementarity. A** Space-filling and ribbon diagram showing two views related by a 90° rotation of the best-scoring docking model of Ca$_V$β$_2$ complexed with actin. One molecule of Ca$_V$β$_2$ (blue) interacts with two adjacent monomers in one actin protofilament, named *n* (yellow) and *n + 2* (pink). Actin subunits are labeled based on their position along the filament from the pointed end (−) to the barbed end (+). For clarity, Ca$_V$β$_2$ is shown in ribbon mode only in the right panel. The four actin subdomains, SD1–SD4, are labeled in the *n* monomer. Subdomains in monomer *n + 2* are indicated with a prime (′); SD2′ and SD3′ were omitted. **B** Surface models showing the contact frequency of Ca$_V$β$_2$ and actin residues. **C** Same as in *B* but showing the electrostatic potential map of both proteins. **D** SDS-PAGE analysis of a representative F-actin cosedimentation assay using the indicated NaCl concentrations. Briefly, the Ca$_V$β$_2$-core was incubated with phalloidin-stabilized actin filaments, the mixture was centrifuged, and the supernatant (S) and pellet (P) fractions separated. The pellet was resuspended in the same volume of SDS-loading buffer as the volume of supernatant, and the supernatant and pellet fractions were resolved by denaturing SDS-PAGE. Control assays included the Ca$_V$β$_2$-core only (lower panel). For clarity, irrelevant lanes were cropped from the source images (as indicated by the white space); full gel images are shown in Supplementary Fig. 8. **E** Fraction of Ca$_V$β$_2$-core bound to actin (calculated by densitometry) against NaCl concentration. Box plot shows the mean (dashed line), median (continuous line), interquartile range (25th–75th percentiles, box edges) and whiskers (1.5× interquartile range). Each dot represents an independent experiment; *p*-values, two-tailed *t*-test. Each experiment was repeated three times. Also see Supplementary Fig. 8.

account for conformational flexibility and ambiguity in the cross-link assignment, in particular for DSSO and DSBU[48–56]. In contrast, control docking simulations without XL-MS restraints resulted in 99% models with Cα-Cα inter-XL distances larger than the upper bound distance of 35 Å (Supplementary Table 6 and Supplementary Fig. 5). This corroborates the importance of using XL-MS restraints in protein-protein docking to identify the Ca$_V$β$_2$–actin interface in line with previous studies[57–62].

The top scoring model for the XL-MS guided protein-protein complex illustrates that Ca$_V$β$_2$ fits onto a hollow formed by two adjacent actin monomers within a single protofilament (Fig. 3A). The GK domain establishes contacts with both monomers, specifically with subdomains SD1 and SD3 and SD1′ and SD2′ of actin subunits *n* and *n + 2*, respectively. The SH3 domain interacts only with SD1′ of the *n + 2* monomer.

The PPI interface between Ca$_V$β$_2$ and the actin dimer was defined by mapping the contact frequency for each residue of the protein complex onto the corresponding three-dimensional structures (Fig. 3B). The residues with the largest contact frequency form continuous patches in Ca$_V$β$_2$, extending along both the SH3 and GK domains, and in actin, extending along both monomers. This binding mode is in agreement with in vitro assays showing that both SH3 and GK domains separately binding to actin filaments[32] and with the protein-protein interface reported for other actin binding proteins[38,63–66].

In contrast, docking simulations with Ca$_V$β$_2$ and a single actin monomer resulted in a protein-protein complex model in which the SH3 domain does not establish any intermolecular contact with actin (Supplementary Figs. 6 and 7 and Supplementary Tables 4C).

Moreover, binding of $Ca_V\beta_2$ to the $n + 2$ subunit would result in the SH3 domain clashing with the adjacent ($n$) subunit (Supplementary Fig. 7B). Therefore, our docking results with monomeric actin are incompatible with the two-domain binding mode of $Ca_V\beta_2$ with actin filaments, whereas the $Ca_V\beta_2$–actin dimer model is in agreement with this experimental evidence[32].

Calculation of the electrostatic potential surface of $Ca_V\beta_2$ and actin showed a positively charged patch on the $Ca_V\beta_2$-binding surface, mainly located in the GK domain, that matches a predominant negative charge distribution on the actin interaction surface, suggesting electrostatic binding complementarity (Fig. 3C). To examine the contribution of electrostatic interactions to $Ca_V\beta_2$–actin dimer complex formation, we measured the dependence on ionic strength. Binding of $Ca_V\beta_2$-core to actin was assessed using the F-actin cosedimentation assay at NaCl concentrations of 30–150 mM. To avoid the effects of the different salt concentrations on actin polymerization, we used phalloidin to stabilize the actin filaments[37]. In this assay, the $Ca_V\beta_2$-core was incubated with in vitro polymerized actin, and the pellet and supernatant fractions obtained by high-speed centrifugation were resolved by SDS-PAGE. In control reactions, the $Ca_V\beta_2$-core alone was centrifuged and both fractions were resolved by SDS-PAGE (Fig. 3D): the $Ca_V\beta_2$-core was virtually excluded from the pellet at all NaCl concentrations tested, whereas the fraction of the $Ca_V\beta_2$-core that cosedimented with actin decreased with increasing NaCl concentrations (Fig. 3E and Supplementary Fig. 8). Therefore, $Ca_V\beta_2$ binding to actin depends on the salt concentration, confirming that electrostatic interactions contribute to the stability of the complex.

## Identification of hotspot residues at the $Ca_V\beta_2$/actin interaction interface

Computational alanine scanning was used to predict which $Ca_V\beta_2$ residues contribute most to the protein-protein binding energy (so-called hotspots) using six different webservers (Anchor, BeAtMuSiC, BUDE Alanine Scan, Mutabind2, Robetta, and SAAMBE-3D)[67–73]. For each of the six predictors, the change in free energy due to alanine substitution ($\Delta\Delta G$) of each $Ca_V\beta_2$ residue at the PPI interface was calculated as an average over the top four structures of the best cluster, which displayed an i-RMSD of $0.9 \pm 0.6$ Å (Supplementary Figs. 9A and Supplementary Table 5A). Average $\Delta\Delta G$ values for all six predictors were used to rank the $Ca_V\beta_2$ residues as hotspots (Supplementary Fig. 9B), and the top seven predicted hotspot residues at the $Ca_V\beta_2$ interaction surface were selected for experimental mutagenesis and in vitro binding assays. Two of these are located on the SH3 domain (K90 and R128) and five on the GK domain (K347, Q350, K354, K358, and N365) (Fig. 4A). The highly ranked hydrophobic residue, L364, that lies within the hotspot region in the GK domain was also included in the experimental validation. The hotspots establish interactions with actin residues located in the $n$ subunit and $n + 2$ subunit (Supplementary Fig. 10 and Supplementary Table 7A).

Alanine substitution of the eight selected hotspot residues in the $Ca_V\beta_2$-core was used to generate two different mutant proteins, $Ca_V\beta_2$-core 8Ala and $Ca_V\beta_2$-core 6Ala, bearing alanine substitutions either at all eight hotspots (GK + SH3) or at the six located within the GK domain (Fig. 4B). These amino acid substitutions did not destabilize the $Ca_V\beta_2$-core in mutant proteins, as indicated by the lack of aggregates and the nearly symmetric single elution peak after size exclusion chromatography (Fig. 4B).

F-actin cosedimentation assays were used to compare the ability of wild-type (WT) and mutant proteins to interact with F-actin. In control reactions, a negligible amount of WT or mutant $Ca_V\beta_2$-core was sedimented by high-speed centrifugation (Fig. 4C, lanes 3, 7, and 11). In the presence of actin, the fraction of cosedimenting $Ca_V\beta_2$-core was significantly reduced for the mutant proteins (bound fraction: $0.24 \pm 0.03$ for $Ca_V\beta_2$-core WT, $0.11 \pm 0.03$ for $Ca_V\beta_2$-core 6Ala, and $0.09 \pm 0.02$ for $Ca_V\beta_2$-core 8Ala) (Fig. 4D and Supplementary Fig. 11 A).

To discount the possibility that the reduced actin-binding activity of $Ca_V\beta_2$-core hotspot mutants was due to their reduced stability in F-actin cosedimentation assay buffer, we compared the intrinsic fluorescence of these proteins. Tiny differences in the thermal unfolding curves and unfolding transition temperature ($T_m$) were observed among WT and mutant $Ca_V\beta_2$-core proteins, indicating that they have comparable structural stability in the assay buffer (Supplementary Fig. 11B).

Inspection of the molecular model for $Ca_V\beta$–actin complex showed no overlap between the actin-binding hotspots identified in this study and those reported to be critical for $Ca_V\beta$ association with the $Ca_V\alpha_1$ high-affinity AID site[25,74] (Fig. 4E). Pull down assays using the AID peptide as bait showed that the actin-association-deficient $Ca_V\beta_2$ mutants retain the ability to associate with the AID (Fig. 4F and Supplementary Fig. 11C).

Together, these results indicate the key residues that contribute to the molecular interaction surface between $Ca_V\beta_2$ and actin.

## Integrating $Ca_V\beta_2$-derived XL-MS data for $Ca_V\beta_4$ and actin computational docking

$Ca_V\beta_4$, the central component of the $Ca_V$ complex in the human brain, also associates with actin[33]. The high degree of conservation in the primary sequence and domain structure among the members of the $Ca_V\beta$ family of proteins, including $Ca_V\beta_2$ and $Ca_V\beta_4$, anticipates a shared interaction interface with actin. Moreover, the amino acid sequence alignment between $Ca_V\beta_2$ and $Ca_V\beta_4$ shows that all four lysine residues involved in the XL-MS-derived distance restraints used for modeling of $Ca_V\beta_2$/actin interaction are conserved in $Ca_V\beta_4$ (Supplementary Fig. 12). We exploited this fact to guide molecular docking for $Ca_V\beta_4$ and actin using the analogous unambiguous restraints, together with bioinformatics-based predictions of interfacial residues to define ambiguous restraints.

Clustering of the $Ca_V\beta_4$–actin docking models, as done for $Ca_V\beta_2$, resulted in five clusters, out of which cluster #3 and #4 were discarded according to the average HADDOCK score of the top four structures (Supplementary Tables 4B and 5B–D). Clusters #1, #2 and #5 showed overlapping HADDOCK scores and selection among them was done based on their similarity with the experimentally-validated $Ca_V\beta_2$–actin model (Supplementary Table 4B). We calculated the average structure of the top four models for each cluster and compared with the average structure of the $Ca_V\beta_2$–actin complex, using the latter as reference for the alignment (Fig. 5A). The average structure of cluster #1 displayed the lowest l-RMSD (4.8 Å, compared to 12.7 Å and 12.2 Å for clusters #2 and #5, respectively). We also evaluated the i-RMSD to assess the similarity of the PPI surfaces by extracting the set of interface residues in common between the two $Ca_V\beta$/actin complex models, that displayed 86% sequence identity for the $Ca_V\beta$ interface residues. The average structure of cluster #1 also showed the lowest i-RMSD (2.7 Å versus 3.3 Å and 6.5 Å for clusters #2 and #5, respectively); thus, cluster #1 was chosen for further analysis.

This cluster was populated with 135 models, with l-RMSD and i-RMSD values of $3.8 \pm 3.4$ Å and $1.9 \pm 1.2$ Å, respectively, with respect to the best HADDOCK scored structure within this cluster. No violations of the upper cutoff distance of 35 Å between the Cα-Cα of two cross-linked residues was observed across the 135 models (Supplementary Fig. 13).

As for $Ca_V\beta_2$/actin, computational alanine scanning of the $Ca_V\beta_4$–actin interaction was carried out using the six predictors and the top four structures of the selected cluster #1, which showed i-RMSD $2.7 \pm 1.8$ Å. Several potential hotspots in $Ca_V\beta_4$ that are implicated in actin binding were identified (Supplementary Fig. 14), including residues analogous to the set of eight residues in $Ca_V\beta_2$ experimentally proven to affect actin binding and forming intermolecular contacts with a common subset of actin residues (Supplementary Table 7).

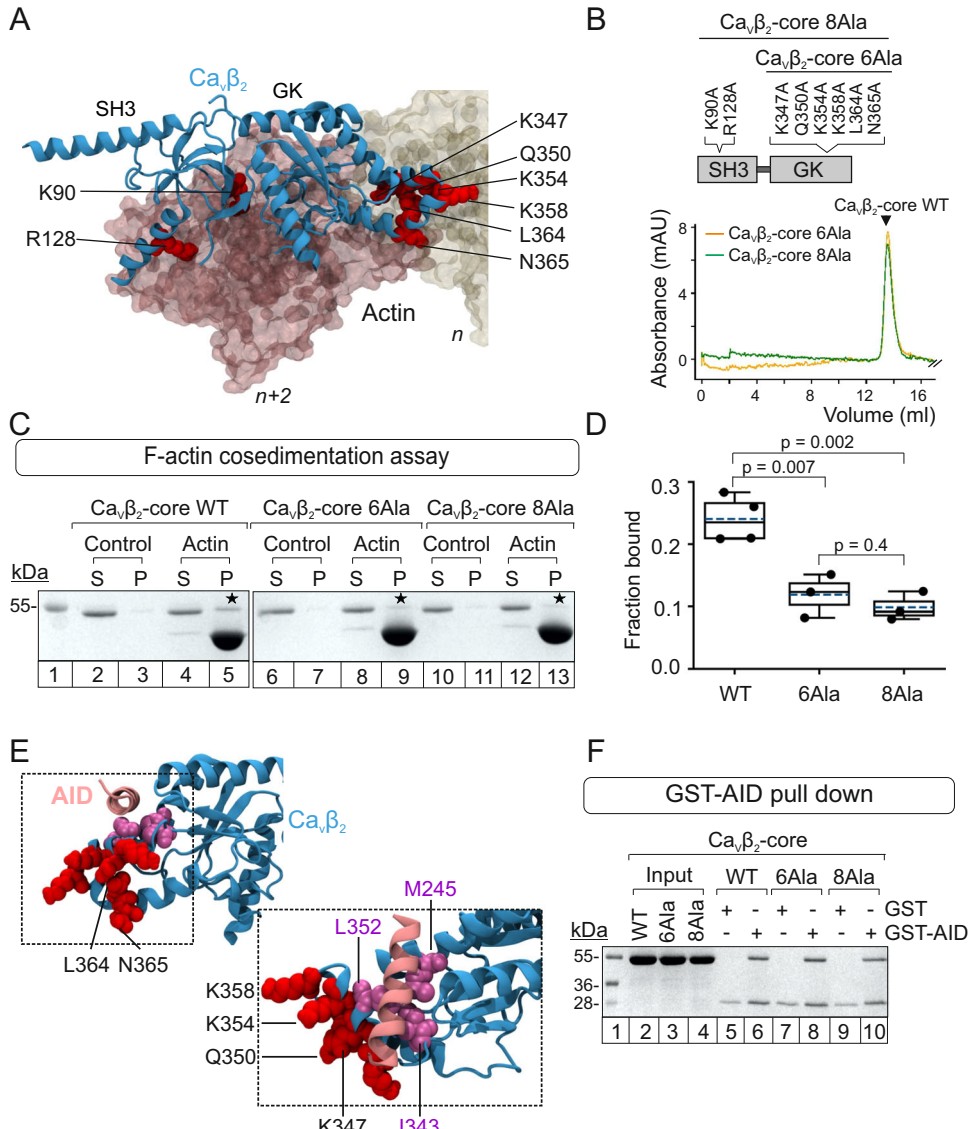

**Fig. 4 | Alanine substitution of predicted PPI hotspots decreases the in vitro affinity of Ca$_V$β$_2$ for actin but not for the AID peptide. A** Structure of Ca$_V$β$_2$ (blue ribbon), indicating the eight predicted hotspot residues (red spheres) mutated to alanine in the Ca$_V$β$_2$-core. Two actin subunits interacting with Ca$_V$β$_2$ are shown in space-filling mode, oriented from the barbed end ($n + 2$, pink) to the pointed end ($n$, yellow). **B** Linear diagram of the Ca$_V$β$_2$-core domain, indicating the positions of hotspot mutations and size exclusion chromatography profiles of the hotspot mutants, Ca$_V$β$_2$-core 6Ala (orange) and Ca$_V$β$_2$-core 8Ala (green). The arrow denotes the elution volume for Ca$_V$β$_2$-core WT. **C** SDS-PAGE of representative F-actin cosedimentation assays for the Ca$_V$β$_2$-core WT (lanes 2–5), Ca$_V$β$_2$-core 6Ala (lanes 6–9), and Ca$_V$β$_2$-core 8Ala (lanes 10–13). MW standard (lane 1). Ca$_V$β$_2$-core-derived proteins were centrifuged either alone (control) or after incubation with polymerized actin (actin). S, supernatant; P, pellet. Asterisks denote bands of Ca$_V$β$_2$-core-derived proteins that pelleted with actin. Each experiment was repeated three

times; full images are shown in Supplementary Fig. 11A. **D** Fraction of the indicated Ca$_V$β$_2$-core proteins bound to actin. Box plot shows the mean (dashed line), median (continuous line), interquartile range (25th–75th percentiles, box edges) and whiskers (1.5× interquartile range). Each dot represents an independent experiment; $p$-values, unpaired two-tailed $t$-test. **E** Ca$_V$β$_2$ complexed with the AID peptide (PDB 5V2P). Ca$_V$β$_2$ is depicted as a blue ribbon with hotspots for actin binding as red van der Waals spheres and the critical residues for Ca$_V$β association (L352, I343, and M245)[25,74] in purple. The AID is shown as a pink ribbon. **F** SDS-PAGE of a representative pull down assay using as bait GST, either alone or fused to AID (GST-AID) and as prey Ca$_V$β$_2$-core WT (WT), Ca$_V$β$_2$-core 6Ala (6Ala), or Ca$_V$β$_2$-core 8Ala (8Ala). MW standards (lane 1), Ca$_V$β$_2$-core proteins used as input (lanes 2–4), elution fractions from the GST (control, lanes 5, 7, and 9) and GST-AID pull down assays (lanes 6, 8, and 10). Each experiment was repeated three times; full images are shown in Supplementary Fig. 11C.

Since recombinant full-length Ca$_V$β$_4$ is not stable enough under the F-actin cosedimentation assay buffer[33], we resorted to the truncated Ca$_V$β$_4$ R482X naturally occurring variant associated with juvenile myoclonic epilepsy[13] for the experimental validation. Ca$_V$β$_4$ R482X appeared the best candidate to investigate differences in actin affinity produced by the hotspot mutations due to its amenability for recombinant expression and purification, and its higher apparent affinity for actin found in F-cosedimentation assays (Figs. 5B and Supplementary Fig. 15A).

Based on Ca$_V$β$_2$, we generated two analogous Ca$_V$β$_4$ R482X mutants containing alanine substitutions in all eight hotspot residues (Ca$_V$β$_4$ R482X 8Ala) or in the six located within the GK domain (Ca$_V$β$_4$ R482X 6Ala) (Fig. 5B). Size exclusion chromatography profiles confirmed the structural integrity of recombinant Ca$_V$β$_4$ R482X bearing no Ala substitution and the derived hotspot mutants (Fig. 5C). Moreover, intrinsic fluorescence measurements in actin cosedimentation assay buffer showed that the unfolding transition temperatures were comparable for all three proteins (Supplementary

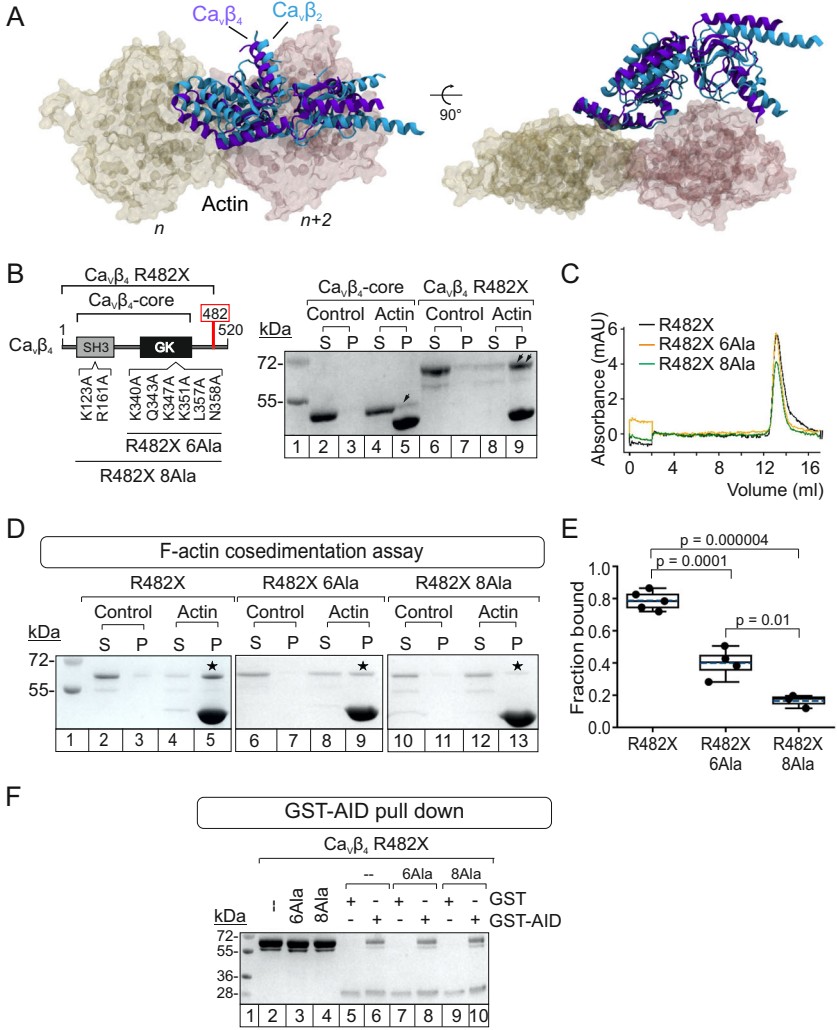

**Fig. 5 | Ca$_V$β$_4$ R482X hotspot mutations blunt association with actin.**
**A** Superimposition of the actin-bound structures of Ca$_V$β$_2$ (cyan ribbon) and Ca$_V$β$_4$ (purple ribbon). Each Ca$_V$β structure represents the average of the four best-scoring models of the selected docking cluster. For clarity, two actin subunits used as a reference for the alignment are shown in space-filling mode (yellow and pink). **B** Linear diagram and comparison of the in vitro actin-binding affinities by F-actin cosedimentation assay for Ca$_V$β$_4$-core (lanes 2–5) and Ca$_V$β$_4$ R482X (lanes 6–9). Numbers denote the positions of predicted hotspots substituted by alanine in the Ca$_V$β$_4$ R482X background according to human Ca$_V$β$_4$ (UniProtKB O00305-1). S and P, supernatant and pellet fractions, respectively. Single and double black arrows indicate the amount of corresponding protein that cosedimented with actin. The assays were repeated at least 3 times (Supplementary Fig. 15A). **C** Size exclusion chromatography profiles for Ca$_V$β$_4$ R482X (carrying no Ala substitutions) (black), Ca$_V$β$_4$ R482X 6Ala (orange), and Ca$_V$β$_4$ R482X 8Ala (green). **D** SDS-PAGE of F-actin

cosedimentation assays for Ca$_V$β$_4$ R482X (lanes 2–5), Ca$_V$β$_4$ R482X 6Ala (lanes 6–9), and Ca$_V$β$_4$ R482X 8Ala (lanes 10–13). S, supernatant; P, pellet. Asterisks indicate protein bands that pelleted with actin. Each experiment was repeated three times; full images are shown in Supplementary Fig. 15C. **E** Fraction of the indicated Ca$_V$β$_4$ R482X proteins bound to actin. In box plot: mean (dashed line), median (continuous line), interquartile range (25th–75th percentiles, box edges), and whiskers (1.5× interquartile range). Each dot represents an independent experiment; *p*-values, unpaired two-tailed *t*-test. **F** SDS-PAGE of a representative pull-down assay using as bait GST, either alone or fused to AID (GST-AID), and as prey Ca$_V$β$_4$ R482X (−), Ca$_V$β$_4$ R482X 6Ala (6Ala), or Ca$_V$β$_4$ R482X 8Ala (8Ala). Molecular weight standard (lane 1), Ca$_V$β$_4$ R482X-derived proteins used as input (lanes 2–4), elution fractions from the pull down assay using either GST (lanes 5, 7, and 9) as control or GST-AID (lanes 6, 8, and 10) and the indicated proteins. Each experiment was repeated three times; full images are shown in Supplementary Fig. 15D.

Fig. 15B). However, the ability of Ca$_V$β$_4$ R482X hotspot mutants to bind actin was dramatically reduced. Whereas > 80% of Ca$_V$β$_4$ R482X cosedimented with actin, only 40% of Ca$_V$β$_4$ R482X 6Ala and < 10% of Ca$_V$β$_4$ R482X 8Ala protein did so (Fig. 5D, E and Supplementary Fig. 15C). In the absence of actin, almost all of the three Ca$_V$β$_4$ R482X proteins remained in the supernatant. Similar to Ca$_V$β$_4$ R482X, both hotspot mutants retained the ability to associate with the AID (Figs. 5F and Supplementary Fig. 15D).

These results demonstrate the crucial contribution of the hotspot residues to Ca$_V$β$_4$–actin complex formation in vitro. They also highlight the very fine agreement between the computationally predicted Ca$_V$β hotspots for actin association and

experimental results and confirm the robustness of our integrative strategy employed to define the interaction surface between Ca$_V$β and actin.

**Ca$_V$β$_2$ hotspot mutations reduce the number of functionally available L-type Ca$_V$1.2 channels at the plasma membrane**
We next assessed the effect of expressing the actin-association-deficient Ca$_V$β$_2$ mutant in living cells. To this aim, full-length WT Ca$_V$β$_2$ or the mutant containing the eight experimentally validated hotspot mutations were fused to mRFP to generate Ca$_V$β$_2$-mRFP fusion constructs (hereafter referred to as Ca$_V$β$_2$ WT and Ca$_V$β$_2$ 8Ala, respectively). The WT protein is targeted to the plasma membrane[75–77], and

laser scanning confocal microscopy images of cells expressing either protein show that $Ca_V\beta_2$ 8Ala retains this ability (Fig. 6A). The degree of colocalization with the plasma membrane marker CellMask™ was not statistically different for both $Ca_V\beta_2$ WT and $Ca_V\beta_2$ 8Ala (Fig. 6B). Moreover, western blot analysis of crude lysates from cells expressing either $Ca_V\beta_2$ WT or $Ca_V\beta_2$ 8Ala showed comparable levels of protein expression (Fig. 6C). Therefore, the presence of hotspot mutations in $Ca_V\beta_2$ has no effect on either the cellular localization or expression of the protein.

To test the effect of $Ca_V\beta_2$ 8Ala on currents mediated by L-type $Ca_V1.2$ channels, we performed whole-cell recordings from cells coexpressing the $Ca_V1.2$ subunit and $Ca_V\beta_2$ 8Ala (in the absence of $Ca_V\alpha_2\delta$). These cells showed a significant decrease in the ionic current amplitude, with an average peak current reduction of about 53.0% compared with cells coexpressing $Ca_V1.2$ and $Ca_V\beta_2$ WT (Imax, $Ca_V1.2$/$Ca_V\beta_2$ WT, $-1.32 \pm 0.12$ nA; $Ca_V1.2$/$Ca_V\beta_2$ 8Ala, $-0.70 \pm 0.07$ nA; $p = 0.0001$) (Fig. 6D).

No appreciable voltage-dependent ionic currents were obtained for cells expressing the $Ca_V1.2$ subunit alone. Comparable curves for the fraction of activated channels versus voltage for the two channel subunit combinations were obtained indicating no major alterations in their open probabilities[32] (Fig. 6D). The minor change in voltage dependence of activation for $Ca_V1.2$/$Ca_V\beta_2$ 8Ala channels (rightward shift of by 4 mV in the half-maximal activation voltage) does not account for the drastic reduction in $Ca_V1.2$-mediated currents induced by $Ca_V\beta_2$ 8Ala.

The amplitude of the ionic current depends on the number of functionally available channels ($n$), the unitary conductance ($\gamma$), and the probability of the channel being in the open state ($Po$) (Eq. 5). We first addressed whether downregulation of $Ca_V1.2$-mediated currents is caused by decreased cell surface expression of the channel induced by $Ca_V\beta_2$ 8Ala. For this, we recorded gating currents from cells coexpressing $Ca_V1.2$ with either $Ca_V\beta_2$ WT or $Ca_V\beta_2$ 8Ala, as previously described[78–81]. Gating currents arise from the voltage-driven movement of charged residues within the voltage sensor−their integral over time provides the total gating charge ($Qon$) moved during the voltage step and is a direct quantification of the number of channels present at the plasma membrane[78–80,82,83]. The $Qon$ value was not statistically different for $Ca_V1.2$ coexpressed with either $Ca_V\beta_2$ WT ($363 \pm 33$ fC) or $Ca_V\beta_2$ 8Ala ($304 \pm 28$ fC; $p = 0.19$), suggesting that impaired $Ca_V\beta_2$/actin association does not compromise the cell surface targeting of $Ca_V1.2$ channels (Fig. 6E). However, gating current measurements compute the contributions of all channel proteins with an intact voltage sensor and do not distinguish between channels that can or cannot mediate ion permeation. Therefore, these results do not preclude the possible coexistence of ion conduction-competent and conduction-incompetent channels, with an increased proportion of the latter in the presence of the actin-association-deficient $Ca_V\beta_2$ mutant.

To address the possibility that ion conduction incompetent channels may contribute to the measured gating currents and to evaluate the unitary properties of the channels, we conducted stationary noise analysis of macroscopic currents. Noise analysis provides information about both the number of channels available for activation and the unitary conductance[84,85]. Conduction silent channels do not contribute to the macroscopic current nor to fluctuations in the macroscopic current around its mean value (variance).

Fluctuations in the whole-cell current in cells coexpressing $Ca_V1.2$ with either $Ca_V\beta_2$ WT or $Ca_V\beta_2$ 8Ala were computed for repeated current recordings at different voltages, as previously described[86]. For each cell, the ratio between current variance and mean current amplitude times driving force ($\frac{\delta^2}{\langle I \rangle (Vm-Vrev)}$) was plotted against whole-cell conductance ($\frac{\langle I \rangle}{Vm-Vrev}$). This yielded a straight line plot from which the unitary conductance $\gamma$ and the number of functionally available

channels $n$ are directly obtained from the $y$-intercept and slope, respectively (Eq. 6, Fig. 7A).

Data obtained from all analyzed cells are summarized in Fig. 7B, C. Box plots for all values calculated for $n$, $\gamma$, and maximal $Po$ ($Po_{max}$, calculated from Eq. 7) and bootstrap distributions[87] showed that $\gamma$ and $Po_{max}$ values for $Ca_V1.2$ channels complexed with $Ca_V\beta_2$ WT and those complexed with $Ca_V\beta_2$ 8Ala are statistically indistinguishable ($\gamma$, $3.9 \pm 0.1$ pS and $4.0 \pm 0.1$ pS; $p = 0.40$ and $Po_{max}$, $0.59 \pm 0.02$ and $0.60 \pm 0.02$; $p = 0.66$, for $Ca_V1.2$/$Ca_V\beta_2$ WT and $Ca_V1.2$/$Ca_V\beta_2$ 8Ala, respectively); however, the mean number of channels differs significantly ($Ca_V1.2$/$Ca_V\beta_2$ WT, $12233 \pm 1219$; $Ca_V1.2$/$Ca_V\beta_2$ 8Ala, $6580 \pm 899$; $p = 0.0002$) (Fig. 7C). The reduction of about 53.8% in the number of functionally available channels closely matches the 53% percentage reduction in the macroscopic peak current. Given that the unitary properties of the remaining channels were unaltered, these findings indicate that a $Ca_V\beta_2$ 8Ala-induced reduction in the number of conduction-competent channels is responsible and sufficient for the downregulation of $Ca_V$-mediated ionic currents.

Earlier experiments showed that pharmacological disruption of the actin filaments by cytochalasin D inhibits the interaction between actin and $Ca_V\beta_2$ in living cells[32]. If the effect of the actin-association-deficient $Ca_V\beta_2$ mutant is mediated by its impaired interaction with actin filaments, then it is expected that subsequent disruption of actin filaments has little further effect on $Ca_V$-mediated currents.

We thus treated cells coexpressing $Ca_V1.2$ with either wild-type $Ca_V\beta_2$ or the 8Ala hotspot mutant with the cytochalasin D actin filament disruptor and recorded ionic and gating currents (Supplementary Fig. 16). Time-lapse confocal fluorescence images of HEK293 cells stained for actin filaments demonstrate effective filament disruption under our experimental conditions. Cytochalasin D reduces ionic currents while preserving gating currents (Qon) from $Ca_V1.2$/$Ca_V\beta_2$ WT channels, as compared with non-treated cells ($p = 0.02$ and $p = 0.24$, respectively), resembling the effect of the $Ca_V\beta_2$ 8Ala hotspot mutant. Moreover, the average peak current amplitudes from cytochalasin D-treated cells coexpressing $Ca_V1.2$ with either $Ca_V\beta_2$ WT or $Ca_V\beta_2$ 8Ala are statistically indistinguishable from each other (Imax, $-0.81 \pm 0.18$ nA for $Ca_V1.2$/$Ca_V\beta_2$ WT and $-0.58 \pm 0.08$ nA for $Ca_V1.2$/$Ca_V\beta_2$ 8 Ala; $p = 0.25$) (Supplementary Fig. 16). These results demonstrate that disruption of the actin filaments blunts the current reduction mediated by the actin-association-deficient $Ca_V\beta_2$ mutant and indicate that the effect of this mutant is a direct consequence of its impaired interaction with actin filaments and not a general effect.

To examine a potential functional impact of $Ca_V\alpha_2\delta$, we expressed this subunit along with the $Ca_V1.2$/$Ca_V\beta_2$ channel core complex. Currents mediated by the $Ca_V1.2$/$Ca_V\beta_2$/$Ca_V\alpha_2\delta_1$ tripartite channel complex marginally differed from the ones obtained with the $Ca_V1.2$/$Ca_V\beta_2$ (Supplementary Fig. 17). $Ca_V\alpha_2\delta_1$ speeds the kinetics of activation of the channel complex, confirming its expression in the recorded cells. In the background of the tripartite channel complex, the actin-association-deficient $Ca_V\beta_2$ hotspot mutant continued downregulating current amplitude while preserving channel number.

## $Ca_V\beta_4$ R482X actin-association-deficient mutant preserves $Ca_V$-mediated ionic currents

It has been shown that $Ca_V\beta_4$ R482X exhibits normal nuclear localization, as compared with the wild-type $Ca_V\beta_4$[34]. Here, $Ca_V\beta_4$ R482X and the actin-association-deficient $Ca_V\beta_4$ R482X mutant, bearing the eight hotspot mutations (8Ala) displayed comparable nuclear localization and levels of expression (Figs. 8A, B and Supplementary Fig. 18A). The $Ca_V\beta_4$ R482X 8Ala mutant showed dramatically decrease in the in vitro binding to actin, but did not significantly alter the magnitude of whole-cell currents through L-type $Ca_V1.2$ channels or of the gating currents as compared with $Ca_V\beta_4$ R482X (Imax, $-0.29 \pm 40.2$ nA and $-0.27 \pm 32.4$ nA; $p = 0.8$ and Qon, $234 \pm 24$ fC and $259 \pm 37$ fC; $p = 0.60$

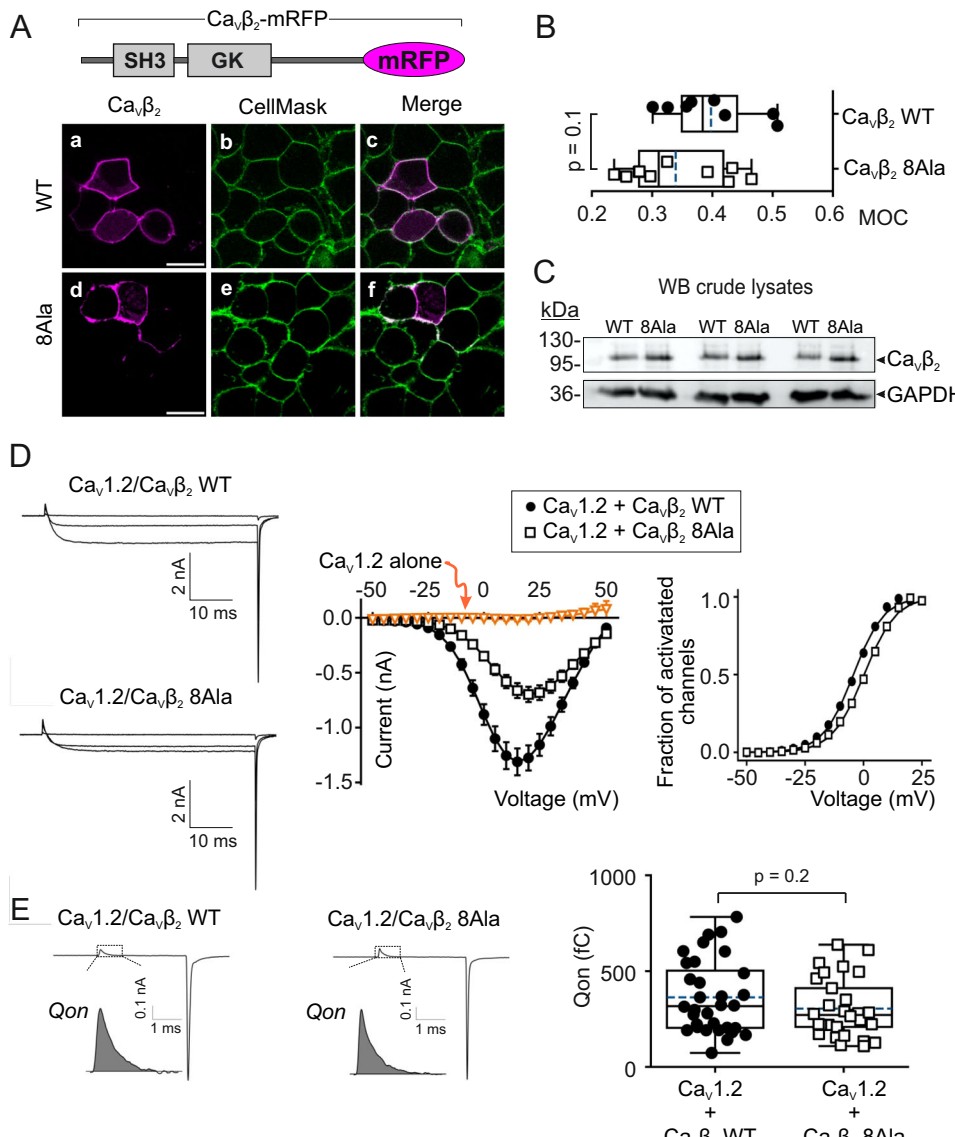

**Fig. 6 | An actin-association-deficient Ca$_V$β$_2$ mutant decreases Ca$_V$1.2-mediated ionic current but not gating currents. A** Schematic of the membrane-associated Ca$_V$β$_2$ fused to mRFP (Ca$_V$β$_2$-mRFP) and laser scanning confocal images of HEK293 cells expressing either the wild-type full-length Ca$_V$β$_2$ (WT) or the eight hotspot mutant (8Ala). Fluorescence images for the indicated Ca$_V$β$_2$ (a;d, magenta) and the plasma membrane marker CellMask™ (b;e, green); overlapping pixels appear as white (c;f). Scale bar, 15 μm (for all images). **B** Colocalization analysis between the indicated Ca$_V$β$_2$ protein and CellMask™. Box plot shows the mean (dashed line), median (continuous line), interquartile range (25th–75th percentiles, box edges) and whiskers (1.5 × interquartilee range). Each dot represents the Mander's overlap coefficient (MOC) for a field of view containing between 150 and 200 cells; *p*-value, two-tailed *t*-test. **C** Western blot (WB) of crude lysates from cells expressing either Ca$_V$β$_2$ WT or Ca$_V$β$_2$ 8Ala. Lysates from three separate experiments were probed with anti-Ca$_V$β$_2$ and anti-GAPDH antibodies. **D** Representative ionic current traces and

plots of ionic current versus voltage (I/V) and fraction of activated channels versus voltage obtained from HEK293 cells cotransfected with Ca$_V$1.2 and either Ca$_V$β$_2$ WT or Ca$_V$β$_2$ 8Ala. Only the traces induced by −40, +15, and +40 mV pulses are shown. Currents were elicited by voltages of −50 to +50 mV in 5 mV increments from a holding potential of −90 mV; Data are presented as mean ± SEM; number of recorded cells = 32, 29 and 5 for Ca$_V$1.2/Ca$_V$β$_2$ WT, Ca$_V$1.2/Ca$_V$β$_2$ 8Ala, and Ca$_V$1.2 alone, respectively. **E** Representative gating currents obtained from cells in panel (**D**) and total charge movement (*Qon*) calculated from the integral of the *On* gating current (shaded area) during the voltage step to the reversal potential for the carrier ion, Ba$^{2+}$. In box plot: mean (dashed line), median (continuous line), interquartile range (25th–75th percentiles, box edges) and whiskers (1.5 × interquartile range). Each dot represents an individual recorded cell; *p*-values, unpaired two-tailed *t*-test. Mean values ± S.E.M: 363 ± 33 fC (*n* = 32) and 304 ± 28 fC (*n* = 29) for Ca$_V$1.2/Ca$_V$β$_2$ WT and Ca$_V$1.2/Ca$_V$β$_2$ 8Ala, respectively.

for Ca$_V$1.2 complexed with Ca$_V$β$_4$ R482X and Ca$_V$β4 R482X 8Ala, respectively, Fig. 8C, D). Correspondingly, noise analysis revealed statistically comparable values of the number of functional channels (*N*, 2901 ± 509 and 3425 ± 437; *p* = 0.43), the single-channel conductance (γ, 3.4 ± 0.1 pS and 3.8 ± 0.3 pS; *p* = 0.15) and the maximal open probability (*Pomax*, 0.61 ± 0.02 and 0.63 ± 0.03; *p* = 0.43) between Ca$_V$1.2/Ca$_V$β$_4$ R482X and Ca$_V$1.2/Ca$_V$β$_4$ R482X 8Ala (Fig. 8E).

We also tested whether Ca$_V$β$_4$ R482X 8Ala affects currents mediated by the presynaptic, Ca$_V$2.2 (N-type) channel that directly mediates

the calcium influx for neurotransmitter release. We observed neither changes in the macroscopic current amplitude nor in the voltage-dependence of activation between Ca$_V$2.2 channels assembled wild-type Ca$_V$β$_4$, Ca$_V$β4 R482X or Ca$_V$β$_4$ R482X 8Ala (Supplementary Figs. 18B, C).

In conclusion, neither Ca$_V$β$_4$ R482X, which improves actin binding, nor the actin-association deficient Ca$_V$β$_4$ R482X mutant alters current amplitudes when coexpressed with either Ca$_V$1.2 or Ca$_V$2.2 pore-forming subunits. This highlights the functional specificity of Ca$_V$β$_2$ uncovered in this study.

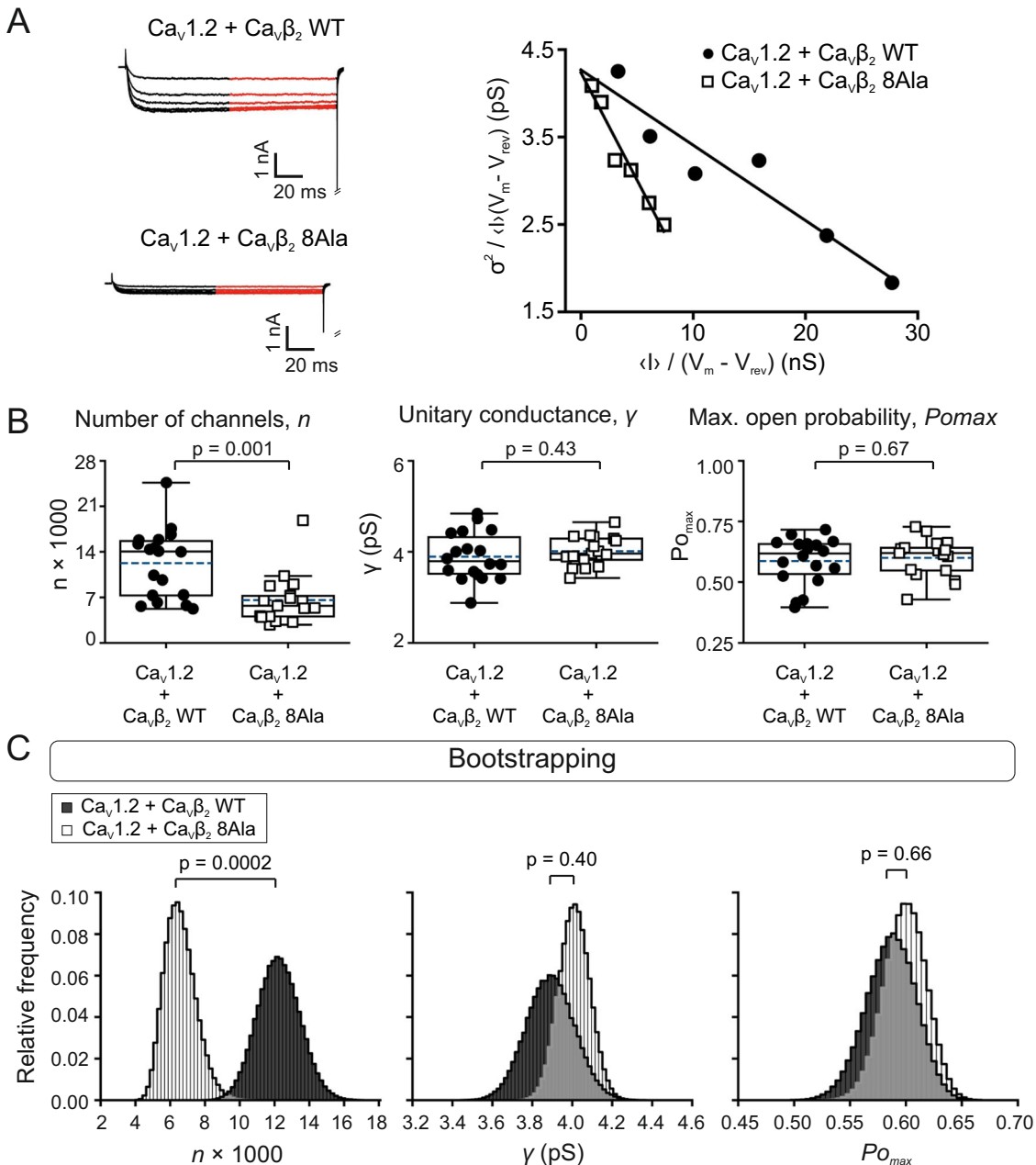

**Fig. 7 | The Ca$_V$β$_2$ actin association-deficient mutant reduces the number of functional Ca$_V$1.2 channels and does not alter the channel conductance or maximal open probability. A** Representative ionic current traces, with red lines delineating the time window used for noise analysis and plotting the relationship between variance and mean macroscopic current $\left(\frac{\sigma^2}{\langle I\rangle(Vm-Vrev)}\right)$ versus $\left(\frac{\langle I\rangle}{Vm-Vrev}\right)$ obtained at different voltages for two representative cells expressing Ca$_V$1.2 complexed with either Ca$_V$β$_2$ WT or Ca$_V$β$_2$ 8Ala. The number of functional channels $n$ and the unitary conductance γ were calculated from the slope and $y$-intercept of the linear fit, respectively (Eq. 6). **B** Box plots for $n$, γ, and $Po_{max}$. The box edges represent interquartile ranges (25th–75th percentiles), the continuous and dashed lines the median and mean values and the ranges of the whiskers denote 1.5× the

interquartile range. Each data point represents an individual recorded cell (number of cells = 18 and 17 for Ca$_V$1.2/Ca$_V$β$_2$ WT and Ca$_V$1.2/Ca$_V$β$_2$ 8Ala, respectively). $P$-values were determined by unpaired two-tailed $t$-test. The smaller unitary conductance value for Ca$_V$1.2 reported here is explained by the absence of Ca$_V$ agonists in our whole-cell current recordings[96]. **C** Bootstrap sample distributions for $n$, γ, and $Po_{max}$ (bootstrap sample N = 500,000). The mean values for $n$, γ, and $Po_{max}$ obtained from the corresponding bootstrap distributions are; $n$, 12233 ± 1219 for Ca$_V$1.2/Ca$_V$β$_2$ WT and 6580 ± 899 for Ca$_V$1.2/Ca$_V$β$_2$ 8Ala; γ, 3.9 ± 0.1 pS for Ca$_V$1.2/Ca$_V$β$_2$ WT and 4.0 ± 0.1 pS for Ca$_V$1.2/Ca$_V$β$_2$ 8Ala; and $Po_{max}$, 0.59 ± 0.02 for Ca$_V$1.2/Ca$_V$β$_2$ WT and 0.60 ± 0.02 for Ca$_V$1.2/Ca$_V$β$_2$ 8Ala; $p$-values were determined by bootstrap $t$-test.

## Discussion

PPIs are involved in almost every biological process and, since their aberrancy has been implicated in several human diseases, they are rapidly emerging as pharmacological targets in drug discovery[88–93]. Therefore, determining the contact surfaces for PPIs advances our understanding of the molecular mechanisms underlying cell function and dysfunction.

In this study, we identified the PPI interface between Ca$_V$β and actin using XL-MS and protein-protein docking experiments. We generated and evaluated the functional impact of Ca$_V$β hotspot mutants with a reduced capacity to bind actin. The Ca$_V$β–actin interface displays electrostatic complementarity that was confirmed by the sensitivity of the binding reaction to the ionic strength of the buffer. Electrostatic interactions between actin-binding proteins

and the negatively charged actin surface have been frequently observed[38,94].

In cells, coexpression of $Ca_V1.2$ and the $Ca_V\beta_2$ hotspot mutant (compared with WT protein) induced a significant decrease in the macroscopic current. A combination of gating current recordings and noise analysis showed no change in the charge movement, the voltage-dependence of activation and unitary properties of the channels, indicating that impaired $Ca_V\beta_2$-actin association still allows the insertion of normal $Ca_V1.2$ channels at the plasma membrane. The fact that the ionic current reduction induced by the actin-association-deficient $Ca_V\beta_2$ mutant occurs at identical channel numbers at the cell surface and single-channel properties necessarily implies the occurrence of an increased fraction of conduction-defective channels, which contribute to gating currents but not to ionic currents. Moreover, the extent of decrease in the number of functionally available $Ca_V1.2$ channels can fully account for the current downregulation.

Based on our data, we cannot unambiguously discount $Ca_V\beta_2$ 8Ala-mediated stabilization of the null gating $Ca_V1.2$ channel mode[95–99]; however, all experimental evidence published to date shows that $Ca_V\beta$ lacks the ability to arrest channels in a silent gating mode. This mechanism has been reported for $Ca_V1.2$ inhibition by the small GTPase Rem[100]. Instead, $Ca_V\beta$ facilitates pore opening by increasing the fraction of time that the channel spends in the open state within the high open gating mode without affecting the proportions of high and low $Po$ and null gating modes[98,99,101].

$Ca_V1.2$ channels at the cell surface are subjected to a highly dynamic endocytic turnover with an internalization rate of a few minutes and depending on an intact actin cytoskeleton[81,102]. Internalized $Ca_V1.2$ channels are effectively replaced by newly inserted $Ca_V1.2$ protein originating from the recycling rather than from the exocytic biosynthetic pathway on a relatively short timescale[102]. This mechanism is able to maintain a steady state level of ionic currents, despite of the constant endocytic loss of channels, for at least 20 h after inhibition of biosynthetic delivery[102]. The main finding of the present study is that an intact $Ca_V\beta$–actin interaction is strictly required to prevent the accumulation of conduction-defective channels at the plasma membrane, supporting the maintenance of stable $Ca_V1.2$ current amplitudes. We cannot unambiguously extrapolate this conclusion to the clearance of other conformationally impaired channels, since our electrophysiological measurements only detect alterations in channel permeation.

The potential source of conduction-defective $Ca_V1.2$ channels has been previously discussed[102]. Conformationally impaired plasma membrane proteins are continuously produced by local unfolding as a consequence of internal or external cellular stimuli, environmental stress, cellular insult and aging, and, in order to maintain proteostasis, they are sensed and endocytically removed[103–106]. The plethora of post-translational modifications and ligand interactions of $Ca_V1.2$ may alter the molecular environment of the channel and introduce local unfolding, including alterations in regions affecting ion permeation. Biosynthetic misfolded proteins that escaped cytosolic quality control checkpoints may also contribute to the pool of defective of channels at the plasma membrane.

A proposed quality control during $Ca_V1.2$ channel biogenesis is the interaction of the $Ca_V1.2/Ca_V\beta$ core complex with the endoplasmic reticulum membrane protein complex and $Ca_V\alpha_2\delta_1$ accessory subunit[107]. In our hands, coexpression of $Ca_V\alpha_2\delta$ along with the $Ca_V1.2/Ca_V\beta_2$ channel core complex marginally altered the biophysical properties of the channel complex and did not prevent the effect of the actin-association-deficient $Ca_V\beta_2$ mutant. Therefore, under our experimental conditions, $Ca_V\alpha_2\delta$ is neither mandatory for the channel functional assembly nor for its clearance at the plasma membrane.

We found no effect of $Ca_V\beta_4$ R482X and the actin-association hotspot mutant (as compared with wild-type $Ca_V\beta_4$) on either $Ca_V1.2$- or $Ca_V2.2$-mediated currents and no changes in their subcellular distribution. Previous reports also showed a fairly modest increase in $Ca_V2.1$ currents with $Ca_V\beta_4$ R482X in *X. laevis* oocytes, and nuclear localization comparable to the wild-type subunit[13,34]. Thus, these results did not explain the pathophysiology behind the diseased condition and led to the suggestion that a non-channel-related function of $Ca_V\beta_4$ (i.e, independent of its modulatory role on $Ca_V\alpha_1$ activity) underlies the neurological dysfunction.

Compared with the WT $Ca_V\beta_4$-core, the naturally occurring $Ca_V\beta_4$ R482X mutant associated with epilepsy appears to have a higher in vitro affinity for actin. Unfortunately, a direct comparison of the actin affinities of full-length $Ca_V\beta_4$ and $Ca_V\beta_4$ R482X in vitro was not viable because the high salt concentration required to stabilize the former precludes actin binding. In the future, it will be interesting to investigate whether an aberrant $Ca_V\beta_4$ R482X–actin interaction causes neuronal hyperactivity and hyperexcitability. This hypothesis is now testable thanks to the actin-association deficient $Ca_V\beta_4$ R482X mutant designed in the present work. It is also worth mentioning that the human $Ca_V\beta_4$ Q343E variant, annotated as potentially linked to idiopathic generalized epilepsy, is a substitution on an actin-binding hotspot identified in our work (Supplementary Table 8). Moreover, several others non-synonymous single nucleotide polymorphisms at the actin-binding hotspots in $Ca_V\beta_4$ and $Ca_V\beta_2$ have been reported, though their clinical significance is still unknown. The identification of $Ca_V\beta$-actin PPI may provide insights on factors contributing to the unresolved etiology of $Ca_V\beta$-associated disorders.

As expected from the high degree of sequence homology and similarity of three-dimensional structures among $Ca_V\beta$s, our data show conservation of the PPIs between actin and the two $Ca_V\beta$ proteins, $Ca_V\beta_2$ and $Ca_V\beta_4$. The lack of crystallographic structural data for the variable N- and C-terminal and HOOK regions within $Ca_V\beta$ prevents an assessment of the contribution of these segments to the interaction. It has been proposed that context-specific PPI networks determine the functional outcome across cellular types and that intrinsically disordered regions mediate molecular recognition in PPI networks[108,109]. Therefore, the unstructured variable regions of $Ca_V\beta$ may also regulate protein interactions and, thus, the specific biological outcome. Within this context, the clearance of $Ca_V1.2$ conducting-defective channels mediated by $Ca_V\beta_2$ expands the list of its specific functions that are not common to all members of the family[2,4].

Our results have identified the molecular interaction surface between $Ca_V\beta_2$ and actin responsible for clearing corrupted $Ca_V1.2$ channels from the plasma membrane (Fig. 9). A surveillance mechanism to monitor the conformational state of $Ca_V1.2$ at the cell surface is essential for proper calcium signaling and for maintaining proteostasis and the integrity of the cell membrane. Malfunctioning protein clearance and the consequent accumulation of damaged proteins are hallmarks of aging and neurodegenerative conditions[110,111]. The molecular model for $Ca_V\beta$–actin interaction presented here increases the potential for drug-targetable PPI interfaces in $Ca_V$-associated channelopathies and age-related diseases[93,112,113].

## Methods

### cDNA constructs and chemicals

Rat $Ca_V\beta_2$ and human $Ca_V\beta_4$ (corresponding to UniProtKB accession numbers Q8VGC3-2 and O00305-1, respectively) were used in this study. The pRSETB plasmids (Invitrogen, Carlsbad, CA, USA) containing the core regions of $Ca_V\beta_2$ (residues 23–422) and $Ca_V\beta_4$ (residues 50–408) were described previously[21,32,33]. The pRSETB vector containing the $Ca_V\beta_4$ R482X epilepsy-causing mutant (residues 1–431) was generated by overlapping PCR and standard molecular biology methods using the pRSETB $Ca_V\beta_4$ vector as a template. All recombinant proteins were expressed with a 6xHistidine tag fused to the N-terminus. $Ca_V1.2$-mNeonGreen (rabbit $Ca_V1.2$; UniProtKB accession number P15381), $Ca_V2.2$-GFP (human $Ca_V2.2$; UniProtKB Q00975-1) and the $Ca_V\beta_2$-mRFP cDNA constructs used for electrophysiology experiments were

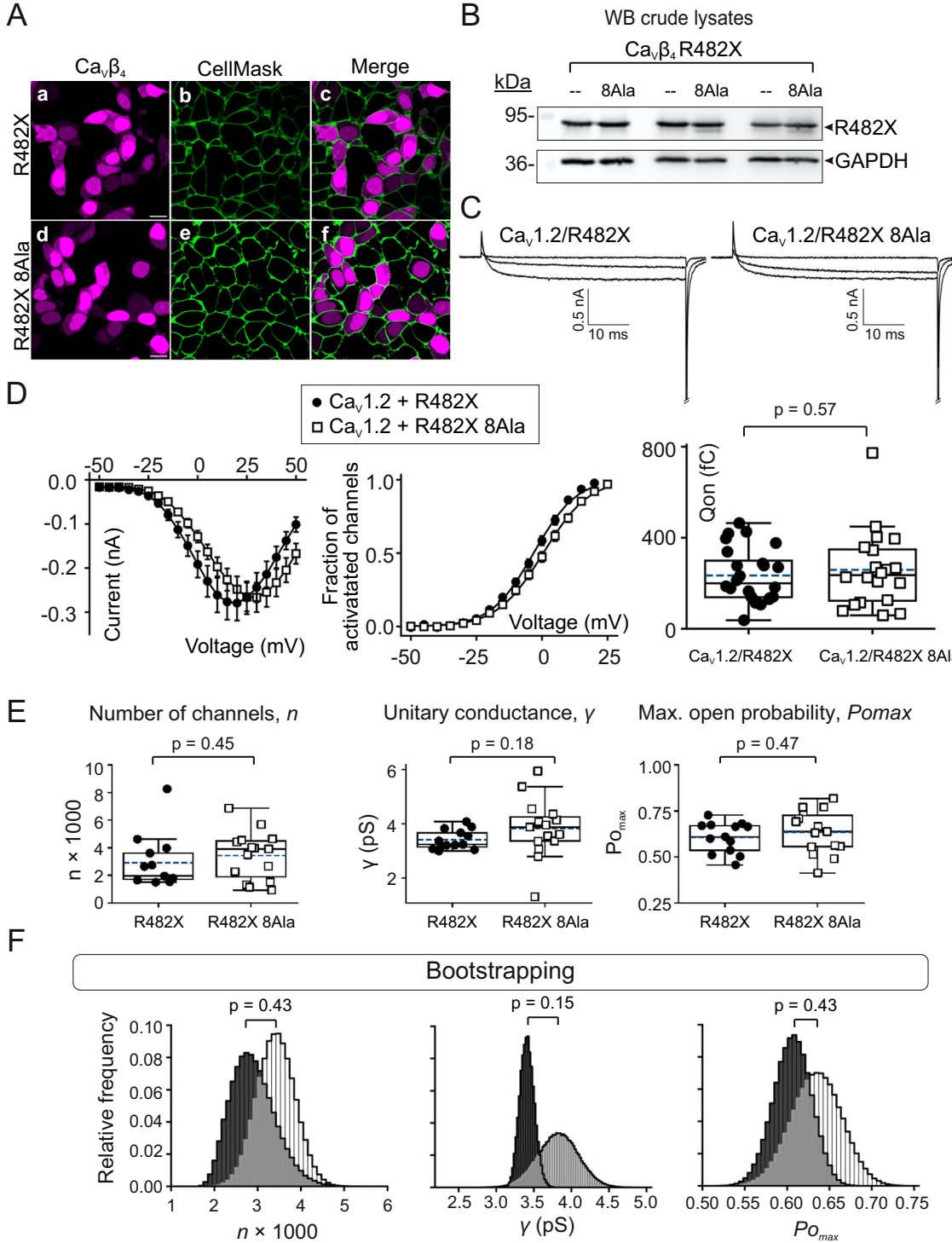

**Fig. 8 | Ca$_V$β$_4$ R482X hotspots mutant keeps Ca$_V$1.2-mediated ionic current intact. A** Representative laser scanning confocal images of HEK293 cells expressing either Ca$_V$β$_4$ R482X (a;c) or Ca$_V$β$_4$ R482X 8Ala (d;f) fused to mCherry. Plasma membrane was stained with CellMask™ (b;e, green). Merged images (c;f). Scale bar, 15 μm (for all images). **B** Western blot of three separate crude lysates from cells expressing either Ca$_V$β$_4$ R482X (--) or the 8Ala mutant (8Ala) probed with anti-Ca$_V$β$_4$ and anti-GAPDH antibodies. Uncropped images are shown in Supplementary Fig. 18A. **C** Representative ionic current traces (induced by −35, +20, and +45 mV pulses) from cells co-expressing Ca$_V$1.2 with either Ca$_V$β$_4$ R482X or Ca$_V$β$_4$ R482X 8Ala. **D** Plots of ionic current and fraction of activated channels versus voltage, and total charge movement (Qon) from the indicated channel subunit combinations. Ionic currents were elicited by voltages of −50 to +50 mV in 5 mV increments from a holding potential of −90 mV; data are presented as mean ± SEM. In the box plot, the edges represent interquartile ranges (25th–75th percentiles), the continuous and dashed lines, the median and mean values and whiskers denote 1.5× the inter-quartile range. Each data point represents an individual recorded cell. Qon mean values ± SEM: 234 ± 24 fC and 259 ± 37 fC (number of cells = 24 and 20) for Ca$_V$1.2/Ca$_V$β$_4$ R482X and Ca$_V$1.2/Ca$_V$β$_4$ R482X 8Ala, respectively. **E** Box plots for $n$, $γ$, and $Po_{max}$ for the indicated channel subunit combinations with interquartile ranges (25th–75th percentiles), median (continuous line), mean (dashed lines) and whiskers (1.5× the interquartile range). Each individual data point represents an individual recorded cell (number of cells = 13 and 15 for Ca$_V$1.2/Ca$_V$β$_4$ R482X and Ca$_V$1.2/Ca$_V$β$_4$ R482X 8Ala, respectively); $p$-values, unpaired two-tailed $t$-test. **F** Bootstrap distributions from the data shown in panel (**E**). Bootstrap sample N = 500,000. The mean values for $n$, $γ$, and $Po_{max}$ are: $n$, 2901 ± 509; $γ$, 3.4 ± 0.1 pS and $Po_{max,}$ 0.61 ± 0.02 for Ca$_V$1.2/Ca$_V$β$_4$ R482X, and $n$, 3425 ± 437; $γ$, 3.8 ± 0.3 pS and $Po_{max}$ 0.63 ± 0.03 for Ca$_V$1.2/Ca$_V$β$_4$ R482X 8Ala; $p$-values, bootstrap $t$-test.

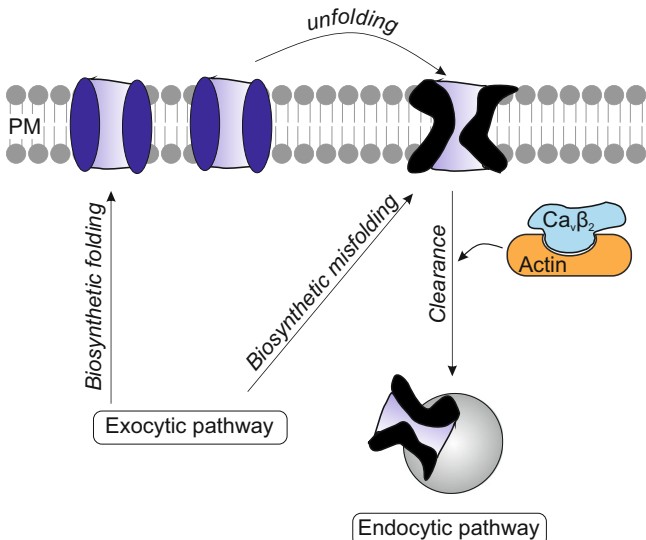

**Fig. 9 | Model for the clearance of corrupted Ca$_V$1.2 channels from the plasma membrane.** Conformationally impaired Ca$_V$1.2 proteins at the plasma membrane can be generated by diverse cellular stimuli, which change the channel environment leading to local unfolding, or by exocytic insertion of misfolded channels that evade cytosolic quality control (biosynthetic misfolding). Ca$_V$β–actin interaction is mandatory for the endocytic removal of corrupted Ca$_V$1.2 channels, which prevents their build-up and maintains stable Ca$_V$1.2 currents, but not for the insertion of functionally folded channels. PM, plasma membrane.

described previously[32,33,81]. Rabbit skeletal muscle actin protein (Uni-ProtKB accession number P68135) and the actin-binding protein Biochem kit were purchased from Cytoskeleton (Denver, CO, USA). The mass spectrometry cleavable cross-linkers DSBU and DSSO were purchased from Thermo Fischer Scientific (Langerwehe, Germany). Key reagents and antibodies are listed in Supplementary Table 9.

## Mutagenesis
The Ca$_V$β$_2$-core 8Ala, Ca$_V$β$_2$-core 6Ala, Ca$_V$β$_4$ R482X 8Ala, and Ca$_V$β$_4$ R482X 6Ala constructs were generated by overlapping PCR and standard molecular biology methods using the pRSETB Ca$_V$β$_2$-core and pRSETB Ca$_V$β$_4$ R482X vectors as templates for the Ca$_V$β$_2$ and Ca$_V$β$_4$ mutants, respectively. Ca$_V$β$_2$ 8Ala-mRFP was inserted into the p156rrl vector in a similar way as for pRSETB Ca$_V$β$_2$-core 8Ala, but using the p156rrl Ca$_V$β$_2$-mRFP plasmid as a template. Likewise, Ca$_V$β$_4$ R482X 8Ala-mCherry was inserted into the Fsy1.1 GW vector using Fsy1.1 GW Ca$_V$β$_4$ R482X mCherry plasmid as a template.

## Expression and purification of Ca$_V$βs
Recombinant Ca$_V$β subunits were expressed in *E. coli* and purified from the soluble fraction of crude lysate using Ni$^{2+}$-affinity chromatography (HisTrap™, Cytiva), followed by size exclusion chromatography using a Superdex-200 16/60 column (Cytiva), as described previously[32,33]. Both chromatographic steps were done using a Fast Performance Liquid Chromatography (FPLC) ÄKTA™ system (ÄKTA™ pure, Cytiva). Fractions eluted from the Superdex-200 column equilibrated with GF-150 buffer (150 mM NaCl, 20 mM phosphate, pH 8) for Ca$_V$β$_2$-core WT and mutants or GF-300 buffer (300 mM NaCl, 20 mM phosphate, pH 8) for Ca$_V$β$_4$-core, Ca$_V$β$_4$ R482X, and the alanine mutants were pooled, concentrated, flash frozen, and stored at −80 °C until use. Rabbit skeletal muscle actin protein was prepared and polymerized according to the manufacturer's instructions (Cytoskeleton).

## Chemical cross-linking and protein digestion
Ca$_V$β$_2$-core and polymerized actin were cross-linked. For this, Ca$_V$β protein in GF-150 buffer was centrifuged at 150,000 × *g* for 45 minutes

at 4 °C to remove aggregates. The supernatant was transferred to a new tube and stored for cross-linking reactions. Ca$_V$β (5 μg) and actin (5 μg) were mixed and incubated at RT for 30 minutes to enable interaction. A molar excess of 5× for DSBU or 2.5× for DSSO was added and the reaction mix was incubated for 60 minutes at RT. The reaction volume was 50 μl. Control reactions were performed by incubating each protein alone with and without cross-linker and both proteins together without cross-linker. The reactions were then quenched by the addition of Tris-HCl pH 8 to a final concentration of 50 mM. The cross-linked products were then analyzed by SDS-PAGE. After staining the gel with R-250 Coomassie blue, the desired bands were excised, dehydrated, and stored at -20 °C until use. The protein bands were cut into small pieces and reduced in 20 mM dithiothreitol for 30 minutes at 56 °C, cooled to RT, and then alkylated with 10 mM iodoacetamide for 30 minutes in the dark. Samples were then digested overnight at 37 °C with MS-grade approved trypsin (SERVA, Germany) using an enzyme:protein ratio of 1:50. Peptides were desalted using home-made C18 STAGE-Tips[114].

## LC-MS/MS analysis
Digested peptides were analyzed using an EASY-nanoLC coupled to a Q Exactive Plus (DSBU samples) or Orbitrap Eclipse mass spectrometer (DSSO samples) (Thermo Fisher Scientific) with a nanospray source. Peptides were separated in a 50 cm long, self-packed column containing 1.9 μm C18-AQ Reprosil-Pur beads (Dr. Maisch) and the column temperature was maintained at 50 °C in a column oven (PRSO-V2). Peptides were eluted with an effective solvent gradient of 2–35% buffer B within 30 or 60 minutes for DSBU and DSSO, respectively.

To identify DSBU/DSSO cross-links, digested proteins were analyzed using a higher-energy collision dissociation and collision-induced dissociation-acquisition method. Full scans were recorded at resolution of 70,000 and a scan range of 375–1750 m/z (AGC 3.6e6, max injection time 20 ms). MS/MS scans were recorded at a resolution of 35,000 and a scan range of 200–2000 m/z (AGC 5e5, max injection time 115 ms for CID isolation width 1.9 m/z). Singly and doubly charged ions were excluded from fragmentation because cross-linked peptides tend to occur at a charge state of 3+ or above. All selected precursors were excluded from fragmentation for 20 seconds.

## Cross-link identification
Raw MS data were converted to the *mzML* format with ProteoWizard Toolkit version 3[115]. Cross-linked peptide fragments were subsequently identified using Merox v.2.0, MetaMorpheus v.1.0.2, and MaxLynx v.2.2.0.0[39–41]. Fasta sequence files for Ca$_V$β$_2$-core (residues 23–422) and actin (corresponding to UniProtKB accession numbers Q8VGC3-2 and P68135, respectively) were provided and searches were performed using the following settings: 10 ppm as precursor, 20 ppm as fragment ion precision, CSM-level false discovery rate of 0.01, fixed modification of cysteine carbamidomethylation, and variable modifications of N-terminus acetylation and methionine oxidation, with up to three missed tryptic cleavages allowed. Contaminant protein databases provided by the software packages were included in the searches.

## Cross-link filtering
The identified unique intermolecular cross-links identified by LC-MS/MS were examined on the DisVis webserver (version 2.3.0) to check the consistency of the cross-link dataset and quantify the information content of the corresponding distance restraints[42,43]. The identified cross-linked residues mapped to a narrow cluster on Ca$_V$β$_2$ and to two discrete patches on different accessible surfaces of the actin monomer. The latter forms a continuous patch exposed across two adjacent actin subunits on one surface of an actin protofilament. Hence, the input files used for DisVis analysis were two adjacent actin subunits (*n* and *n + 2* or chains A and C, respectively, from PDB 5OOE) as the

fixed chain and Ca$_V$β$_2$ (PDB 5V2P) as the scanning chain[113,116]. Distance restraints were defined considering that the Ca$_V$β$_2$/actin cross-links can be formed with either monomer of two adjacent actin subunits. The allowed distance range of these restraints was set to 0–30 Å to account not only for the lengths of the Lys side chains and the cross-linker, but also for conformational dynamics[117,118] and potential errors in the side chain crystallographic position[55]. The same upper bound distance was used for all XL-MS-derived distance restraints, as some inter-XLs were identified with both DSSO and DSBU. DisVis filtering was applied iteratively: in each step, restraints with a high probability of being false positives were discarded and the remaining restraints were used as subsequent input for the next DisVis iteration. This multistep filtering procedure resulted in a set of self-consistent distance restraints from the XL-MS data that were used to guide protein-protein docking with the HADDOCK 2.4 webserver[44,45]. Structural analysis and visualization were carried out with Visual Molecular Dynamics, VMD (version 1.9.3)[119].

## Bioinformatics predictions

The XL-MS derived distance restraints are long range (~30 Å for the DSSO and DSBU cross-linkers). Therefore, additional information on the putative interfaces was used to guide docking, i.e., from bioinformatics predictions[118,120,121]. First, we used NACCESS (version 2.1.1) to identify residues with solvent accessibility of ≥ 40% in each protein[122]. Then, the list of NACCESS residues was filtered in DisVis using complete scanning of the dimeric actin and Ca$_V$β$_2$ structures and the set of identified self-consistent XL-MS derived distance restraints[42,43]. Residues predicted to have an interaction fraction of > 0.5 in each protein were kept. However, for Ca$_V$β$_2$ such filtering yielded only putative interacting residues in the GK domain because all XL-MS derived distance restraints were located in this domain. Therefore, prediction of additional putative protein-protein interface residues in the Ca$_V$β$_2$ SH3 domain and in actin was carried out using the Consensus Prediction Of interface Residues in Transient complexes (CPORT) webserver (version 0.2.0)[123]. The list of CPORT-predicted actin residues was manually curated to select residues facing the same side of the filament as the cross-linked residues. The manually curated list of CPORT residues for Ca$_V$β$_2$ SH3 and actin, and the DisVis-filtered list of NACCESS residues for Ca$_V$β$_2$ GK and actin were used as active residues for the information-driven docking of both proteins.

## Protein-protein docking

The X-ray structure of rat Ca$_V$β$_2$ (PDB 5V2P)[113] or rat Ca$_V$β$_4$ (PDB 1VYV)[26] and a dimeric structure of rabbit skeletal muscle actin (extracted from PDB 5OOE) were used for XL-MS-guided protein-protein docking on the HADDOCK 2.4 webserver[44,45]. The sequence identity between rat (used for docking) and human Ca$_V$β$_4$ (used for experimental assays) is 98.8%. Distance restraints were defined using a hybrid protocol. Unambiguous restraints were defined using the DisVis-filtered distance restraints between Ca$_V$β$_2$ and actin, allowing a Cα-Cα distance between cross-linked residues of 0–30 Å[55,117,118]. The same upper bound distance was used for all XL-MS-derived distance restraints, as some inter-XLs were identified with both DSSO and DSBU. In addition, ambiguous restraints were defined using the bioinformatics-predicted residues as active residues, as explained above[42,43] and a center of mass restraint was included to enforce contact between the two partners. The docking calculation comprises three stages: (i) rigid body docking to generate 1000 initial protein-protein complex structures; (ii) semi-flexible simulated annealing simulation with flexible interface residues to refine the 200 best-scored poses from stage (i); and (iii) a final refinement of those 200 poses with a 2.5 ns molecular dynamics simulation in water in which the two proteins are fully flexible. The final 200 docking structures were clustered based on the (positional) interface ligand RMSD. This i-l-RMSD was calculated by first superimposing the docking structures on the flexible interface backbone atoms of actin and then calculating the RMSD on the flexible interface backbone atoms of Ca$_V$β,

with the flexible interface defined by default as all residues that make intermolecular contacts within 5 Å, considering all docking models. HADDOCK default clustering parameters were used, specifically an i-l-RMSD cutoff value of 7.5 Å and a number of neighbors of 4. The clusters thus obtained were ranked based on the average HADDOCK score of their top four members. Such HADDOCK score was calculated as $1.0\,E_{vdw} + 0.2\,E_{elec} + 1.0\,E_{desol} + 0.1\,E_{air}$, where the terms $E_{vdw}$ and $E_{elec}$ represent the van der Waals and electrostatic intermolecular energies, $E_{desol}$ the desolvation energy and $E_{air}$ the distance restraints energy. The latter includes violations of both the unambiguous restraints (i.e., the XL-MS-derived distance restraints) and of the ambiguous restraints (here based on bioinformatics predictions).

Besides the i-l-RMSD used for clustering, we further assessed model similarity by calculating additional RMSD-based metrics[46,47,124]. The global RMSD is calculated using the alpha carbon atoms of all residues of both proteins and thus provides information on the overall complex similarity of the docking models with respect to the best scored model (instead of only the Ca$_V$β residues predicted to form the PPI surface for the i-l-RMSD). The interface RMSD (i-RMSD) is the backbone RMSD of all interface residues of both proteins (defined using an interatomic distance of 10 Å), after superimposing these interfacial residues. Complementarily, the ligand RMSD (l-RMSD) is the backbone RMSD calculated on Ca$_V$β after fitting on actin. Whereas the i-RMSD provides information on the similarity of the PPI surfaces of both actin and Ca$_V$β, the l-RMSD gives an idea of the position and orientation of Ca$_V$β on the surface of actin. For all RMSD calculations described below, the reference structure is the top model with the best (lowest) HADDOCK score, unless otherwise stated.

The cluster with the best average HADDOCK score was selected for further analysis. In case of several clusters with HADDOCK scores within standard deviation of the top cluster, we compared the clusters with overlapping HADDOCK scores and selected the best one based on additional experimental information. For each cluster, the docking structures were checked for consistency with the XL-MS derived distance restraints by calculating the Cα-Cα distance for each Ca$_V$β–actin pair of the set of self-consistent inter-XLs. In addition, for the top cluster of the Ca$_V$β–actin docking simulation, we evaluated the residue contact frequency as the number of structures in the top cluster in which residue $i$ of one protein partner is within 5 Å of the other protein in the complex, and then normalized by the total number of structures in the top cluster. We assumed that residues with higher contact frequencies have a higher probability of being located at the protein-protein interface, as previously reported[123,125]. The electrostatic potential of each protein partner was calculated using the corresponding crystal structure and the APBS webserver (version 3.4.1)[126].

## Control docking simulations

To validate the integration of XL data in our protein-protein docking protocol, we performed several docking simulations without XL-MS-derived distance restraints. To ensure close proximity of the two proteins, these simulations made use of the ab initio docking options of HADDOCK, as summarized in Supplementary Table 6. In addition, to account for the ambiguity of the XL-MS data, which cannot distinguish whether the cross-linked residues on the actin surface come from either one or two actin subunits, we also performed a docking simulation with Ca$_V$β$_2$ and one single actin monomer. DisVis filtering was performed as described in section "Cross-link filtering", but using the structure of only one actin subunit as fixed chain (chain A from PDB 5OOE) and defining XL-MS-derived distance restraints with this single actin monomer. The distance restraints classified by DisVis as self-consistent were employed as unambiguous restraints in HADDOCK, together with ambiguous restraints based on bioinformatics analyzes. For the latter, active residues were defined as explained in section "Bioinformatics predictions",

using NACCESS[122] and CPORT[123] for prediction of putative protein-protein interface residues between monomeric actin and the GK and SH3 domains of Ca$_V$β$_2$, respectively.

## Computational alanine scanning

The top four structures of the selected HADDOCK cluster were analyzed using computational alanine scanning to predict putative hotspots (i.e., residues at the Ca$_V$β–actin interaction surface that most contribute to the binding energy of the complex). A set of six different webtools was used to calculate ΔΔG; three are based on empirical energy functions (Anchor, BUDE Alanine Scan, and Robetta), and three on machine-learning-trained statistical potentials (BeAtMuSiC, Mutabind2, and SAAMBE-3D)[67–73]. Residues were ranked according to the ΔΔG values averaged over the four docked structures analyzed (to account for protein flexibility) and then over the six webtools used (to obtain a consensus prediction).

The eight Ca$_V$β hotspots selected for experimental validation were further characterized using the PRODIGY webserver (version 2.2.2)[127,128]. Intermolecular contacts between the Ca$_V$β hotspots and actin residues were considered to be present if any of their heavy atoms is within a distance of 5.5 Å. As done for the computational alanine scanning analysis, the top 4 models of the first cluster were analyzed for each actin–Ca$_V$β docking calculation, in order to account for protein dynamics.

## In vitro F-actin cosedimentation assay

F-actin cosedimentation assays were performed according to the manufacturer's instructions (except that Tris-HCl was replaced by HEPES in the polymerization buffer), as previously described[32,33]. Actin was polymerized by incubation in actin-polymerizing buffer containing 0.2 mM CaCl$_2$, 50 mM KCl, 2 mM MgCl$_2$, 1 mM ATP, and 5 mM HEPES, pH 8.0. Recombinant Ca$_V$β proteins were incubated with polymerized actin in polymerization buffer (50 µl reaction volume), followed by high-speed centrifugation for 20 minutes at 150,000 × $g$ and 4 °C. The supernatant and pellet fractions were resolved by denaturing SDS-PAGE. Control reactions contained Ca$_V$β alone to quantify protein sedimentation in the absence of actin. Phalloidin was used in all experiments at a concentration of 25 µM to stabilize actin filaments. Raw intensities of protein bands were estimated by densitometry using Fiji ImageJ software (version 1.44p)[129]. The association of Ca$_V$β proteins with F-actin was reported as the fraction of bound Ca$_V$β, as calculated from the ratio of band intensities of pelleted over total protein (i.e., protein found in supernatant and pellet).

## Pull down assays

GST fused to the AID peptide (GST-AID) or GST alone (negative control) was used as bait in a standard GST pull down assay, as previously described[130]. Briefly, GST-AID or GST was immobilized onto glutathione-Sepharose beads and then incubated with the desired Ca$_V$β constructs at RT. After several washing steps, bound proteins were eluted with SDS-loading buffer and resolved by SDS-PAGE.

## Intrinsic fluorescence spectroscopy

Intrinsic fluorescence experiments were performed using a FS5 Fluorescence Spectrometer (Edinburgh Instruments, UK) equipped with a 500 µl cuvette coupled to a SC-25 temperature holder and a TC-1S temperature controller (Quantum Northwest, WA, USA). Excitation was done at 280 nm over a temperature range of 20–74 °C and emission spectra were recorded at 300–430 nm in 1 nm steps. For data analysis, the spectral center of mass (CM) was determined as follows, and plotted against the temperature:

$$CM_i = \frac{\sum_i F_i \lambda_i}{\sum_i F_i} \tag{1}$$

where $F$ is the fluorescence intensity and $\lambda$ the wavelength at a temperature $i$.

Transition temperatures were then obtained by non-linear regression fitting using a two-state transition model:

$$y = \frac{\alpha_n + \beta_n T}{1 + e^{\frac{4T_m(T - T_m)}{T\Delta T}}} + \frac{\alpha_d + \beta_d T}{1 + e^{\frac{4T_m(T_m - T)}{T\Delta T}}} \tag{2}$$

where $y$ is CM, $\alpha$ and $\beta$ are the $y$-intercept and slope of the native ($N$) or denatured ($D$) states, respectively, $T$ is the temperature, $T_m$ the is the transition temperature, $\Delta T$ is the width of the transition, and $e$ is the exponential function.

## Cell culture and transfection

HEK293 cells were cultured in Dulbecco's modified Eagle medium supplemented with 10% fetal bovine serum (Sigma-Aldrich, Darmstadt, Germany) and L-glutamine (2 mM), and incubated in a 5% CO$_2$ atmosphere. For electrophysiological experiments, cells were seeded onto 50 mm dishes and transfected using Lipofectamine 2000 (Thermo Fisher Scientific) following the manufacturer's instructions using 3 µg of Ca$_V$1.2-mNeonGreen and 1 µg of Ca$_V$β$_2$-mRFP (WT or 8Ala mutant) expression constructs, or 4 µg of Ca$_V$2.2-GFP with 2 µg of Ca$_V$β$_4$-mCherry (WT, R482X or R482X 8Ala). For western blot analysis, the same protocol was followed except that cells were transfected with only the wild-type and mutant Ca$_V$βs expression constructs. For laser scanning confocal fluorescence microscopy, HEK293 cells were seeded onto 35 mm glass-bottom dishes (µ-Dish ibidi, Ibidi, Gräfelfing, Germany) and transfected with 2 µg of the corresponding Ca$_V$β expression constructs using also Lipofectamine 2000 (Thermo Fisher Scientific).

## Western blot analysis of cell lysates

Cells were lysed using RIPA buffer supplemented with a protease inhibit cocktail (Sigma-Aldrich). Lysates were clarified by centrifugation and soluble proteins were resolved by SDS-PAGE. Proteins were then blotted onto nitrocellulose membrane (GE Healthcare, Life Science, Solingen, Germany) for 1 h. Membranes were blocked in 5% BSA in TBS buffer (10 mM Tris, 150 mM NaCl, pH 7.5) and incubated with the primary antibodies, anti-CACNB2 (Novus Biologicals, Wiesbaden Nordenstadt, Germany), anti-CACNB4 (Novus Biologicals, Wiesbaden Nordenstadt, Germany) and anti-GADPH (Cell Signaling, Leiden, Netherlands). After several washes, membranes were incubated with a goat anti-rabbit IgG HRP-conjugated secondary antibody (Thermo Fisher Scientific). Protein bands were detected using a chemiluminescent detection kit (SuperSignal West Femto Chemiluminescent Substrate, Thermo Fisher Scientific).

## Laser scanning confocal fluorescence microscopy

Confocal imaging of live HEK293 cells transiently transfected with Ca$_V$β$_2$-mRFP constructs (wild-type and 8Ala mutant) and Ca$_V$β$_4$-mCherry (R482X and R482X 8Ala mutant) was performed using an inverted Leica confocal microscope equipped with a 63× oil immersion objective lens. Images were acquired 18–24 h post-transfection. Before imaging, cells were stained with the plasma membrane marker CellMask™ Green (Thermo Fisher Scientific) following the manufacturer's instructions. mRFP and mCherry fusion proteins were excited using a 594-nm laser, and emission was detected in the 600–650 nm range. CellMask™ Green was excited at 514 nm, and emission was monitored in the 520–540 nm range. Colocalization analysis was conducted by calculating the Mander´s overlap coefficient using the JACoP plugin integrated into ImageJ 1.44p software, as previously described[130,131]. For each protein, 8–10 different fields of view, derived from two independent transfections and comprising 150–250 transfected cells, were analyzed. Statistical comparison between samples was performed using a two-tailed Student's $t$-test. All values are presented as mean ± SEM. For imaging actin filaments, cells

were seeded on coverslips and stained with Phalloidin-Atto 488 (Sigma-Aldrich) as described[102]. Briefly, control and cytochalasin D-treated cells were fixed with 4% PFA, and simultaneously permeabilized, blocked and incubated with phalloidin using 5% chemi-BLOCKER (Merck) supplemented with 0.05% Triton-X100. After several washes, the coverslips were mounted on glass slides using Aqua-Poly-Mount (Polysciences) and imaged the next day.

### Electrophysiology

Whole-cell patch clamp recordings of cells transiently coexpressing either $Ca_V1.2$-mNeonGreen or $Ca_V2.2$-GFP with $Ca_V\beta_2$-mRFP or $Ca_V\beta_4$-mCherry derivatives expression constructs were performed 24 h after transfection using the EPC-10 amplifier equipped with Patch-Master software (HEKA Elektronik, Stuttgart, Germany). Fusion of the channel subunits to the fluorescence proteins (mNeonGreen, mRFP and mCherry) enable the recognition of transfected cells. Recording and analysis were done blind with respect to the $Ca_V\beta$ construct used.

Ionic currents were elicited by 60 ms voltage steps from +50 mV to +70 mV in 5 mV increments from a holding potential of -90 mV and using $Ba^{2+}$ as the charge carrier. Gating currents were recorded during voltage steps near to the reversal potential for $Ba^{2+}$, as determined empirically by stepping to different potentials in 2 mV increments[79,81]. The total charge movement was obtained by integrating the *On* gating current elicited at the reversal potential for $Ba^{2+}$ over time. Borosilicate glass patch pipettes (Harvard Apparatus, Holliston, MA, USA) were pulled using a Sutter P-1000 puller (Harvard Apparatus) and fire-polished using a Narishige MF-830 microforge (Tokyo, Japan). Patch pipettes with an electrical resistance of 1.0–2.0 MΩ were used. Series resistance compensation was applied, resulting in a voltage error of > 5 mV. The pipette solution contained 135 mM cesium methane-sulfonate, 10 mM EGTA, 5 mM CsCl, 1 mM $MgCl_2$, and 10 mM HEPES, adjusted to pH 7.3 with CsOH. The extracellular recording solution contained 140 mM tetraethylammonium-$MeSO_3$, 10 mM $BaCl_2$, and 10 mM HEPES buffer, adjusted to pH 7.3 with tetraethylammonium hydroxide. Data were analyzed with Python version 3.8 using common libraries including NumPy, Pandas, Pyplot, SciPy, and Seaborn and presented as the mean ± SEM. Differences between data sets were analyzed using the *t*-test. For the pharmacological treatment with cytochalasin D (Sigma-Aldrich), cells expressing the corresponding $Ca_V1.2/Ca_V\beta_2$ complexes were incubated for 40 min with 10 μM of cytochalasin D, as described[32].

The voltage dependence of the relative open probabilities ($P_{open}(V)$) was calculated by normalizing steady-state current amplitudes to the maximal value ($I/Imax$) and fitted with a Boltzmann function to obtain the half-activation voltage ($V_{1/2}$) (Eq. 3).

$$P_{open}(V) = \frac{1}{1 + e^{\frac{-(V_m - V_{1/2})}{k}}} \tag{3}$$

where $V_m$ is the membrane potential, $V_{rev}$ is the reversal potential, and $k$ is the slope.

### Stationary noise variance analysis

Analysis of the variance ($\sigma^2$) of the mean macroscopic current ($\langle I \rangle$) of repeated current recordings obtained at different voltages was performed as previously described[86]. Currents were elicited by 150 ms voltage steps from +50 mV to +70 mV in 5 mV increments from a holding potential of −90 mV and using $Ba^{2+}$ as the charge carrier. The number of functional channels and unitary conductance were computed for each cell. Since the unitary (single-channel) conductance of $Ca_V$s does not change with voltage[95,132,133] and, therefore, the single-channel current ($i$) varies linearly with the driving force, $i$ is given by Eq. 4 and the ionic current recorded through a defined population of

ion channels by Eq. 5,

$$i = \gamma(V_m - V_{rev}) \tag{4}$$

$$I = n \times \gamma(V_m - V_{rev}) \times Po \tag{5}$$

where $n$ is the number of functionally available channels unitary channel conductance, $\gamma$ is the unitary conductance, ($V_m - V_{rev}$) is the driving force (where $V_m$ is the applied membrane potential and $V_{rev}$ the reversal potential), and $Po$ is the probability of the channel being in the open state.

In this case, the relationship between the variance and mean macroscopic current ($\langle I \rangle$) is given by Eq. 6 [86], and the number of functionally available channels $n$ and the unitary conductance $\gamma$ are obtained by fitting the data.

$$\frac{\sigma^2}{\langle I \rangle (V_m - V_{rev})} = -\frac{1}{n}\left(\frac{\langle I \rangle}{V_m - V_{rev}}\right) + \gamma \tag{6}$$

Having determined $n$ and $\gamma$, the maximum open probability of the channel ($Po_{max}$) is obtained from Eq. 7.

$$Po_{max} = \frac{I_{max}}{n \times i} \tag{7}$$

Values for the three parameters, $n$, $\gamma$, and $Po_{max}$, obtained from the original set of data were subjected to bootstrap iterations[87], in which random samples with replacements were taken from the original set of measurements to generate a synthetic distribution for each parameter. This procedure was repeated 500,000 times and the reported values are given by the mean value and SD. Differences between data sets were analyzed using a bootstrap *t*-test.

Electrophysiological data were analyzed using Python 3.8. Parameters estimated from the noise analysis were compared using a bootstrap *t*-test. All data are presented as mean ± SEM. Box-and-whisker plots show the interquartile range (IQR), median, and the 25th (Q1) and 75th (Q3) percentiles. Whiskers extend to 1.5 times the IQR below Q1 and above Q3. Normality was assessed using the Shapiro-Wilk test. Statistical significance between groups was assessed using a two-tailed *t*-test.

### Reporting summary

Further information on research design is available in the Nature Portfolio Reporting Summary linked to this article.

## Data availability

The following protein sequences were used in this study: rat $Ca_V\beta_2$, UniProtKB Q8VGC3-2 [https://www.uniprot.org/uniprotkb/Q8VGC3] human $Ca_V\beta_4$ UniProtKB O00305-1 [https://www.uniprot.org/uniprotkb/O00305] rabbit $Ca_V1.2$, UniProtKB human $Ca_V2.2$, UniProtKB Q00975-1 [https://www.uniprot.org/uniprotkb/Q00975] human $Ca_V\alpha_2\delta_1$, UniProt P54289-1 [https://www.uniprot.org/uniprotkb/P54289] rabbit skeletal muscle actin, UniProtKB P68135. Raw mass spectrometry and search data have been deposited to the ProteomeXchange Consortium via the PRIDE partner repository[134] with the dataset identifiers PXD053456 (DSBU) and PXD053481 (DSSO) respectively. Data required to reproduce the docking results and resulting $Ca_V\beta$-F/actin complex models, along with the source data underlying Supplementary Figs 4, 6, 9, 13 and 14 and Supplementary Tables 4 and 6, are available at Zenodo [https://doi.org/10.5281/zenodo.8276447]. The following experimental protein structures were used for the docking calculations: rabbit skeletal muscle actin, PDB 5OOE; rat $Ca_V\beta_2$, PDB 5V2P; rat $Ca_V\beta_4$, PDB 1VYV. The source data underlying Figs. 3E, 4D, 5E, 6B, D, E, 7A, B and 8D, E and Supplementary Figs. 2B, 11B, 15B, 16B, C, 17 and 18C are

provided as a source data file. Source data are provided with this paper.

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

## Acknowledgements

We thank Prof. Marcus Krüger and Dr. Andreas Schmidt (Cluster of Excellence Cellular Stress Responses in Aging-Associated Diseases, University of Cologne, Germany) for Orbitrap mass spectrometry measurements. We are very grateful to Prof. Christoph Fahlke, Prof. Alan Neely and Dr. Bassam Haddad for helpful discussions. FC was funded by the Networking doctoral candidate (Vernetzungsdoktoranden) program, Forschungszentrum Jülich awarded to BSS, MAP and PH. The FP7 WeNMR (project# 261572), H2020 West-Life (project# 675858), the EOSC-hub (project# 777536) and the EGI-ACE (project# 101017567) European e-Infrastructure projects are acknowledged for the use of their web portals, which make use of the EGI infrastructure with the dedicated support of CESNET-MCC, INFN-LNL-2, NCG-INGRID-PT, TW-NCHC, CESGA, IFCA-LCG2, UA-BITP, TR-FC1-ULAKBIM, CSTCLOUD-EGI, IN2P3-CPPM, CIRMMP, SURFsara and NIKHEF, and the additional support of the national GRID Initiatives of Belgium, France, Italy, Germany, the Netherlands, Poland, Portugal, Spain, UK, Taiwan and the US Open Science Grid.

## Author contributions

F.C. performed, designed, acquired, and analyzed all data collected for mass spectrometry, protein-protein docking, in vitro cross-linking, binding assays, western blots, and protein expression and purification experiments, and prepared the figures for the manuscript; V.L. designed, performed, recorded, and analyzed all electrophysiological experiments and prepared all the corresponding figures; E.M.L. designed the mutagenesis and captured, analyzed, and discussed the laser scanning confocal microscopy images; N.J. generated the cDNA constructs, expressed and purified recombinant proteins, and cultured cells; P.F.H. contributed to mass spectrometry discussions; B.S.S. designed, analyzed, and discussed all mass spectrometry data; M.A.P. designed, analyzed, and discussed all computational biology experiments; and P.H. conceived the project, designed the experiments, and wrote the manuscript.

## Funding

## Competing interests

The authors declare no competing interests.
