## [Transparent Peer Review file · Nature Communications]

Mapping the interaction surface between $\text{Ca}_v\beta$ and actin and its role in calcium channel clearance

Corresponding Author: Professor Patricia Hidalgo

Version 0:

Reviewer comments:

Reviewer #1

(Remarks to the Author)

This ms details actin binding to $\text{Ca}_v\beta 2$ and $\beta 4$. In the absence of a high resolution structure of the two proteins, the ms endeavors to probe how the two proteins $\text{cav}\beta$ and actin interact. The major part of the results details the protein-protein interaction interface between $\text{Cav}\beta$ and actin, using solid biochemical and biophysical methods, cross-linking, site directed mutagenesis, modelling etc.

Basic results concerning the interaction between β subunits and actin have been previously reported: Stölting, G. et al. Direct interaction of Ca_v with actin up-regulates L-type calcium currents in HL-1 cardiomyocytes. *The Journal of biological chemistry* 290, 4561-4572, doi:10.1074/jbc.M114.573956 (2015). Guzman, G. A., Guzman, R. E., Jordan, N. & Hidalgo, P. A Tripartite Interaction Among the Calcium Channel $\alpha 1$ - and β -Subunits and F-Actin Increases the Readily Releasable Pool of Vesicles and Its Recovery After Depletion. *Frontiers in cellular neuroscience* 13, 125, doi:10.3389/fncel.2019.00125 (2019).

Specific comments

1. From the biochemical and modelling results, a mutant $\beta 2$ subunit is designed which does not interact with actin. Functional studies are then performed, using control $\beta 2$ and the mutant. Fig 5 shows a reduction in ionic current due to the mutant $\beta 2$, but no change in gating current. Fig 6 shows a reduction in number of functional channels in the membrane, no change in single channel conductance, and a non-significant reduction in pomax (Fig 6B,C). However, the mean Pomax and the distribution of the Pomax appears very different, even if not statistically significantly reduced. In Fig 6B, the pomax does not appear to be normally distributed, from the relatively small number of values.
2. The point of using the $\beta 4$ R-X mutant is unclear to me. It is found to have a higher affinity for actin. In the original paper describing this mutation (Escayg et al, 2000), the ionic currents were found to be slightly larger for the mutant, but this result is not tested here. Indeed, no functional data are included for this mutant.
3. The model in Fig 7 shows actin being involved in quality control of “defective” calcium channels, but this model is not supported by the data presented. There is no information in the ms about what would constitute an endogenous defective channel, or on the endocytosis or degradation route of the “defective” channels. The mutant channels appear to be in the cell membrane to the same extent, from the gating charge measurements, and are not obviously internalized from the immunofluorescence results in Fig 5A, although the proportion of cells expressing the $\beta 2$ at the membrane appears less from Fig 5A.
4. End of Results “these findings suggest that the association between $\text{Ca}_v\beta$ and actin regulates a process that monitors, removes, and replaces conduction-defective $\text{Ca}_v1.2$ proteins, thereby preventing overpopulation of damaged channels at the cell surface”. This is very speculative, and the speculation about actin’s role in removal of defective channel proteins continues in the Discussion.
5. There is no attempt to use a conduction-defective calcium channel mutant which would be a way to examine these claims.
6. The title is similarly speculative and does not mention the main topic of the paper which is mapping the actin- $\text{Ca}_v\beta$ interaction interface.

Reviewer #2

(Remarks to the Author)

PPIs are involved in almost every biological process and, since their aberrancy has been implicated in several human diseases and pharmacological targets in drug discovery. Therefore, determining the contact surfaces for PPIs advances our understanding of the molecular mechanisms underlying cell function and dysfunction.

Authors identified the PPI interface between CaV β and actin using XL-MS and protein protein docking experiments. I was asked by the editor to specifically review the protein cross-linking analysis and therefore I would like to give a recommendation only for this part of the manuscript.

From a global perspective, I don't find the XL analysis very transparent and see a huge need for improvement in this respect. Major issues:

Nobody can determine an FDR with only 2 proteins. Therefore, all results are not trustworthy. Only one score is simply not enough to determine the quality of the MS/MS results. Above all, it is unclear why a certain score cutoff was chosen. The results of DSBU and DSSO are of totally different quality. The results of the individual analyses were not listed and presented transparently in the manuscript.

Minor issues:

One of the first experiments a bachelor's or master's student performs at xl is to determine the correct concentration of cross-linker. This was apparently not done here in a nature publication. This is exactly how the response time was not optimized. If this is too long, there are many artifacts.

Lack of transparency in the upload of data files (results, settings,...)

The workflow for the analysis was not described exactly (e.g. size exclusion, negative controls SDS page,...) or missing negative controls.

I would therefore recommend contacting a cross-linking expert to validate the results again. Transparent presentation of all findings (even if they are negative) in tabular form is essential.

I would therefore recommend assessing this publication with a major revision for the XL part, because it is simply very important for modeling that the XL results are correct.

Reviewer #3

(Remarks to the Author)

Castilla et al. study the structural basis and functional consequences of the interaction between the voltage-gated calcium channel β -subunit CaV β 2 and actin. An integrated structural biology approach combining cross-linking mass spectrometry and restrained protein-protein docking is used to build a molecular model of the CaV β 2-actin complex. The model suggests that CaV β 2 binds to two adjacent actin monomers via a charged interaction surface. Furthermore, this interface does not overlap with the interaction site of CaV β 2 for the CaV α 1 subunit. Consistent with the model, alanine mutations of six or eight residues in CaV β 2, which are predicted to be energetically most important for binding, reduced the interaction ability of CaV β 2 with actin in cosedimentation assays. The results by Castilla et al. are noteworthy because, in the absence of a crystal structure of the CaV β 2-actin complex, they provide new insights into the structural basis of this interaction and its role in the regulation of CaV channel function.

To study the functional relevance of the CaV β 2-actin interaction, electrophysiological recordings of CaV1.2 in HEK293 cells, co-expressing a CaV β 2 mutant with a reduced actin-association ability, were conducted. In the absence of an intact CaV β 2-actin interaction, a lower current amplitude and a reduced number of functionally available channels were observed. This result suggests that the CaV β 2-actin interaction plays a role in the clearance of non-functional CaV channels from the plasma membrane. The authors propose a working model in which the CaV β 2-actin interaction has a quality control function to monitor the state of CaV channels at the cell membrane.

The results and proposed model are very interesting and the used methods are sound. However, I have some recommendations for the molecular modeling experiments:

1) Can the authors justify the choice of the upper distance bound of 30 Å used in modeling? This distance is larger than the lengths of extended DSSO or DSBU cross-linkers, respectively, when connected to two extended lysine side-chains.

DSSO: $10.3\text{Å} + 2 \times 6.4\text{Å} (\text{Lysine}) = 23.1\text{Å}$

DSBU: $12.5\text{Å} + 2 \times 6.4\text{Å} (\text{Lysine}) = 25.3\text{Å}$

2) For modeling the interaction of CaV β 4 with actin, the PDB file of rat CaV β 4 (PDB 1VYV) (line 474) was used. However, the in vitro experiments (cosedimentation and pulldown assays) were performed with human CaV β 4. Can the authors clarify if they indeed used the structure of rat CaV β 4 or if they constructed e.g. a homology model of human CaV β 4 before docking?

3) The RMSD cutoff (7.5 Å) used for clustering the HADDOCK-generated models of CaV β 2-actin is quite large. Consequently, 190 of 200 models were in the first cluster. How different were the models in this cluster? What was the average pairwise RMSD? And how different were the 4 models used for the subsequent in silico alanine mutational scan? It would be helpful to see the structural variability of the docking models to get an idea of the convergence of the docking calculations.

4) What kind of interactions are formed by the identified 8 hot spot residues in CaV β 2 with actin? Can you provide a zoom-in view on the atomic model and show the intermolecular contacts for those hot spot residues with actin?

5) The HADDOCK docking scores of the CaV β 4-actin complex models are all positive (SI Figure 1). Why? Were there clashes in the models?

6) Did the authors try AlphaFold-Multimer for modeling the CaV β 2-actin interaction?

7) Line 116: The XL-MS distances should be referred to as 'restraints' not 'constraints'.

Other questions:

8) The CaV β 2-actin interaction is sensitive to increased salt concentrations (Figure 2E). Even at medium NaCl concentrations (120mM) the interaction is unstable. Can the authors explain if there is evidence how stable this interaction is inside the cell?

9) Are there non-synonymous SNPs in the genes for CaV β 2 or actin that could affect the proposed structural interaction? Or in other words are there SNPs in the CaV β 2-actin complex that are associated with calcium channelopathies?

Reviewer #4

(Remarks to the Author)

The research focuses on understanding the molecular interactions between L-type CaV1.2 calcium channels and the actin cytoskeleton, highlighting the importance of these interactions in cellular quality control mechanisms that prevent the accumulation of damaged channels. Employing a combination of cross-linking mass spectrometry (XL-MS) and computational biology, the study identifies specific hotspot residues within the CaV β subunit that are crucial for its interaction with actin, impacting calcium channel functionality and signalling without affecting its interaction with the α 1 interaction domain of CaV1.2. The technical validity of the study is supported by a thorough experimental approach, including XL-MS, computational modelling, electrophysiological recordings, and mutagenesis, showcasing the critical role of the CaV β -actin interaction in maintaining functional calcium channel pools.

The findings bear significant implications for the understanding of ion channel regulation and the development of novel therapeutic strategies for neurodegenerative diseases linked to calcium signalling dysregulation. The paper suggests that disrupting the CaV β -actin interaction could serve as a potential therapeutic target for calcium channelopathies. While the data and methodology employed are robust, offering a detailed molecular insight into the CaV β -actin interaction, suggestions for improvement include a more detailed explanation of computational methods to enhance reproducibility and analytical depth:

Additionally, a detailed review of the crosslinking data revealed some major weaknesses of the study.

Remarks:

Major:

- Crosslinker concentration titration is missing to determine optimal parameters for DSSO and DSBU respectively.
- Merox and MetaMorpheus XL data have been searched without contaminant proteins. The FDR control of single proteins is most likely not reliable and hence contaminant proteins should be included in the fasta file
- DSBU datasets show very low scores of identified crosslink matches in the MaxLynx and MeroX result files, Specifically the MaxLynx results of DSBU show only 2 unique interlinks between Actin and CaV β ,
- The numbers of unique links were boosted by searching with KSTY
- DSSO dataset: zhmr file is missing for sample 14, Merox shows only quality data for sample 12 with 5 high scoring inter links, MaxLynx reports only 11 unique inter links with relatively low scores, MetaMorpheus reports 43 unique inter links.
- Interestingly, MaxLynx and MetaMorpheus are searched with K-K and MeroX with KSTY, why is this the case? The search should be performed similarly for all search engines
- A Venn diagram would be valuable to show the overlap between the search results of all search engines (for DSSO and DSBU separately)
- Additionally, the overlap of DSSO and DSBU should also be shown as a Venn diagram to pinpoint the reader to the final crosslinks used for modelling and structure refinement.
- The description of the size exclusion experiment is missing

Minor:

- Non-crosslink control missing in Fig. 1 panels C and D
- 60 min crosslink reaction time is quite long, is the CaV β -actin complex formation/ stability influenced by such a long time at RT? Please show data to validate that the complex is stable for such a long time at RT
- The level of FDR estimation is missing, three different software tools were used to analyze crosslinking data, but each software tool reports FDR on a different crosslink level, e.g MeroX reports only CSM level FDR
- If possible, config files of the searches should also be uploaded as well, these files contain all details and settings of the search and FDR estimation.
- The representation of crosslinks in Fig. 1 E bottom is not ideal, maybe choose the line representation to see from which residue to which residue the crosslink goes
- It would be also interesting to see the results in combination with AlphaLink (<https://github.com/Rappsilber-Laboratory/AlphaLink2>)

Overall, the study could benefit from a more comprehensive explanation of computational modelling techniques, statistical analyses and improvement of the crosslinking data/ presenting the data, to make the findings more accessible to a wider audience. Finally, this research provides valuable insights into the molecular details of calcium channel quality control mechanisms and their implications for cell physiology and pathology, setting a foundation for future studies in this field.

Reviewer #5

(Remarks to the Author)

This paper outlines contacts between the beta subunit of the calcium channel and F-actin in the cell, proposing a role in this interaction in regulation of damaged channel proteins in the cell. This work represents an advance in our understanding of calcium currents generated during various cellular processes, and the beta subunit/F-actin models built will provide many testable hypotheses dealing with disease states arising from defects in calcium signaling pathways. The data presented are clear and supportive of the conclusions, however, some clarifications in the text are needed.

I have some questions about the docking calculations used to generate the model of CaV beta bound to F-actin. Some more discussion of why certain clusters were discarded would be helpful. As a control, was the docking ever run without XL constraints and did the same pose come up near the top of the list? If so, that would strengthen the use of the docking program in the structure determination. The XL constraints also spanned two actin monomers along the long pitch helix which assumes the CaV beta fragments bind to the shallow aspect of actin, which is fine since a number of proteins bind to actin in this spot, but was the docking ever run assuming binding to a single actin monomer gave rise to the XL data? If the assumption about two actin binding is correct, one would expect poor docking poses in terms of scores, buried surface, etc when assuming the XL comes from only one actin, again strengthening the approach taken here.

The iRMSD values need to be better defined to understand what they mean. I am assuming here the iRMSD compares the highest scoring docking pose calculated using the residues predicted to form the PPI to all the other docking poses. The value only really shows how effective the clustering was. A more useful analysis is the figures in Supplemental Figure 1B and D, where we can see how well the docking matches the XL results. Comparison of this result (fit of the XL data) among the other clusters would be useful to strengthen the dismissal of those clusters.

The actin numbers in the text/figures all appear to be off by 2. Residue 330 in PDB 5OOE is not Lys; it's Ile. I think it should be Lys328. Similarly, Ser62 should be Ser60, etc.

On page 6, the RMSD comparing CaV beta 2 and CaV beta 4 dockings of 3.1 Å suggests significant differences between the poses, not similarity as stated. More information on how this number was generated is needed. For instance, what is the RMSD of the two CaV models to each other when aligned? If it's high, that would also be reflected in a high RMSD for the docked poses. Additionally, the calculation of the RMSD should be performed with only the PPI residues' backbones, to show conservation of the actin binding site. What is the sequence identity of the predicted PPIs?

In Figure 1D, the dotted box appears to be in the standard lane, not the sample lane. It's correct in Figure 1C.

PDB 5OOE was used for the actin model in docking; this is an ATP analog bound to F-actin. Why was this structure used? That would be the form of actin on the barbed end of "young" actin, shouldn't the CaV beta bind "aged" ADP actin? Not only that, but there are better resolution structures out there. However, this is unlikely to affect the results much, I'm just curious.

Version 1:

Reviewer comments:

Reviewer #1

(Remarks to the Author)

1. It now seems clear to me there is no $\alpha 2\delta$ cDNA included in the co-expression and functional studies, which means that the folding and function of the channel complex will be abnormal. The EMC complex in the endoplasmic reticulum interacts with both the CaV1.2 and the β subunit and this is displaced by $\alpha 2\delta$ [1]. Therefore without this key component of the calcium channel complex, the findings are unreliable.
2. Abstract "... uncovered a role in replenishing damaged CaV1.2 channels". The authors provide no evidence that there are such damaged channels in the membrane, or what they mean by a damaged channel.
3. The mutant $\beta 2$ with the 8 Alanine mutations clearly is poorly functional with respect to the function of the channels, the authors have not shown directly that this is due to the effect on actin interaction.
4. The reason for the inclusion of the R-X $\beta 4$ mutant is still unclear to me, it apparently interacts less well with actin, but still produces normal calcium currents, which does not support their hypothesis.

1 Chen, Z., Mondal, A., Abderemane-Ali, F., Jang, S., Niranjana, S., Montano, J. L., Zaro, B. W., Minor, D. L., Jr. 2023 EMC chaperone-Ca(V) structure reveals an ion channel assembly intermediate. Nature. (10.1038/s41586-023-06175-5)

Reviewer #3

(Remarks to the Author)

I thank the authors for their responses which have satisfactorily answered all my questions. I find the extra information provided in their responses very helpful. The authors have clearly improved the quality of their manuscript and I think it is ready for publication now.

Reviewer #4

(Remarks to the Author)

Reviewer comments, revised manuscript:

My comments have been fully addressed, resulting in new figures in the main text and supplement as well as new data uploads and updated text paragraphs. Figure 1 really helps the reader to follow the experimental steps and design. In general, changes regarding the crosslink section improved the understanding of the data analysis and selection of crosslinks used for modeling/ docking a lot. Thank you for your effort. I have just a few minor comments about the crosslink experiments: The SDS-Pages of crosslinker optimization suggest, that the crosslinker concentration with 60 min and 100x molar excess worked the best. One would be maybe afraid that over crosslinking has happened, but with over crosslinking you would probably see background grease or something like this on the gel as well. Did the authors also measure the 100x samples?

The difference between the DSSO and DSBU data, does probably not come from the spacer length difference because both spacer length are longer than structurally needed and the SDS gel from the DSBU experiment looks even better as for the DSSO experiment. I rather think measuring the DSBU samples on a 50 cm column with 30min gradient was a not ideal combination. Even with low input amount, crosslink samples benefit from rather longer gradients than shorter ones. Using different instruments adds variability on top of that. Why did the authors not choose one instrument for both crosslinker experiments? The Eclipse would be the better instrument for crosslink samples in this case. The quality of the DSBU samples is not very good and it also does not add new information to your dataset, why did the authors include this data (just out of curiosity)? Selecting crosslinks that have been identified in at least 60% of the replica and with two search engines would be enough to argue your selection.

FDR is very critical in crosslinking experiments, although I know that not all search engines report on residue pair level or PPI level, it is essential to choose a search engine that can calculate on those levels and not only on CSM level. The authors have used the crosslinking data additionally to their biological findings and have reasoned their selection of crosslinks, even with the low quality of the data, the usage of crosslinking data as support data is valid for this study. In any case I would recommend using software tools that can control FDR for crosslinking data on higher levels than CSM level in the future.

Overall, the quality of this publication on the crosslinking part has improved significantly and crosslinking data details are now reported in a transparent manner.

Reviewer #5

(Remarks to the Author)

The response of the authors and the revised manuscript satisfy all of my comments. I have no further criticisms.

Version 2:

Reviewer comments:

Reviewer #1

(Remarks to the Author)

1) I thank the authors for testing the hypothesis whether $\alpha 2\delta$ is involved in the proposed β interaction with actin, with respect to effects on functional effects. The authors should explicitly state at the beginning of the Results for the functional studies that $\alpha 2\delta$ was not included (line 334).

2) Line 475-479: In this new paragraph in the Discussion, "Data not shown" is not acceptable as a statement. These new data should be shown as a supplemental figure with the description in the Results section.

"A proposed quality control during CaV1.2 channel biogenesis is the interaction of the CaV1.2/CaV β core complex with the endoplasmic reticulum membrane protein complex and CaV $\alpha 2\delta 1$ accessory subunit107. In our hands, coexpression of CaV $\alpha 2\delta$ along with the CaV1.2/CaV $\beta 2$ channel core complex marginally altered the biophysical properties of the channel complex and did not prevent the effect of the actin-association-deficient CaV $\beta 2$ mutant (data not shown).

3) The remainder of my concerns (points 2-4) have either been addressed or are matters of opinion and the changes to the ms are appropriate.

Reviewer #1 (Remarks to the Author):

This ms details actin binding to Cav β 2 and β 4. In the absence of a high resolution structure of the two proteins, the ms endeavors to probe how the two proteins cav β and actin interact. The major part of the results details the protein-protein interaction interface between Cav β and actin, using solid biochemical and biophysical methods, cross-linking, site directed mutagenesis, modelling etc.

Specific comments

COMMENT 1: *From the biochemical and modelling results, a mutant β 2 subunit is designed which does not interact with actin. Functional studies are then performed, using control β 2 and the mutant. Fig 5 shows a reduction in ionic current due to the mutant β 2, but no change in gating current. Fig 6 shows a reduction in number of functional channels in the membrane, no change in single channel conductance, and a non-significant reduction in P_{omax} (Fig 6B-C). However, the mean P_{omax} and the distribution of the P_{omax} appears very different, even if not statistically significantly reduced. In Fig 6B, the P_{omax} does not appear to be normally distributed, from the relatively small number of values.*

RESPONSE 1: To address this concern, we have now increased the number of cells as sample sizes to $n=18$ and $n=17$ for cells expressing Cav1.2 channel with either the wild-type or with the actin association-deficient Cav β ₂ mutant bearing eight hotspot mutations (8Ala), respectively. The new data set (Figure 7B) is normally distributed according to Shapiro-Wilk test for the number of functional channels (N), the single-channel conductance (γ) and the maximal open probability (P_{omax}) for both samples, Cav1.2/WT Cav β ₂ and Cav1.2/Mutant Cav β ₂ ($0.9 \leq p\text{-values} \geq 0.07$). As only exception, numbers of channels (N) of Cav1.2/Mutant Cav β ₂ are not normally distributed, with $p\text{-value} = 0.0014$.

To further evaluate the statistical significance also of the not normally distributed data, we applied bootstrapping to estimate sample distributions with no distributional assumption (Efron and Tibshirani, 1993). Bootstrapping simulates replica experiments by repeated resampling the empirical (original) data with replacement. The new data-driven simulated samples – referred to as bootstrap samples – were used to generate the distribution plots, for N , γ and P_{omax} . No statistical difference was found for the original as well as for the bootstrap sample data using t-test statistics for the mean values of γ , ($p \geq 0.4$) and P_{omax} ($p \geq 0.66$) between Cav1.2 channel co-expressed with the wild-type and with the actin association-deficient Cav β ₂ mutant. However, N significantly differed between the two groups ($p \leq 0.001$).

We are thus confident about the statistical reliability of our results that indicate neither statistically relevant alterations in the conductance nor in the maximal open probability for Cav1.2 channel coexpressed with the actin association-deficient Cav β ₂, but a 53.8% decrease in the number of functionally active channels mediated by the actin-association deficient Cav β ₂, which fully accounts for the 53.0% reduction in the average macroscopic peak current.

We modified the plots to accommodate the new experimental data, and for better comparison, we explicitly added the mean values and the p -values in text and all plots (Figures 6-8).

COMMENT 2: *The point of using the β 4 R-X mutant is unclear to me. It is found to have a higher affinity for actin. In the original paper describing this mutation (Escayg et al, 2000), the ionic currents were found*

to be slightly larger for the mutant, but this result is not tested here. Indeed, no functional data are included for this mutant.

RESPONSE 2: We now clarify the reason for using the β_4 R-X mutant ($\text{Ca}_v\beta_4$ R482X) originally described by Escayg et al, 2000 (Escayg et al., 2000). We have added new results on the effect of this mutant on the $\text{Ca}_v2.2$ -mediated ionic currents, and a full data set comprising whole-cell recordings, gating current measurements and noise analysis on the impact of $\text{Ca}_v\beta_4$ R482X and $\text{Ca}_v\beta_4$ R482X 8Ala bearing the hotspot mutations that impair binding to actin on $\text{Ca}_v1.2$ channels. Also laser confocal imaging of cells and Western blot analysis of cells expressing the $\text{Ca}_v\beta_4$ R482X constructs were added.

For clarity, we divided our answers to the reviewer's comments in the following two sections

2.1. Point of using the β_4 R-X mutant.

The use of another $\text{Ca}_v\beta$ subtype, besides type 2 ($\text{Ca}_v\beta_2$), offered the opportunity to test the robustness of our integrative strategy applied to define the interaction surface between $\text{Ca}_v\beta$ and actin. We have previously shown that $\text{Ca}_v\beta_4$ also binds to actin (Guzman et al., 2019), and the alignment between the amino acid sequences of $\text{Ca}_v\beta_2$ and $\text{Ca}_v\beta_4$ showed that all four lysine residues in $\text{Ca}_v\beta_2$ involved in the inter-molecular cross-links used for the modelling of $\text{Ca}_v\beta_2$ /actin interaction are conserved in $\text{Ca}_v\beta_4$ (alignment added to Supplementary Figure 12).

We used $\text{Ca}_v\beta_2$ analogous XL-derived distance restraints to perform the HADDOCK docking with $\text{Ca}_v\beta_4$ and actin. To validate the resulting model, *in vitro* binding assays were performed using recombinant wild-type and hotspot $\text{Ca}_v\beta_4$ mutant proteins. Recombinant full-length $\text{Ca}_v\beta_4$ is not stable enough under the lower salt conditions used for actin polymerization (Guzman et al., 2019), however, the natural occurring $\text{Ca}_v\beta_4$ R482X lacking the 39 C-terminal amino acids turned out to be amenable to recombinant expression and purification, and appeared to have a higher *in vitro* affinity for actin compared with the $\text{Ca}_v\beta_2$ and $\text{Ca}_v\beta_4$ core regions. Therefore, $\text{Ca}_v\beta_4$ R482X mutant appeared as an excellent candidate for studying possible differences in the affinity caused by the hotspot mutations. At the same time, the identification of the actin interaction surface on $\text{Ca}_v\beta_4$ (experimentally validated in $\text{Ca}_v\beta_4$ R482X) might provide insights on factors contributing to the unresolved etiology of the neurological disorder associated with R482X genetic mutation (Etemad et al., 2014) (please see also the answer to point 2.2 below).

We have modified the manuscript with an improved description of the reasons for using $\text{Ca}_v\beta_4$ R482X truncated mutant (page 8, line 265; page 9, line 297).

2.2. In the original paper describing this mutation (Escayg et al, 2000), the ionic currents were found to be slightly larger for the mutant, but this result is not tested here. Indeed, no functional data are included for this mutant.

We now included new functional data on the effect of the truncated mutant $\text{Ca}_v\beta_4$ R482X and of its actin-association deficient mutant ($\text{Ca}_v\beta_4$ R482X 8Ala) on neuronal $\text{Ca}_v2.2$ channels and also on $\text{Ca}_v1.2$, which we mainly used for the electrophysiological analysis in this study for its amenability to gating currents recordings. We engineered three new constructs for expression in mammalian HEK293 cells; $\text{Ca}_v\beta_4$, $\text{Ca}_v\beta_4$ R482X, and $\text{Ca}_v\beta_4$ R482X 8Ala all fused to mRFP.

We found neither differences in the current-to-voltage ($p = 0.9$ for the peak current) nor in the voltage dependent of activation plots ($p = 0.4$ for $V_{0.5}$) between $\text{Ca}_v2.2$ channels associated with WT $\text{Ca}_v\beta_4$ or $\text{Ca}_v\beta_4$

R482X deletion mutant. Please note that the slight effect on ionic currents reported by Escayg et al (Escayg et al., 2000) was observed in *Xenopus laevis* oocytes heterologously expressing Ca_vβ₄ R482X with Ca_v2.1. Also, Ca_vβ₄ R482X 8Ala actin-association deficient mutant yielded Ca_v2.2-mediated currents statistically undistinguishable from WT Ca_vβ₄ ($p = 0.63$ for the peak current and $p = 0.18$ for $V_{0.5}$). The new data has been added to the Supplementary Figure 16.

Ca_v1.2 co-expressed with either Ca_vβ₄ R482X or Ca_vβ₄ R482X 8Ala actin-association deficient mutant showed no significant differences in whole-cell current and gating current amplitudes ($p = 0.8$ and $p = 0.6$ for the peak current and the charge movement, respectively). Correspondingly, noise analysis revealed statistically comparable values between Ca_v1.2/WT Ca_vβ₄ and Ca_v1.2/Ca_vβ₄ R482X 8Ala for the number of functional channels (N , $p = 0.45$), the single-channel conductance (γ , $p = 0.18$) and the maximal open probability (P_{max} , $p = 0.47$). Furthermore, Ca_vβ₄ R482X, which displays normal nuclear targeting as compared to WT (Etemad et al., 2014), and Ca_vβ₄ R482X 8Ala showed comparable nuclear localization and levels of expression, as assessed by fluorescence confocal imaging and Western blot, respectively. This new set of data is shown in Figure 8 and described in a new section (page 12, line 385).

Our finding that Ca_vβ₄ R482X 8Ala actin-association deficient mutant has no effect Ca_v-mediated ionic currents expressed is anticipated from the lack of effect on Ca_vβ₄ R482X on Ca_v function observed here in mammalian cells and the fairly small effect originally described in *X. laevis* oocytes. The latter is not sufficient to explain the pathophysiology behind the diseased condition and led to the suggestion that dysregulation of Ca_vβ₄ non-channel related functions (i.e, separately from its modulatory effect on Ca_v channel activity) underlies the neurological dysfunction (Escayg et al., 2000; Etemad et al., 2014). The present observation that Ca_vβ₄ R482X displays a relatively high apparent affinity for actin together with previous results showing that Ca_vβ₄ increases synaptic strength in hippocampal neurons independent of its association state to Ca_v, but dependent on an intact actin cytoskeleton (Guzman et al., 2019), insinuates that an aberrant Ca_vβ₄ R482X / F-actin interaction might play a role in the Ca_vβ₄ R482X-associated neurological disorder. This hypothesis is now testable thanks to the actin-association deficient Ca_vβ₄ R482X mutant designed in the present work. We now explicitly discuss this outcome in the Discussion section of the revised manuscript (page 14, lines 452-471).

COMMENT 3: *The model in Fig 7 shows actin being involved in quality control of “defective” calcium channels, but this model is not supported by the data presented. There is no information in the ms about what would constitute an endogenous defective channel, or on the endocytosis or degradation route of the “defective” channels. The mutant channels appear to be in the cell membrane to the same extent, from the gating charge measurements, and are not obviously internalized from the immunofluorescence results in Fig 5A, although the proportion of cells expressing the β2 at the membrane appears less from Fig 5A.*

RESPONSE 3: First, we would like to apologize for any unclearness in the text concerning the immunofluorescence results in Figure 5A (now Figure 6A) that led to the misinterpretation that the signal originated from mutant channels. The fluorescence confocal microscopy images show HEK293 cells transfected only with Ca_vβ type 2 subunit (Ca_vβ₂) fused to mRFP. This Ca_vβ₂ is targeted to the plasma membrane even in the absence of the pore-forming Ca_vα₁ subunit due to the presence of a specific palmitoylation site (Miranda-Laferte et al., 2012; Qin et al., 1998). The images show that the hotspot mutations introduced in Ca_vβ₂, which decreased its affinity for actin, do not alter its degree of colocalization with the plasma membrane, as quantified by Mander’s overlap coefficient. We added a scheme of Ca_vβ₂ to emphasize that images show only Ca_vβ₂ fluorescence signal.

The Reviewer is right that we have not sufficiently contextualized the model proposed in Figure 7 (now Figure 9). The model is based on the present and earlier work that demonstrates that the density of voltage-gated calcium channels at the plasma membrane is regulated by dynamic exo- and endocytosis. This highly dynamic cell surface $\text{Ca}_v1.2$ turnover occurs on a relatively short timescale with an internalization rate of a few minutes and depends on an intact actin cytoskeleton. Internalized $\text{Ca}_v1.2$ channels are effectively replaced by newly inserted $\text{Ca}_v1.2$ protein originating from the recycling rather than from the exocytic biosynthetic pathway. This mechanism is able to maintain a steady state level of ionic currents, despite of the constant loss of channels due to endocytosis, for at least 20 hr after inhibition of biosynthetic delivery (Conrad et al., 2018). On the other hand, we have recently demonstrated that $\text{Ca}_v\beta$ regulates the residency time of $\text{Ca}_v1.2$ at the cell surface and thus, the endocytic turnover of the channel (Conrad et al., 2021).

We have now applied a multi-approach strategy to identify the $\text{Ca}_v\beta$ -actin interaction surface and study its role in channel turnover. The main discovery of the present study is that impairing the association of $\text{Ca}_v\beta$ with actin results in reduced ionic currents at identical channel numbers and single-channel properties, demonstrating that under these conditions the percentage of non-conducting channels is increased. This implies that a competent $\text{Ca}_v\beta$ -actin interaction is necessary for removal of the conducting-defective channels, which normally are dynamically removed by endocytosis and replaced by healthy ones in order to maintain a steady state level of ionic currents. We do not claim that this interaction specifically monitors conduction-defective channels but from all potential conformational damages on the channel protein, we can only certainly ascertain those altering channel permeation when using electrophysiological techniques. This is now discussed in Discussion section (page 13, line 415; pages 13-14, lines 432-451).

Even though we do not directly address the potential source of “an endogenous defective channel” in the present work, this issue has been extensively discussed elsewhere (Conrad et al., 2018). We now address explicitly the issue of what constitutes an endogenous defective channels in the discussion (page 14, line 443); different cellular stimuli can alter the molecular environment of the channel leading to the continuous generation of conformationally damaged integral membrane proteins. Conformationally impaired proteins are normally recognized and modified at their exposed cytosolic domains to signal their endocytic removal (Apaja and Lukacs, 2014; Apaja et al., 2010; Babst, 2014). (Please also refer to answer to comment 5).

In response to the Reviewer’s criticism, and to avoid overinterpretation, we have simplified the cartoon of the model to merely highlight the main conclusion of the present study; i.e. that a competent $\text{Ca}_v\beta$ -actin interaction is required for clearing corrupted $\text{Ca}_v1.2$ channels from the plasma membrane preventing their accumulation (Figure 9). The original model presented was based on the assumption that defective ion channels are removed and post-endocytic trafficked using the same endocytic and sorting machineries as healthy $\text{Ca}_v1.2$ channels, which has been previously studied (Conrad et al., 2018).

Taken altogether, the accumulation of conduction-defective channels mediated by an impaired $\text{Ca}_v\beta$ -actin association is well explained by assuming that those channels become reluctant to endocytic removal escaping the well-established actin-dependent high turnover rate of $\text{Ca}_v1.2$ channels. Noteworthy, impaired $\text{Ca}_v\beta$ -actin association still allows the insertion of normal $\text{Ca}_v1.2$ channels at the plasma membrane.

COMMENT 4: End of Results “*these findings suggest that the association between $Ca_v\beta$ and actin regulates a process that monitors, removes, and replaces conduction-defective $Ca_v1.2$ proteins, thereby preventing overpopulation of damaged channels at the cell surface*”. This is very speculative, and the speculation about actin’s role in removal of defective channel proteins continues in the Discussion.

RESPONSE 4: We have removed this sentence from the Result section and simplified the scheme of the proposed model to summarize the main finding of the involvement of $Ca_v\beta_2$ -actin association in $Ca_v1.2$ clearance at the cell surface (Figure 9). We have also improved the description of arguments that led us to our conclusion in the Discussion section (page 13, line 432). We feel that speculations or educated guesses are important to open new hypotheses in relevant and especially, unclear topics such as how cells deal with non-functional proteins at the plasma membrane to safeguard the cell integrity and proteostasis.

We think that our data unambiguously show that impairment of $Ca_v\beta_2$ -actin association via introducing hotspot mutations in the interaction surface of $Ca_v\beta_2$ with actin lead to an accumulation of conduction-defective channels. This $Ca_v\beta_2$ mutant displayed no other functional alteration; its cellular localization, expression levels, association capability with the $Ca_v\alpha_1$ pore-forming interaction domain and structural integrity were preserved. Coexpression with $Ca_v1.2$ resulted in ionic currents, gating and permeation properties that were virtually undistinguishable from coexpression with the wild-type $Ca_v\beta_2$ but leads to a reduction in the macroscopic ionic current with a concomitant decrease in the amount of functionally active channels assembled at the plasma membrane. This necessarily implies the occurrence of an increased fraction of conduction-defective channels at the plasma membrane, i.e. channels that contribute to gating currents (intact voltage sensor) but not to the macroscopic ionic current.

Given the well-established role of actin in the endocytosis of ion channels, and in particular, the strict requirement of intact actin cytoskeleton for the turnover and normal post-endocytic trafficking of $Ca_v1.2$ channels (Chakrabarti et al., 2021; Conrad et al., 2018; Jin et al., 2022; Mooren et al., 2012; Wu and Chan, 2022), a natural conclusion is that the actin association-deficient $Ca_v\beta_2$ mutant exert its effect through the concerted physiological action of beta with actin. For this reason, we do feel safe in postulating a role of actin via its association with $Ca_v\beta_2$, in the clearance of defective $Ca_v1.2$ channels at the plasma membrane.

COMMENT 5: *There is no attempt to use a conduction-defective calcium channel mutant which would be a way to examine these claims.*

RESPONSE 5: This is a very interesting proposal. At first glance, one could expect that a conduction-defective calcium channel mutant will stay longer at the plasma membrane when $Ca_v\beta$ -actin interaction is impaired. However, we are not attempting to state that the loss of ion permeability is an attribute that specifically triggers ion channel clearance at the plasma membrane in normal conditions. We now present a better explanation of our reasoning: we argue that local unfolding, including alterations in regions affecting ion permeation, is the signal that prompts the removal of misfolded channels by endocytosis (page 14, line 443). A conduction-defective channel brought about by a point mutation is not necessarily equivalent to a non-conducting channel generated by local unfolding that ultimately exposes signals (for example ubiquitination/degradation) prompting clearance mechanisms. A mutation in $Ca_v1.2$ subunit can also modify other aspects of channel function, for example forward and backward trafficking routes, that are not readily experimentally assessable. (Please also refer to answer to comment 3).

COMMENT 6: *The title is similarly speculative and does not mention the main topic of the paper which is mapping the actin-Ca_vβ interaction interface.*

RESPONSE 6: We have replaced the title by:

Mapping the interaction surface between Ca_vβ and actin and its role in calcium channel clearance

References for Reviewer 1

- Apaja, P.M., and G.L. Lukacs. 2014. Protein homeostasis at the plasma membrane. *Physiology (Bethesda, Md.)*. 29:265-277.
- Apaja, P.M., H. Xu, and G.L. Lukacs. 2010. Quality control for unfolded proteins at the plasma membrane. *The Journal of cell biology*. 191:553-570.
- Babst, M. 2014. Quality control: quality control at the plasma membrane: one mechanism does not fit all. *The Journal of cell biology*. 205:11-20.
- Chakrabarti, R., M. Lee, and H.N. Higgs. 2021. Multiple roles for actin in secretory and endocytic pathways. *Current biology : CB*. 31:R603-R618.
- Conrad, R., D. Kortzak, G.A. Guzman, E. Miranda-Laferte, and P. Hidalgo. 2021. Ca(V) β controls the endocytic turnover of Ca(V) 1.2 L-type calcium channel. *Traffic (Copenhagen, Denmark)*. 22:180-193.
- Conrad, R., G. Stölting, J. Hendriks, G. Ruello, D. Kortzak, N. Jordan, T. Gensch, and P. Hidalgo. 2018. Rapid Turnover of the Cardiac L-Type Ca_v1.2 Channel by Endocytic Recycling Regulates Its Cell Surface Availability. *iScience*. 7:1-15.
- Efron, B., and R. J. Tibshirani. 1993. An Introduction to the Bootstrap. Monographs on Statistics and Applied Probability. Chapman & Hall/CRC, Boca Raton, Florida, USA.
- Escayg, A., M. De Waard, D.D. Lee, D. Bichet, P. Wolf, T. Mayer, J. Johnston, R. Baloh, T. Sander, and M.H. Meisler. 2000. Coding and noncoding variation of the human calcium-channel beta4-subunit gene CACNB4 in patients with idiopathic generalized epilepsy and episodic ataxia. *Am.J.Hum.Genet.* 66:1531-1539.
- Etemad, S., M. Campiglio, G.J. Obermair, and B.E. Flucher. 2014. The juvenile myoclonic epilepsy mutant of the calcium channel β_4 subunit displays normal nuclear targeting in nerve and muscle cells. *Channels (Austin, Tex.)*. 8:334-343.
- Guzman, G.A., R.E. Guzman, N. Jordan, and P. Hidalgo. 2019. A Tripartite Interaction Among the Calcium Channel α_1 - and β -Subunits and F-Actin Increases the Readily Releasable Pool of Vesicles and Its Recovery After Depletion. *Frontiers in cellular neuroscience*. 13:125.
- Jin, M., C. Shirazinejad, B. Wang, A. Yan, J. Schöneberg, S. Upadhyayula, K. Xu, and D.G. Drubin. 2022. Branched actin networks are organized for asymmetric force production during clathrin-mediated endocytosis in mammalian cells. *Nature communications*. 13:3578.
- Miranda-Laferte, E., S. Schmidt, A.C. Jara, A. Neely, and P. Hidalgo. 2012. A short polybasic segment between the two conserved domains of the beta2a-subunit modulates the rate of inactivation of R-type calcium channel. *The Journal of biological chemistry*. 287:32588-32597.
- Mooren, O.L., B.J. Galletta, and J.A. Cooper. 2012. Roles for actin assembly in endocytosis. *Annual review of biochemistry*. 81:661-686.
- Qin, N., D. Platano, R. Olcese, J.L. Costantin, E. Stefani, and L. Birnbaumer. 1998. Unique regulatory properties of the type 2a Ca²⁺ channel beta subunit caused by palmitoylation. *Proc.Natl.Acad.Sci.USA*. 95:4690-4695.
- Wu, L.G., and C.Y. Chan. 2022. Multiple Roles of Actin in Exo- and Endocytosis. *Frontiers in synaptic neuroscience*. 14:841704.

Reviewer #2 (Remarks to the Author):

PPIs are involved in almost every biological process and, since their aberrancy has been implicated in several human diseases and pharmacological targets in drug discovery. Therefore, determining the contact surfaces for PPIs advances our understanding of the molecular mechanisms underlying cell function and dysfunction. Authors identified the PPI interface between CaV β and actin using XL-MS and protein protein docking experiments.

I was asked by the editor to specifically review the protein cross-linking analysis and therefore I would like to give a recommendation only for this part of the manuscript.

From a global perspective, I don't find the XL analysis very transparent and see a huge need for improvement in this respect.

Thank you for the comment. We had initially attempted to keep the description concise, but agree that this should be extended. We have now incorporated the following additional information:

- We now show the full images of the SDS-PAGE for the optimization of the XL reaction experiments by varying the concentration of the cross-linkers and reaction time (Supplementary Figure 1).
- We now document all identified CSMs (total, inter, intra, looplink and deadend) in a new Supplementary Table 1.
- We have replaced the former Supplementary Table 1 by Supplementary Table 2, which provides an overview of the 22 identified unique intermolecular cross-linked peptides (inter-XLs) using DSSO and DSBU as cross-linkers. The table also includes information on the unique inter-XLs selected for the DisVis analysis and identified as self-consistent.
- We have uploaded all relevant files to PRIDE, including a table summarizing the search parameters used in the three XL software programs (MaxLynx, Merox and Metamorpheus).
- We have added three Venn diagrams in the Supplementary Figure 3:
 - i) Total intermolecular XLs identified in five replicates by Metamorpheus, Merox and MaxLynx with DSSO as cross-linker (Supplementary Figure 3A)
 - ii) Total intermolecular XLs identified in four replicates by Metamorpheus, Merox and MaxLynx with DSBU as cross-linker (Supplementary Figure 3B)
 - iii) Unique inter-XLs obtained by DSSO and DSBU (Supplementary Figure 3C)
- We have included a schema depicting the workflow for the XL-MS data and the protein-protein docking (Figure 1).

COMMENT 1 (Major issues): *Nobody can determine an FDR with only 2 proteins. Therefore, all results are not trustworthy. Only one score is simply not enough to determine the quality of the MS/MS results. Above all, it is unclear why a certain score cutoff was chosen.*

The results of DSBU and DSSO are of totally different quality. The results of the individual analyses were not listed and presented transparently in the manuscript.

RESPONSE 1: For clarity, the reviewer's comments with the corresponding answers have been divided in three.

1.1. Nobody can determine an FDR with only 2 proteins. Therefore, all results are not trustworthy. Only one score is simply not enough to determine the quality of the MS/MS results. Above all, it is unclear why a certain score cutoff was chosen

We agree that it is not possible to calculate an FDR for two proteins. However, in contrast to large-scale studies, we do not attempt to establish the presence of a protein-protein interaction. It is well established that $\text{Ca}_v\beta_2$ and actin interact with each other (Stölting et al., 2015), and we have here used cross-linking mass spectrometry to obtain distance restraints to support the modeling of the interaction and to guide further experimental validation as described in the manuscript.

Because the scores are not readily comparable between the different programs, we have repeated the searches with all three software packages (MaxLynx, Merox and Metamorpheus) considering potentially contaminating proteins and now applied a consistent FDR of 0.01 for cross-linked-peptide-to-spectrum matches (CSMs). We now explicitly clarify this point in the manuscript (page 17, line 562-565).

To increase data consistency and reliability, only cross-linked peptides identified by at least two independent search engines were subjected to filtering with DisVis to detect false positive distance restraints. Further details are provided in Supplementary Table 2.

1.2. The results of DSBU and DSSO are of totally different quality.

We agree that the results vary in quality. This is most likely due to the different instrumentation used for the analysis. DSSO samples were analysed on an Orbitrap Eclipse with a 60 minutes gradient, while DSBU samples could only be analysed with a Q Exactive Plus with a 30 minutes gradient time. Additionally, the two cross-linking reagents differ in length by about 2.2 Å, which may result in different peptide identifications. Considering this experimental variation, we were reassured to observe a fair overlap between the two cross-linkers (see Supplementary Figure 3C and Supplementary Table 2).

We have explicitly reported in the manuscript the LC-MS/MS data acquisition settings used for each cross-linker in page 17, lines 541-546.

1.3. The results of the individual analyses were not listed and presented transparently in the manuscript.

We now provide detailed information on the individual search results in the manuscript, and have added a table (Supplementary Table 1) listing the identified CSMs (classified in total, inter, intra, looplink and dead ends) for each cross-linking reagent. In addition, a new Supplementary Figure 3 shows Venn diagrams for the overlap of interXLs identified by the three search engines (MaxLynx, Merox and Metamorpheus) and with the two cross-linkers (DSBU and DSSO).

In addition, we now describe in more detail how the XL data were filtered to derive the list of 22 unique cross-linked peptide sequences, for which we provide detailed information on their identification in the revised Supplementary Table 2. This table also indicates the unique inter-XLs that were subjected to DisVis analysis and the thus identified self-consistent inter-XLs that were used for generating the structural model of the interaction between $\text{Ca}_v\beta$ and actin.

We now describe the results of the individual analyses for the three software packages (MaxLynx, Merox and Metamorpheus) and the two different cross-linkers (DSSO and DSBU) in the revised manuscript (pages 4-5, lines 129-146).

COMMENT 2 (Minor issues): *One of the first experiments a bachelor's or master's student performs at xl is to determine the correct concentration of cross-linker. This was apparently not done here in a nature publication. This is exactly how the response time was not optimized. If this is too long, there are many artifacts.*

RESPONSE 2: We apologize for omitting the results of the optimization of the XL reaction which gave the impression that we randomly selected the concentration and incubation time for each crosslinker. In fact, extensive work was necessary to establish suitable conditions. Full images of the SDS-PAGE gels of the XL reactions for different reaction times (0, 15, 30 and 60 minutes) and different molar ratio cross-linker excess (2.5x, 5x, 10x, 20x, 50x and 100x) are now shown in the supplementary data (Supplementary Figure 1). After multiple rounds of optimization, a low amount of cross-linker (2.5x or 5x excess, for DSSO and DSBU, respectively) was established to be the concentration that minimized the occurrence of over-crosslinked products.

COMMENT 3 (Minor issues): *Lack of transparency in the upload of data files (results, settings,...)*

RESPONSE 3: We have documented the search settings in detail and uploaded the documents in a new PRIDE submission (PXD05456 and PXD53481).

COMMENT 4 (Minor issues): *The workflow for the analysis was not described exactly (e.g. size exclusion, negative controls SDS page,...) or missing negative controls.*

RESPONSE 4: We agree and have now implemented a workflow describing the individual steps from the preparation of the proteins to the experimental validation, including XL-MS and protein-protein docking (Figure 1). Negative controls and SDS-PAGE for the optimization of the cross-linking reaction are shown in Figure 2C and Supplementary Figure 1) (Please see also Response 2). Note that due to the low yield of cross-linked products, we did not use size-exclusion chromatography upstream of C18 peptide separation for purification/enrichment of XL-linked peptides. Size-exclusion chromatography was performed following Ni²⁺ affinity chromatography to obtain highly-purified recombinant Ca_vβ (page 16, line 513).

COMMENT 5 (Minor issues): *I would therefore recommend contacting a cross-linking expert to validate the results again. Transparent presentation of all findings (even if they are negative) in tabular form is essential.*

RESPONSE 5: We agree that the documentation of the experiment was too short. We have now extensively documented our XL-MS results in tabular form (Supplementary Tables 1 and 2). Together with the answers to the above comments, we hope that we now meet the standards of transparency in the presentation of our results.

COMMENT 6 (Minor issues): *I would therefore recommend assessing this publication with a major revision for the XL part, because it is simply very important for modeling that the XL results are correct.*

RESPONSE 6: We have extensively revised the XL-MS analysis. Please refer to the list of revisions included at the beginning of the response. We feel confident that our strategy combining XL-MS and protein-protein docking generated a reliable structural model of the interaction between Ca_vβ and actin, which was unequivocally validated by experimental work.

References for Reviewer 2

Stölting, G., R.C. de Oliveira, R.E. Guzman, E. Miranda-Laferte, R. Conrad, N. Jordan, S. Schmidt, J. Hendriks, T. Gensch, and P. Hidalgo. 2015. Direct interaction of Ca_vβ with actin up-regulates L-type calcium currents in HL-1 cardiomyocytes. *The Journal of biological chemistry*. 290:4561-4572.

Reviewer #3 (Remarks to the Author):

Castilla et al. study the structural basis and functional consequences of the interaction between the voltage-gated calcium channel β -subunit CaV β 2 and actin. An integrated structural biology approach combining cross-linking mass spectrometry and restrained protein-protein docking is used to build a molecular model of the CaV β 2-actin complex. The model suggests that CaV β 2 binds to two adjacent actin monomers via a charged interaction surface. Furthermore, this interface does not overlap with the interaction site of CaV β 2 for the CaV α 1 subunit. Consistent with the model, alanine mutations of six or eight residues in CaV β 2, which are predicted to be energetically most important for binding, reduced the interaction ability of CaV β 2 with actin in cosedimentation assays. The results by Castilla et al. are noteworthy because, in the absence of a crystal structure of the CaV β 2-actin complex, they provide new insights into the structural basis of this interaction and its role in the regulation of CaV channel function.

To study the functional relevance of the CaV β 2-actin interaction, electrophysiological recordings of CaV1.2 in HEK293 cells, co-expressing a CaV β 2 mutant with a reduced actin-association ability, were conducted. In the absence of an intact CaV β 2-actin interaction, a lower current amplitude and a reduced number of functionally available channels were observed. This result suggests that the CaV β 2-actin interaction plays a role in the clearance of non-functional CaV channels from the plasma membrane. The authors propose a working model in which the CaV β 2-actin interaction has a quality control function to monitor the state of CaV channels at the cell membrane. The results and proposed model are very interesting and the used methods are sound. However, I have some recommendations for the molecular modeling experiments:

COMMENT 1: *Can the authors justify the choice of the upper distance bound of 30 Å used in modeling? This distance is larger than the lengths of extended DSSO or DSBU cross-linkers, respectively, when connected to two extended lysine side-chains.*

DSSO: $10.3\text{Å} + 2 \times 6.4\text{Å} (\text{Lysine}) = 23.1\text{Å}$

DSBU: $12.5\text{Å} + 2 \times 6.4\text{Å} (\text{Lysine}) = 25.3\text{Å}$

RESPONSE 1: We chose an upper distance bound value that took into account the crosslinker length and that of the two extended Lys side chains (as mentioned by the reviewer) plus an additional tolerance distance (based on previous studies). The Rappsilber group proposed to increase the upper limit by up to 1.5 Å for surface lysines to account for potential errors in the side chain position in the crystal structure (Rappsilber, 2011). This would increase the upper bound distances suggested by the reviewer up to 26.1 and 28.3 Å for DSSO and DSBU, respectively. Orban-Nemeth and coworkers suggested an extended Lys side chain length of 5.5 Å together with a tolerance of 13 Å to account for protein conformational dynamics, besides the crosslinker length (Orban-Nemeth et al., 2018). This would result in upper bound distances of 34.3 and 35.5 Å for DSSO and DSBU, respectively. The 30 Å upper bound distance used in our work would then be between the two aforementioned estimates. Moreover, comparison of crosslinking and molecular dynamics simulation data showed that a distance restraint between 26 and 30 Å is appropriate for DSS and BS3, cross-linkers that both have a linker length of 11.4 Å, in between DSSO and DSBU (Merkley et al., 2014). Moreover, upper bound values of 30 Å (or even 35 Å) for DSSO and DSBU have been applied in the literature when analyzing XL-MS datasets with both cross-link and structural information available (Bullock et al., 2018; Cohen and Schneidman-Duhovny, 2023; Iacobucci et al., 2019; Matthew Allen Bullock et al., 2016; Mintseris and Gygi, 2020; Piersimoni et al., 2022; Schiffrin et al., 2020; Wang et al., 2017).

We have now explained our choice of an upper bound distance of 30 Å for the definition of the XL-MS-derived distance restraints (page 18, lines 577-581; page 19, lines 612-613).

COMMENT 2: For modeling the interaction of Ca_vβ₄ with actin, the PDB file of rat Ca_vβ₄ (PDB 1VYV) (line 474) was used. However, the in vitro experiments (cosedimentation and pulldown assays) were performed with human Ca_vβ₄. Can the authors clarify if they indeed used the structure of rat Ca_vβ₄ or if they constructed e.g. a homology model of human Ca_vβ₄ before docking?

RESPONSE 2: We did use the X-ray structure of rat Ca_vβ₄ (PDB 1VYV; Uniprot ID D45055-2) in our protein-protein docking calculations. The sequence identity between the rat Ca_vβ₄ and human Ca_vβ₄ sequences (Uniprot IDs D4A055-2 and O00305-1, respectively) is 98.8% (please see the alignment below, for the reviewer only). When considering the region used for the X-ray study in reference (Chen et al., 2004) (residues 49-407), there is only one conservative change T197/S198 (rat/human Ca_vβ₄). Moreover, the structural superposition of the rat Ca_vβ₄ X-ray structure (PDB 1VYV; Uniprot ID D45055-2) and a homology model of human Ca_vβ₄ (based on the rat Ca_vβ₄ template) yielded a backbone RMSD of 0.54 Å, compared to a backbone RMSD of 0.52 Å between the two chains present in the asymmetric unit of the X-ray structure of rat Ca_vβ₄. Similarly, the AlphaFold model of human Ca_vβ₄ (AF-O00305-F1, version 4) showed a backbone RMSD of 0.86 Å with respect to the X-ray structure of rat Ca_vβ₄. Given this high sequence and structural similarity between the two Ca_vβ₄ orthologs, we do not expect differences in docking models generated with either rat Ca_vβ₄ or human Ca_vβ₄.

We thank the reviewer for raising this point and have added this information in the manuscript (page 19, lines 608-609).

```

sp | D4A055 | CACB4_RAT      -MSSSYAKNGAADGPHSPSSQVARGTTTRRSRLKRSDBGSTTSTSFILRQGSADSYTSRPS 59
sp | O00305 | CACB4_HUMAN  MSSSSYAKNGTADGPHSPTSQVARGTTTRRSRLKRSDBGSTTSTSFILRQGSADSYTSRPS 60
*****:*****:*****

sp | D4A055 | CACB4_RAT      DSDVSL EEDREAIRQEREQQAIIQLERAKSKPVAFVAVKTNVSYCGALDEDVVPVSTAISF 119
sp | O00305 | CACB4_HUMAN  DSDVSL EEDREAIRQEREQQAIIQLERAKSKPVAFVAVKTNVSYCGALDEDVVPVSTAISF 120
*****:*****:*****

sp | D4A055 | CACB4_RAT      DAKDFLHIKEKYNDWWIGRLVKEGCEIGFIPSPRLLENIRIQEQKGRGRFHGGKSSGNS 179
sp | O00305 | CACB4_HUMAN  DAKDFLHIKEKYNDWWIGRLVKEGCEIGFIPSPRLLENIRIQEQKGRGRFHGGKSSGNS 180
*****:*****:*****

sp | D4A055 | CACB4_RAT      SSSLGEMVSGTFRATPTTTAKQKQVTEHIPYDVVPSMRPVVLVGPSPKGYEVTDMMQK 239
sp | O00305 | CACB4_HUMAN  SSSLGEMVSGTFRATPTTAKQKQVTEHIPYDVVPSMRPVVLVGPSPKGYEVTDMMQK 240
*****:*****:*****

sp | D4A055 | CACB4_RAT      ALFDFLKHFRDGRISITRVTDISLAKRSVLNPNPKRAIERSNTRSSLAEVQSEIERIF 299
sp | O00305 | CACB4_HUMAN  ALFDFLKHFRDGRISITRVTDISLAKRSVLNPNPKRAIERSNTRSSLAEVQSEIERIF 300
*****:*****:*****

sp | D4A055 | CACB4_RAT      ELARSLQLVVL DADTINHPAQLIKTSLAPIIVHVKVSSPKVLQRLIKSRGKSQSKHLNVQ 359
sp | O00305 | CACB4_HUMAN  ELARSLQLVVL DADTINHPAQLIKTSLAPIIVHVKVSSPKVLQRLIKSRGKSQSKHLNVQ 360
*****:*****:*****

sp | D4A055 | CACB4_RAT      LVAADKLAQCPEMFVDVILDENQLEDACEHLGEYLEAYWRATHTSSTPMTPLLRNVLGS 419
sp | O00305 | CACB4_HUMAN  LVAADKLAQCPEMFVDVILDENQLEDACEHLGEYLEAYWRATHTSSTPMTPLLRNLGS 420
*****:*****:*****

sp | D4A055 | CACB4_RAT      TALSPYPTAISGLQSRMRHSNHSTENSP IERRSLMTSDENYHNERARKSRNRLSSSSQH 479
sp | O00305 | CACB4_HUMAN  TALSPYPTAISGLQSRMRHSNHSTENSP IERRSLMTSDENYHNERARKSRNRLSSSSQH 480
*****:*****:*****

sp | D4A055 | CACB4_RAT      SRDHYPLVEEDY PDSYQD TYKPHNRN RSGPGGCSHDSRHRL 519
sp | O00305 | CACB4_HUMAN  SRDHYPLVEEDY PDSYQD TYKPHNRN RSGPGGSHDSRHRL 520
*****:*****:*****

```

Figure for reviewer only. Alignment of rat and human sequences (Uniprot IDs D4A055-2 and O00305-1) using the ClustalOmega webserver. The region used for the X-ray study in reference (Chen et al., 2004) (residues 49-407) is shaded in grey and the single conservative change in this region (T197/S198) is indicated with a yellow triangle.

COMMENT 3: *The RMSD cutoff (7.5 Å) used for clustering the HADDOCK-generated models of CaVβ2-actin is quite large. Consequently, 190 of 200 models were in the first cluster. How different were the models in this cluster? What was the average pairwise RMSD? And how different were the 4 models used for the subsequent in silico alanine mutational scan? It would be helpful to see the structural variability of the docking models to get an idea of the convergence of the docking calculations.*

RESPONSE 3:

The HADDOCK-generated models of Ca_vβ₂-actin complex were clustered based on the interface ligand RMSD (i-l-RMSD), using the default value of 7.5 Å. We have clarified the definition of the i-l-RMSD (page 19, lines 620-625).

To further assess the structural variability of the docking models, we have now calculated two other RMSD-based metrics employed in the CAPRI community-wide experiment (Collins et al., 2024) to evaluate protein-protein docking results: (a) the backbone interface RMSD (i-RMSD) and (b) the ligand RMSD (l-RMSD), which are also defined in the revised manuscript (page 20, lines 637-642). These new RMSD-based analyses, as well as those mentioned below, have been carried out using the Ca_vβ₂-actin and Ca_vβ₄-actin docking models newly generated in the revised version of the manuscript, to incorporate XL-MS searches with only K-K as potential cross-linking for all three search engines, according to the suggestion of reviewer #4.

These analyses indicate that the 198 Ca_vβ₂-actin complex models in the first cluster display similar protein-protein interfaces (i-RMSD value of 1.0±0.4 Å, respectively), with a slight shift of the position and orientation of Ca_vβ outside the contact interface (l-RMSD of 4.3±4.1 Å). The similarity of the protein-protein interface of the models in the first cluster is further validated by their consistency with the experimental Ca_vβ—actin inter-crosslinks (Supplementary Figure 4D). We have now added this information (pages 5-6, lines 169-177). As for the top 4 structures used in the computational alanine scanning, they show i-RMSD value of 0.9±0.6 Å (page 7, line 230).

In addition, we tested the effect of varying the i-l-RMSD cutoff on the clustering convergence; the results have been added to Supplementary Table 5A. In short, the size of the first cluster, as well as the three RMSD-based metrics (i-l-RMSD, i-RMSD and l-RMSD), changed only slightly when using both smaller and larger i-l-RMSD cutoff values than the 7.5 Å default (i.e. between 5.0 and 10.0 Å, in 0.5 Å intervals). Moreover, the top 4 structures of the first cluster are always the same, regardless of the clustering cutoff used.

COMMENT 4: *What kind of interactions are formed by the identified 8 hot spot residues in CaVβ2 with actin? Can you provide a zoom-in view on the atomic model and show the intermolecular contacts for those hot spot residues with actin?*

RESPONSE 4: We thank the reviewer for the suggestion to improve the presentation of the hotspot data. Images showing the short-range interactions formed by Ca_vβ₂ hotspot residues at the protein-protein interface with actin are shown in Supplementary Figure 10. Such images have been generated using the top (best-scored) docking structure of the top cluster. In addition, we have checked the intermolecular contacts in all top 4 structures of the first cluster using the PRODIGY webserver, as explained (page 21, lines 683-687). The results of this PRODIGY analysis, for both Ca_vβ₂ and Ca_vβ₄, are presented in the new Supplementary Table 7 and in the Results, section Identification of hotspot residues at the Ca_vβ₂/actin interaction interface (page 7, lines 236-238).

COMMENT 5: *The HADDOCK docking scores of the Ca_vβ₄-actin complex models are all positive (SI Figure 1). Why? Were there clashes in the models?*

RESPONSE 5: To investigate the presence of clashes in the Ca_vβ₄-actin models, we analyzed the top 4 structures of the selected cluster for each Ca_vβ-actin docking (Chen et al., 2010) using the MolProbity webserver. The Ca_vβ₄-actin models show a clashscore (defined as the number of steric overlaps at less than 0.4 Å per 1,000 atoms) of 7.67±0.73, which is considered to be good quality (Chen et al., 2010), and comparable to that of the Ca_vβ₂-actin models (7.47±0.22), which showed a negative HADDOCK score. Therefore, clashes are unlikely to be responsible for the positive HADDOCK scores of the Ca_vβ₄-actin models. Instead, inspection of the different energetic terms contributing to the HADDOCK score (shown in Supplementary Table 4B) show that the positive scores for the Ca_vβ₄-actin models are due to the restraint violation energy, containing contributions of both unambiguous restraints (XL-MS-derived) and ambiguous restraints (bioinformatics-based). As the Ca_vβ₄-actin models are consistent with the XL-MS data (see Supplementary Figure 13), the restraint violation energy likely stems from the bioinformatics-derived restraints.

COMMENT 6: *Did the authors try AlphaFold-Multimer for modeling the Ca_vβ₂-actin interaction?*

RESPONSE 6: We did not attempt to use AlphaFold-Multimer. Taking advantage of our long lasting experience in producing recombinant Ca_vβ, our first choice was protein-protein docking guided by experimental data.

Nonetheless, as suggested by reviewer #4, we looked into AlphaLink2 that combines AlphaFold-Multimer predictions with experimental XL-MS-based distance restraints (Stahl et al., 2024). Unfortunately, in our hands, the predictions of the Ca_vβ-actin complex were inconsistent with our data. Although we tried different settings, we were unable to predict the experimentally validated actin dimer interface that forms the binding site for Ca_vβ.

COMMENT 7: *Line 116: The XL-MS distances should be referred to as 'restraints' not 'constraints'.*

RESPONSE 7: We thank the reviewer for catching this typo, which has now been corrected.

Other questions:

COMMENT 8: *The Ca_vβ₂-actin interaction is sensitive to increased salt concentrations (Figure 2E). Even at medium NaCl concentrations (120mM) the interaction is unstable. Can the authors explain if there is evidence how stable this interaction is inside the cell?*

RESPONSE 8: The existence of stable Ca_vβ-actin complexes *in vivo* has been demonstrated using fluorescence lifetime imaging microscopy (FLIM-FRET) and single-molecule localization microscopy in HeLa cells heterologous expression system and in HL-1 cardiomyocytes (Stolting et al., 2015). We now explicitly refer to these previous results (page 2, lines 70-72).

COMMENT 9: *Are there non-synonymous SNPs in the genes for Ca_vβ₂ or actin that could affect the proposed structural interaction? Or in other words are there SNPs in the Ca_vβ₂-actin complex that are associated with calcium channelopathies?*

RESPONSE 9: We thank the reviewer for the suggestion. We have now included the new

Supplementary Table 8 with a list of the nsSNPs of human Ca_vβ₂ and Ca_vβ₄ at residue positions analogous to the eight Ca_vβ hotspots at the protein-protein interface with actin. We have also commented on this data (page 14, lines 465-469).

References for Reviewer 3

- Bullock, J.M.A., N. Sen, K. Thalassinou, and M. Topf. 2018. Modeling Protein Complexes Using Restraints from Crosslinking Mass Spectrometry. *Structure (London, England : 1993)*. 26:1015-1024.e1012.
- Chen, V.B., W.B. Arendall, J.J. Headd, D.A. Keedy, R.M. Immormino, G.J. Kapral, L.W. Murray, J.S. Richardson, and D.C. Richardson. 2010. : all-atom structure validation for macromolecular crystallography. *Acta Crystallogr D*. 66:12-21.
- Chen, Y.H., M.H. Li, Y. Zhang, L.L. He, Y. Yamada, A. Fitzmaurice, Y. Shen, H. Zhang, L. Tong, and J. Yang. 2004. Structural basis of the α₁-β subunit interaction of voltage-gated Ca²⁺ channels. *Nature*. 429:675-680.
- Cohen, S., and D. Schneidman-Duhovny. 2023. A deep learning model for predicting optimal distance range in crosslinking mass spectrometry data. *Proteomics*. 23:e2200341.
- Collins, K.W., M.M. Copeland, G. Brysbaert, S.J. Wodak, A.M.J.J. Bonvin, P.J. Kundrotas, I.A. Vakser, and M.F. Lensink. 2024. CAPRI-Q: The CAPRI resource evaluating the quality of predicted structures of protein complexes. *J Mol Biol*. 436.
- Iacobucci, C., C. Piotrowski, R. Aebersold, B.C. Amaral, P. Andrews, K. Bernfur, C. Borchers, N.I. Brodie, J.E. Bruce, Y. Cao, S. Chaignepain, J.D. Chavez, S. Claverol, J. Cox, T. Davis, G. Degliesposti, M.Q. Dong, N. Edinger, C. Emanuelsson, M. Gay, M. Götze, F. Gomes-Neto, F.C. Gozzo, C. Gutierrez, C. Haupt, A.J.R. Heck, F. Herzog, L. Huang, M.R. Hoopmann, N. Kalisman, O. Klykov, Z. Kukačka, F. Liu, M.J. MacCoss, K. Mechtler, R. Mesika, R.L. Moritz, N. Nagaraj, V. Nesati, A.G.C. Neves-Ferreira, R. Ninnis, P. Novák, F.J. O'Reilly, M. Pelzing, E. Petrotchenko, L. Piersimoni, M. Plasencia, T. Pukala, K.D. Rand, J. Rappsilber, D. Reichmann, C. Sailer, C.P. Sarnowski, R.A. Scheltema, C. Schmidt, D.C. Schriemer, Y. Shi, J.M. Skehel, M. Slavin, F. Sobott, V. Solis-Mezarino, H. Stephanowitz, F. Stengel, C.E. Stieger, E. Trabjerg, M. Trnka, M. Vilaseca, R. Viner, Y. Xiang, S. Yilmaz, A. Zelter, D. Ziemianowicz, A. Leitner, and A. Sinz. 2019. First Community-Wide, Comparative Cross-Linking Mass Spectrometry Study. *Analytical chemistry*. 91:6953-6961.
- Matthew Allen Bullock, J., J. Schwab, K. Thalassinou, and M. Topf. 2016. The Importance of Non-accessible Crosslinks and Solvent Accessible Surface Distance in Modeling Proteins with Restraints From Crosslinking Mass Spectrometry. *Molecular & cellular proteomics : MCP*. 15:2491-2500.
- Merkley, E.D., S. Rysavy, A. Kahraman, R.P. Hafen, V. Daggett, and J.N. Adkins. 2014. Distance restraints from crosslinking mass spectrometry: mining a molecular dynamics simulation database to evaluate lysine-lysine distances. *Protein science : a publication of the Protein Society*. 23:747-759.
- Mintseris, J., and S.P. Gygi. 2020. High-density chemical cross-linking for modeling protein interactions. *Proceedings of the National Academy of Sciences of the United States of America*. 117:93-102.
- Orban-Nemeth, Z., R. Beveridge, D.M. Hollenstein, E. Rampler, T. Stranzl, O. Hudecz, J. Doblmann, P. Schlogelhofer, and K. Mechtler. 2018. Structural prediction of protein

models using distance restraints derived from cross-linking mass spectrometry data. *Nature protocols*. 13:478-494.

- Piersimoni, L., P.L. Kastritis, C. Arlt, and A. Sinz. 2022. Cross-Linking Mass Spectrometry for Investigating Protein Conformations and Protein-Protein Interactions horizontal line A Method for All Seasons. *Chem Rev*. 122:7500-7531.
- Rappsilber, J. 2011. The beginning of a beautiful friendship: cross-linking/mass spectrometry and modelling of proteins and multi-protein complexes. *Journal of structural biology*. 173:530-540.
- Schiffrin, B., S.E. Radford, D.J. Brockwell, and A.N. Calabrese. 2020. PyXlinkViewer: A flexible tool for visualization of protein chemical crosslinking data within the PyMOL molecular graphics system. *Protein science : a publication of the Protein Society*. 29:1851-1857.
- Stahl, K., R. Warneke, L. Demann, R. Bremenkamp, B. Hormes, O. Brock, J. Stülke, and J. Rappsilber. 2024. Modelling protein complexes with crosslinking mass spectrometry and deep learning. *Nature communications*. 15:7866.
- Stolting, G., R.C. de Oliveira, R.E. Guzman, E. Miranda-Laferte, R. Conrad, N. Jordan, S. Schmidt, J. Hendriks, T. Gensch, and P. Hidalgo. 2015. Direct interaction of CaVbeta with actin up-regulates L-type calcium currents in HL-1 cardiomyocytes. *J Biol Chem*. 290:4561-4572.
- Wang, X., P. Cimermancic, C. Yu, A. Schweitzer, N. Chopra, J.L. Engel, C. Greenberg, A.S. Huszagh, F. Beck, E. Sakata, Y. Yang, E.J. Novitsky, A. Leitner, P. Nanni, A. Kahraman, X. Guo, J.E. Dixon, S.D. Rychnovsky, R. Aebersold, W. Baumeister, A. Sali, and L. Huang. 2017. Molecular Details Underlying Dynamic Structures and Regulation of the Human 26S Proteasome. *Molecular & cellular proteomics : MCP*. 16:840-854.

Reviewer #4 (Remarks to the Author):

The research focuses on understanding the molecular interactions between L-type CaV1.2 calcium channels and the actin cytoskeleton, highlighting the importance of these interactions in cellular quality control mechanisms that prevent the accumulation of damaged channels. Employing a combination of cross-linking mass spectrometry (XL-MS) and computational biology, the study identifies specific hotspot residues within the CaV β subunit that are crucial for its interaction with actin, impacting calcium channel functionality and signalling without affecting its interaction with the α 1 interaction domain of CaV1.2. The technical validity of the study is supported by a thorough experimental approach, including XL-MS, computational modelling, electrophysiological recordings, and mutagenesis, showcasing the critical role of the CaV β -actin interaction in maintaining functional calcium channel pools.

The findings bear significant implications for the understanding of ion channel regulation and the development of novel therapeutic strategies for neurodegenerative diseases linked to calcium signalling dysregulation. The paper suggests that disrupting the CaV β -actin interaction could serve as a potential therapeutic target for calcium channelopathies. While the data and methodology employed are robust, offering a detailed molecular insight into the CaV β -actin interaction, suggestions for improvement include a more detailed explanation of computational methods to enhance reproducibility and analytical depth:

Additionally, a detailed review of the crosslinking data revealed some major weaknesses of the study.

COMMENT 1 (Major remarks): *Crosslinker concentration titration is missing to determine optimal parameters for DSSO and DSBU respectively.*

RESPONSE 1: We apologize for omitting the results of the optimization of the XL reaction that gave the impression that we randomly selected the concentration and incubation time for each crosslinker. In fact, extensive work was necessary to establish appropriate conditions. Full images of the SDS-PAGE gels of the XL reactions for different reaction times (0, 15, 30 and 60 minutes) and different molar ratio crosslinker excess (2.5x, 5x, 10x, 20x, 50x and 100x) are now shown in the supplementary data (Supplementary Figure 1). After multiple rounds of optimization, a low amount of crosslinker (2.5x or 5x excess, for DSSO and DSBU, respectively) was found to be the concentration that minimized the occurrence of over-crosslinked products.

COMMENT 2 (Major remarks): *Merox and MetaMorpheus XL data have been searched without contaminant proteins. The FDR control of single proteins is most likely not reliable and hence contaminant proteins should be included in the fasta file*

RESPONSE 2: We agree with this point. We have now repeated the searches with Merox and MetaMorpheus (as we did for MaxLynx) using contaminant proteins. The identified intermolecular cross-links between Ca ν β and actin were not affected. The PRIDE uploads (PXD053456 and PXD053481) have been modified accordingly.

COMMENT 3 (Major remarks): *DSBU datasets show very low scores of identified crosslink matches in the MaxLynx and MeroX result files, Specifically the MaxLynx results of DSBU show only 2 unique interlinks between Actin and CaV β .*

RESPONSE 3: We agree that the results differ in quality. Most likely this arises from different instrumentation used for the analysis. The DSSO samples were analysed on an Orbitrap Eclipse with a gradient time of 60 minutes, while the DSBU samples could only be analysed with an Q Exactive Plus with a gradient time of 30 minutes. Additionally, the two cross-linking reagents differ in length by approximately 2.2 Å, which may result in different peptide identifications. Considering this experimental variation, we were reassured to observe a fair overlap between the two crosslinkers (see Supplementary Figure 3C and Supplementary Table 2).

We have explicitly reported the LC-MS/MS data acquisition settings used for each crosslinker (page 17, lines 541-543).

COMMENT 4 (Major remarks): *The numbers of unique links were boosted by searching with KSTY*

RESPONSE 4: We have now have repeated the searches considering only K-K as a potential cross-link site in the Merox software, as done with the MaxLynx and Metamorpheus packages.

The cross-linked peptides identified with the K-K search are the same as with the KSTY search. When using the K-K search, only one intermolecular cross-link **S62**-K354 (actin-Ca $\nu\beta$, found with the KSTY search in Merox) is assigned to the consecutive lysine residue **K63**-K354. The K63-K354 cross-link was also identified with MaxLynx and Metamorpheus. To account for this intermolecular cross-link, all actin-Ca $\nu\beta$ docking models were newly generated. Figures 2D-F and Supplementary Table 2 have been revised accordingly.

COMMENT 5 (Major remarks): *DSSO dataset: zhrm file is missing for sample 14, Merox shows only quality data for sample 12 with 5 high scoring inter links, MaxLynx reports only 11 unique inter links with relatively low scores, MetaMorphues reports 43 unique inter links.*

RESPONSE 5: We have now re-uploaded all relevant data files in a new PRIDE submission (PXD053456 and PXD053481, for DSSO and DSBU, respectively). Detailed information on the individual search results can also be found in the revised manuscript (pages 4-5, line 129-146).

The relatively low scores obtained by Merox, MaxLynx and Metamorpheus are probably due to the low yield of Ca $\nu\beta_2$ -actin cross-linked peptides obtained, despite the major efforts to optimize the cross-linking reactions. Nevertheless, we rationalized that using the information contained in these relatively low scoring inter-XLs is an effective strategy because (i) we are dealing with a well-established protein-protein interaction between Ca $\nu\beta_2$ and actin, (ii) we now considered contaminating proteins and applied an FDR of 0.01 at the CSM level for all three search engines, and (iii) we implemented additional confidence criteria to cross-check the quality of the XL-MS data by: discarding the inter-XLs that were identified by only one search engine and then filtering the remaining inter-XLs (i.e. identified by at least two programs) with the DisVis webserver to detect false positives at the distance restraint level.

In summary, we feel confident about the reliability of our strategy, as the PPI surface predicted by the combination of these inter-XLs and protein-protein docking was unequivocally validated by site-directed mutagenesis and *in vitro* and *in cell* assays.

COMMENT 6 (Major remarks): *Interestingly, MaxLynx and MetaMorpheus are searched with K-K and MeroX with KSTY, why is this the case? The search should be performed similarly for all search engines*

RESPONSE 6: We now repeated the Merox search with K-K as a potential crosslinking site, as we did for the MaxLynx and Metamorpheus software searches. Please refer also to answer 4.

COMMENT 7 (Major remarks): *A Venn diagram would be valuable to show the overlap between the search results of all search engines (for DSSO and DSBU separately)*

RESPONSE 7: We have added the requested Venn diagrams in Supplementary Figure 3A-B and summarized the search results in more detail in Supplementary Tables 1 and 2.

COMMENT 8 (Major remarks): *Additionally, the overlap of DSSO and DSBU should also be shown as a Venn diagram to pinpoint the reader to the final crosslinks used for modelling and structure refinement.*

RESPONSE 8: We have added the requested Venn diagram showing the overlap of DSSO and DSBU (22 unique inter-XLs) in Supplementary Figure 3C. In addition, we present in Supplementary Table S2 the final crosslinks used for XL-MS-guided protein-protein docking (11 self-consistent inter-XLs) that were obtained by filtering those identified with two or more search engines (16 selected unique inter-XLs) with the DisVis webserver.

COMMENT 9 (Major remarks): *The description of the size exclusion experiment is missing*

RESPONSE 9: We did not use size-exclusion chromatography upstream of C18 peptide separation for purification/enrichment of XL-linked peptides due to the low yield of cross-linked products. Size-exclusion chromatography was performed after Ni²⁺ affinity chromatography to obtain highly purified recombinant Cavβ (page 16, lines 513-518). We have also implemented a workflow that describes the individual steps from the preparation of the proteins to the experimental validation (Figure 1).

COMMENT 10 (Minor remarks): *-Non-crosslink control missing in Fig. 1 panels C and D*

RESPONSE 10: We have included the non-crosslink controls in Figure 2C of the revised version (formerly Figure 1C-D).

COMMENT 11 (Minor remarks): *60 min crosslink reaction time is quite long, is the Cav β -actin complex formation/ stability influenced by such a long time at RT? Please show data to validate that the complex is stable for such a long time at RT*

RESPONSE 11: We have now performed additional F-actin co-sedimentation assays for Cav β and actin with a prolonged incubation time to test the stability of the complex formation over time. We compared the fraction of Cav β bound to actin after incubation periods of 30 minutes (the incubation time used in the experiments throughout the manuscript) and 90 minutes. The results show statistically non-significant differences in the fraction of Cav β bound to actin after 30 and 90 minute incubation time, indicating that the complex formation is stable for at least 90 minutes. These new results are included in Supplementary Figure 2.

COMMENT 12 (Minor remarks): *The level of FDR estimation is missing, three different software tools were used to analyze crosslinking data, but each software tool reports FDR on a different crosslink level, e.g MeoX reports only CSM level FDR*

RESPONSE 12: We applied an FDR of 0.01 at the CSM level for all three software tools (page 17, line 562). Contaminant protein databases provided by the software packages were included in the searches (page 17, lines 564-565).

COMMENT 13 (Minor remarks): *If possible, config files of the searches should also be uploaded as well, these files contain all details and settings of the search and FDR estimation.*

RESPONSE 13: We have documented the search settings in detail and uploaded the data files including the config files in a new PRIDE submission (PXD053456 and PXD053481).

COMMENT 14 (Minor remarks): *The representation of crosslinks in Fig. 1 E bottom is not ideal, maybe choose the line representation to see from which residue to which residue the crosslink goes*

RESPONSE 14: We have replaced the former Figure 1E with the new Figure 2D with the line representation of the crosslinks.

COMMENT 15 (Minor remarks): *It would be also interesting to see the results in combination with AlphaLink (<https://github.com/Rappsilber-Laboratory/AlphaLink2>)*

RESPONSE 15: Thank you for the suggestion to use the AlphaLink2 software. Unfortunately, in our hands, the predictions of the Cav β -actin complex were inconsistent with our data. Although we tried multiple configurations – AB, AAB, ApGAB, and AAAB (with A, actin monomer; B, Cav β ; ApGA, dimer of actin monomers n and $n+2$ joined by a polyglycine linker (pG); AAA, trimer of actin monomers n , $n+1$ and $n+2$) – we were unable to predict the actin dimer interface that according to our XL-MS data encompasses the binding site for Cav β . Although AlphaLink2 appears promising for deep learning-based structural

prediction of protein-protein complexes, we feel that further optimization of the Cav β -actin model is out of the scope of our manuscript.

COMMENT 16: *Overall, the study could benefit from a more comprehensive explanation of computational modelling techniques, statistical analyses and improvement of the crosslinking data/ presenting the data, to make the findings more accessible to a wider audience. Finally, this research provides valuable insights into the molecular details of calcium channel quality control mechanisms and their implications for cell physiology and pathology, setting a foundation for future studies in this field.*

RESPONSE 16: We thank the reviewer for the positive comments. We have revised extensively the XL-MS and modeling results according to the comments of all reviewers. We feel confident that our strategy of combining XL-MS and protein-protein docking has generated a reliable structural model of the interaction between Cav β and actin, which was unequivocally validated by experimental work.

Reviewer #5 (Remarks to the Author):

This paper outlines contacts between the beta subunit of the calcium channel and F-actin in the cell, proposing a role in this interaction in regulation of damaged channel proteins in the cell. This work represents an advance in our understanding of calcium currents generated during various cellular processes, and the beta subunit/F-actin models built will provide many testable hypotheses dealing with disease states arising from defects in calcium signaling pathways. The data presented are clear and supportive of the conclusions, however, some clarifications in the text are needed.

COMMENT 1: *I have some questions about the docking calculations used to generate the model of Ca_vβ bound to F-actin. Some more discussion of why certain clusters were discarded would be helpful. As a control, was the docking ever run without XL constraints and did the same pose come up near the top of the list? If so, that would strengthen the use of the docking program in the structure determination. The XL constraints also spanned two actin monomers along the long pitch helix which assumes the Ca_vβ fragments bind to the shallow aspect of actin, which is fine since a number of proteins bind to actin in this spot, but was the docking ever run assuming binding to a single actin monomer gave rise to the XL data? If the assumption about two actin binding is correct, one would expect poor docking poses in terms of scores, buried surface, etc when assuming the XL comes from only one actin, again strengthening the approach taken here.*

RESPONSE 1: Here and in the responses below we refer to the Ca_vβ₂-actin and Ca_vβ₄-actin docking models newly generated to incorporate XL-MS searches with only K-K as potential cross-linking for all three search engines, according to the suggestion of reviewer #4.

For clarity, the first comment of the reviewer with the corresponding answers have been divided in three parts.

1.1. I have some questions about the docking calculations used to generate the model of Ca_vβ bound to F-actin. Some more discussion of why certain clusters were discarded would be helpful.

We followed the HADDOCK protocol in which clusters are discarded based on the value of the HADDOCK score, i.e. the clusters whose HADDOCK scores did not overlap within their standard deviations with that of the best scored (top) cluster are discarded. The Ca_vβ₂-actin docking resulted in one single cluster and the Ca_vβ₄-actin docking in five. For Ca_vβ₄, clusters #1, #2 and #5 followed the HADDOCK score selection criterion, whereas clusters #3 and #4 could be discarded (Supplementary Table 4B). Among the three clusters with overlapping HADDOCK scores for Ca_vβ₄, we selected cluster #1 due to its similarity with the Ca_vβ₂-actin model. The ligand RMSD (i.e. the backbone RMSD of Ca_vβ after superimposing the actin dimer) is 4.8 Å for cluster #1, compared to 12.7 and 12.2 Å for clusters #2 and #5, respectively. Similarly, the interface RMSD (i.e. the backbone RMSD of the PPI surface after superimposing Ca_vβ and actin residues within 10 Å of the other partner) is 2.7, 3.3 and 6.5 Å for clusters #1, #2 and #5, respectively.

The rationale for this choice is that the protein-protein interface with actin is expected to be conserved across the Ca_vβ family of proteins, due to their high sequence and structure conservation. Moreover, the alignment between the amino acid sequences of Ca_vβ₂ and Ca_vβ₄ (new Supplementary Figure 12) showed that all four lysine residues involved in the set of self-consistent inter-molecular cross-links identified in Ca_vβ₂ and used for the modeling of Ca_vβ₂/actin interaction are fully conserved in Ca_vβ₄. Computational and *in vitro* mutagenesis, followed by binding assays with F-actin, further supported our cluster selection criteria for Ca_vβ₄.

We have now explained better the cluster selection criteria (page 9, lines 274-286).

1.2 As a control, was the docking ever run without XL constraints and did the same pose come up near the top of the list? If so, that would strengthen the use of the docking program in the structure determination.

We have now carried out several control simulations without XL constraints, using the *ab initio* docking options available in HADDOCK. Between 98.5 and 99.5% of the models obtained showed $\text{Ca}_v\beta_2$ -actin inter-crosslink $\text{C}\alpha$ - $\text{C}\alpha$ distances that violate the upper bound distance of 35 Å used to check consistency with the XL-MS data (new Supplementary Table 6). Moreover, the top 10 scored models of the control simulations display l-RMSD values around 70 Å with respect to the experimentally validated $\text{Ca}_v\beta_2$ -actin model. Furthermore, the *ab initio* model closest the XL-MS-guided docking model (with an l-RMSD of 14.0 Å and shown in Supplementary Figure 5) was not among the best scored models. These results show that the use of restraints to guide HADDOCK docking is fundamental to identify the $\text{Ca}_v\beta_2$ -actin interface.

We have explained these control simulations without restraints (page 6, lines 184-188).

1.3. The XL constraints also spanned two actin monomers along the long pitch helix which assumes the Ca_v beta fragments bind to the shallow aspect of actin, which is fine since a number of proteins bind to actin in this spot, but was the docking ever run assuming binding to a single actin monomer gave rise to the XL data? If the assumption about two actin binding is correct, one would expect poor docking poses in terms of scores, buried surface, etc when assuming the XL comes from only one actin, again strengthening the approach taken here.

We now show the results of the docking simulation assuming binding of $\text{Ca}_v\beta_2$ to a single actin monomer (Supplementary Tables 3B and 4C and Supplementary Figures 6 and 7). Out of the eleven $\text{Ca}_v\beta_2$ -actin inter-crosslink $\text{C}\alpha$ - $\text{C}\alpha$ distances, three displayed distributions spanning beyond the upper bound distance of 35 Å, compared to none for the actin dimer docking (Supplementary Figure 6D). In addition, two out of these eleven self-consistent inter-XLs were estimated to have a higher average violated fraction for monomeric than dimeric actin (Supplementary Table 3B).

The reported HADDOCK scores and buried surface areas can only be compared meaningfully across models within the same docking run, which precludes the comparison between $\text{Ca}_v\beta_2$ bound to dimeric actin and $\text{Ca}_v\beta_2$ bound to monomeric actin. This is because these parameters depend on the size of the complex, the number of restraints used and the solvent accessible surface area of the two interacting proteins alone, which are not comparable between docking runs with ternary and binary complexes.

Besides that, the $\text{Ca}_v\beta_2$ -actin monomer docking model disagrees with previous experimental data on the $\text{Ca}_v\beta_2$ /actin interaction (Stolting et al., 2015). *In vitro* assays have shown that both GK and SH3 domains contribute to binding actin filaments. However, the SH3 domain does not establish any intermolecular contacts with actin in the $\text{Ca}_v\beta_2$ -actin monomer model (new Supplementary Figure 6A). Moreover, *in cell* experiments have shown that $\text{Ca}_v\beta_2$ interacts with actin filaments and pharmacological depolymerization of actin filaments disrupts the interaction with $\text{Ca}_v\beta_2$, suggesting that $\text{Ca}_v\beta$ does not bind to G-actin. In contrast, the docking model of $\text{Ca}_v\beta_2$ to one actin subunit is incompatible with the interaction of $\text{Ca}_v\beta$ with actin filaments, as binding of $\text{Ca}_v\beta_2$ to the $n+2$ subunit would result in the SH3 domain clashing with the adjacent (n) subunit (see new Supplementary Figure 6B). Altogether, the inconsistency of the $\text{Ca}_v\beta_2$ -actin monomer complex model with the available experimental data supports our assumption that $\text{Ca}_v\beta_2$ binds to two adjacent actin subunits, at a

protein-protein interface used by other actin binding proteins (Merino et al., 2018; Merino et al., 2020; Oosterheert et al., 2022).

We have explained the results of this control simulation with an actin monomer (pages 6-7, lines 201-207).

COMMENT 2: *The iRMSD values need to be better defined to understand what they mean. I am assuming here the iRMSD compares the highest scoring docking pose calculated using the residues predicted to form the PPI to all the other docking poses. The value only really shows how effective the clustering was. A more useful analysis is the figures in Supplemental Figure 1B and D, where we can see how well the docking matches the XL results. Comparison of this result (fit of the XL data) among the other clusters would be useful to strengthen the dismissal of those clusters.*

RESPONSE 2:

The reviewer is right assuming that the iRMSD “compares the highest scoring docking pose calculated using the residues predicted to form the PPI to all the other docking poses”, with the exception that the interfacial residues belonging to actin were used for the superposition of the docking structures and only those belonging to Ca_vβ are included in the RMSD calculation. For clarity, we now use the acronym i-I-RMSD to refer to this interface ligand RMSD (where Ca_vβ is considered the “ligand” of the protein-protein complex due to its smaller size compared to actin). We have now better explained that the RMSD used for the clustering is the i-I-RMSD (page 19, lines 620-626).

We thank the reviewer for the suggestion that “Comparison of this result (fit of the XL data) among the other clusters would be useful to strengthen the dismissal of those clusters”. The consistency with the interXL distances is now shown for *all* docking clusters in Supplementary Figure 4D (Ca_vβ₂) and Supplementary Figure 13D-H (Ca_vβ₄). To respond to a specific comment of reviewer #4, we performed new XL-MS searches with only K-K as potential cross-linking to uniform the search with the three software (Merox, Metamorpheus and MaxLynk) and re-generated all actin–Ca_vβ HADDOCK docking models. The new docking simulations resulted in a single cluster for Ca_vβ₂ and five for Ca_vβ₄. For the latter, and following the “HADDOCK best practice guide”, clusters 3 and 4 were discarded on the basis of their HADDOCK score being significantly less favorable than for the other three clusters (see Supplementary Table 4B). Clusters #1, #2 and #5 showed HADDOCK scores within their standard deviations and thus we checked the consistency of the respective models with the XL-MS data. All Cα–Cα cross-linked pair distances (arising from the 11 self-consistent inter-XLs) between Ca_vβ₄–actin are within the 35 Å threshold for the three clusters with overlapping HADDOCK scores (Supplementary Figure 13D, E and H).

Given that the fit of the XL data did not allow dismissal of any of those clusters, we consider the similarity with the experimentally validated Ca_vβ₂-actin model to select cluster #1; see in the Results, section Integrating Ca_vβ₂-derived XL-MS data for Ca_vβ₄ and actin computational docking (page 9, lines 274-286). *In vitro* mutagenesis and binding assays with Ca_vβ₄ and actin supported our cluster selection; please see also our answer to comment #1.

COMMENT 3: The actin numbers in the text/figures all appear to be off by 2. Residue 330 in PDB 500E is not Lys; it's Ile. I think it should be Lys328. Similarly, Ser62 should be Ser60, etc.

RESPONSE 3: We thank the reviewer for catching this shift. The numbering in the PDB is shifted by 2 with respect to the numbers used throughout the manuscript, which are based on the sequence of

alpha actin from rabbit skeletal muscle, as listed in UniProtKB (ID P68135). Residue 330 in the UniProt sequence is Lys and it corresponds to Lys328 in the PDB; similarly, Ser62 in UniProt is Ser60 in the PDB, etc. We have now added the accession number to the legend of Figure 2D.

COMMENT 4: *On page 6, the RMSD comparing CaV beta 2 and CaV beta 4 dockings of 3.1 Å suggests significant differences between the poses, not similarity as stated. More information on how this number was generated is needed. For instance, what is the RMSD of the two CaV models to each other when aligned? If it's high, that would also be reflected in a high RMSD for the docked poses. Additionally, the calculation of the RMSD should be performed with only the PPI residues' backbones, to show conservation of the actin binding site. What is the sequence identity of the predicted PPIs?*

RESPONSE 4:

We used the ligand RMSD (l-RMSD) that reports the backbone RMSD of Ca_vβ₄ to Ca_vβ₂ after superimposing the actin dimers of both complexes. In order to account for protein flexibility, we did not consider only the best scored model, but an average of the top 4 docking structures of the best cluster for each Ca_vβ. According to the newly generated docking models (see previous responses), the updated l-RMSD is 4.8 Å. We now additionally calculated the backbone RMSD of the two Ca_vβs after superimposing the average docking models using Ca_vβ as reference (resulting in an RMSD of 1.6 Å) and the backbone RMSD of the PPI residues (i-RMSD = 2.7 Å), as well as the sequence identity of the Ca_vβ interfacial residues in common between Ca_vβ₂-actin and Ca_vβ₄-actin complexes (86%).

To address the reviewer's comment, we added the i-RMSD and the PPI sequence identity in the manuscript (page 9, lines 282-286).

COMMENT 5: *In Figure 1D, the dotted box appears to be in the standard lane, not the sample lane. It's correct in Figure 1C.*

RESPONSE 5: We are sorry for this misunderstanding that is due the fact that the SDS-PAGE gel shown in Figure 1D (now Figure 2C) was cropped removing the lane with the molecular mass standard, so that lane 1 is indeed the cross-link reaction with DSBU; the full-size image is shown in Supplementary Figure 1B. For clarity, the dotted box is now replaced by an arrow.

COMMENT 6: *PDB 5OOE was used for the actin model in docking; this is an ATP analog bound to F-actin. Why was this structure used? That would be the form of actin on the barbed end of "young" actin, shouldn't the CaV beta bind "aged" ADP actin? Not only that, but there are better resolution structures out there. However, this is unlikely to affect the results much, I'm just curious.*

RESPONSE 6: We selected PDB 5OOE for the actin model because it was one of the highest resolution cryo-EM structures of F-actin available at the time we started this project in October 2020. We agree with the reviewer that the docking results will not be affected much by the actin structure used, since the ADP, APD+Pi and ATP-bound actin are similar and the nucleotide state does not induce large protein rearrangements (Oosterheert et al., 2022; Reynolds et al., 2022).

We thank the reviewer for bringing to our attention the possibility of Ca_vβ sensing actin's age. We looked into the actin structural elements whose conformational changes have been shown to act as age markers, i.e. the D-loop (residues 41-55, using the UniProt numbering), the W loop (residues 167-174) and the C-terminal tail (residues 351-377) (Merino et al., 2018; Oosterheert et al., 2022). The actin residues V45, G48 and M49 (D-loop), Y168 and E169 (W-loop) and L351, S352 and T353 (C-terminal tail) form intermolecular contacts with Ca_vβ hotspots (Supplementary Table 7). Although, this

evidence hints at a potential ability of $Ca_v\beta$ to distinguish between aged and young actin, further experiments would be needed to test this idea.

References for Reviewer 5

- Merino, F., S. Pospich, J. Funk, T. Wagner, F. Kullmer, H.D. Arndt, P. Bieling, and S. Raunser. 2018. Structural transitions of F-actin upon ATP hydrolysis at near-atomic resolution revealed by cryo-EM. *Nature structural & molecular biology*. 25:528-537.
- Merino, F., S. Pospich, and S. Raunser. 2020. Towards a structural understanding of the remodeling of the actin cytoskeleton. *Seminars in cell & developmental biology*. 102:51-64.
- Oosterheert, W., B.U. Klink, A. Belyy, S. Pospich, and S. Raunser. 2022. Structural basis of actin filament assembly and aging. *Nature*. 611:374-379.
- Reynolds, M.J., C. Hachicho, A.G. Carl, R. Gong, and G.M. Alushin. 2022. Bending forces and nucleotide state jointly regulate F-actin structure. *Nature*. 611:380-386.
- Stolting, G., R.C. de Oliveira, R.E. Guzman, E. Miranda-Laferte, R. Conrad, N. Jordan, S. Schmidt, J. Hendriks, T. Gensch, and P. Hidalgo. 2015. Direct interaction of CaVbeta with actin up-regulates L-type calcium currents in HL-1 cardiomyocytes. *J Biol Chem*. 290:4561-4572.

ANSWER TO REVIEWER'S COMMENTS

Reviewer #1 (Remarks to the Author):

1. *It now seems clear to me there is no $\alpha 2\delta$ cDNA included in the co-expression and functional studies, which means that the folding and function of the channel complex will be abnormal. The EMC complex in the endoplasmic reticulum interacts with both the $Ca_v1.2$ and the β subunit and this is displaced by $\alpha 2\delta$ [1]. Therefore without this key component of the calcium channel complex, the findings are unreliable.*

We respectfully disagree with the reviewer's comment that in the absence of $\alpha 2\delta$ "the folding and function of the channel complex will be abnormal" and that this issue impacts our findings. Our electrophysiological recordings of macroscopic currents and gating currents, in the absence of $\alpha 2\delta$, unequivocally show the occurrence of trafficking- and functionally-competent $Ca_v1.2\alpha_1/Ca_v\beta 2$ channel core complexes assembled at the plasma membrane. These argue against the occurrence of major abnormalities in the $\alpha 2\delta$ -deficient $Ca_v1.2$ channel complexes. In line with this, pioneer experiments using transiently transfected HEK cells for expression of Ca_v channel complexes showed that expression of $Ca_v\alpha 2\delta$ along with $Ca_v1.2/Ca_v\beta 2$ core complex resulted in no differences in the voltage-dependence of activation but in faster activation kinetics and around two-fold increase in the ionic current amplitude by around a factor of 2 accompanied with increased charge moved (Q_{on}) as compared with $Ca_v1.2/Ca_v\beta 2$ core alone (Bangalore et al., 1996). On the other hand, cardiomyocytes from $Ca_v\alpha 2\delta$ knock-out animals show normal $Ca_v1.2$ currents with a 10 mV shift towards hyperpolarized voltages in the activation curve and decreased current density as compared with wild-type cells (Fuller-Bicer et al., 2009). Similar results with $Ca_v\alpha 2\delta$ knockdown dysgenic myotubes reconstituted with $Ca_v1.2$ were obtained (Obermair et al., 2008). Altogether, these studies show that $Ca_v\alpha 2\delta$ subtly modulates certain channel properties (depending on the Ca_v s coexpressed) and does not compromise channel function.

We have tested the Reviewer's idea and repeated the electrophysiological experiments in cells coexpressing the $Ca_v1.2/\beta$ channel core complex along with $\alpha 2\delta$ subunit. Currents mediated by the $Ca_v1.2/\beta 2/\alpha 2\delta 1$ tripartite channel complex channel marginally differs from that obtained with the $Ca_v1.2/\beta 2$ core complex. We did observe the effect of $\alpha 2\delta 1$ in speeding the kinetics of activation of the channel complex, which confirmed the expression of this subunit in the recorded cells. In the background of this tripartite channel complex, the actin-association-deficient $Ca_v\beta 2$ hotspot mutant continued downregulating current amplitudes while preserving channel numbers. These results further confirm our original findings using the $Ca_v1.2/\beta$ channel core complex and show that under our experimental conditions, $\alpha 2\delta$ is not as necessary as suggested by the Reviewer; $\alpha 2\delta$ is neither mandatory for the channel functional assembly nor for its clearance at the plasma membrane. The results are summarized in the Figure for Reviewer, and explicitly contextualized in the discussion according to Chen et al. Please refer to page 15, line 475-481.

Figure for Reviewer 1. Expression of Ca_vα₂δ₁ subunit marginally affects the functional competence of the Ca_v1.2/Ca_vβ₂ core complex in HEK293 cells. **A**, Representative normalized ionic current traces at +20 mV and plot of the mean τ_½ values (± S.E.M) for the current activation at different voltages from cells expressing Ca_v1.2/Ca_vβ₂ (either Ca_vβ₂ WT or Ca_vβ₂ 8Ala hotspot mutant) with and without Ca_vα₂δ₁. * p < 0.05; ***p < 0.001; t-test. Ca_vα₂δ₁ speeds up the activation kinetics. **B**, Current-to-voltage (I/V) plot and fraction of activated channels versus voltage plot obtained from cells expressing Ca_v1.2/β₂/α₂δ₁ channels encompassing either Ca_vβ₂ WT (n = 20) or Ca_vβ₂ 8Ala hotspot mutant (n = 15). Ionic currents were elicited by steps to voltages between -50 to +50 mV in 5 mV increments from a holding potential of -90 mV. Inset shows the box and whisker plot of the total charge movement (Qon). Qon was calculated from the integral of the On gating current during the voltage step to the reversal potential for the carrier ion determined empirically by stepping to several potentials in 2 mV increments. Median and mean values of the data are indicated in each box by a continuous and dashed lines, respectively, and whiskers indicate 95% confidence. Mean values ± S.E.M: I_{max}, -1.1 ± 0.1 nA and -0.7 ± 0.1 nA (p = 0.03), Qon, 491 ± 52 fC and 432 ± 107 fC for Ca_v1.2/β₂/α₂δ₁ encompassing Ca_vβ₂ WT and 8Ala mutant (p = 0.6) respectively. UniProtKB accession numbers: Ca_v1.2 (P15381), Ca_vβ₂ (Q8VGC3-2) and Ca_vα₂δ₁ (P54289-1). Ca_v1.2 and Ca_vβ₂ have been used throughout this study and Ca_vα₂δ₁ in Scholl et al., *Nature Genetics* **45**, 1050-4 (2013).

2. Abstract “... uncovered a role in replenishing damaged Ca_v1.2 channels”. The authors provide no evidence that there are such damaged channels in the membrane, or what they mean by a damaged channel.

Noise analysis established that the reduction in the ionic current mediated by the actin-association-deficient Ca_vβ2 hotspot mutant occurs at identical single-channel properties and channel numbers at the cell surface. Current downregulation can be fully explained by the decrease in the number of functionally available channels. This necessarily implies the occurrence of an increased fraction of conduction-defective channels that contribute to gating but not to ionic currents. In our original version, we referred to those as conduction-damaged or damaged channels. In response to the reviewer’s concern, we now refrain from using the term damaged alone, and use all over the manuscript the term conduction-defective channels, which exactly describes the results of our experiments.

3. “The mutant β2 with the 8 Alanine mutations clearly is poorly functional with respect to the function of the channels”, the authors have not shown directly that this is due to the effect on actin interaction.

The mutant β2 with the 8 Alanine mutations is **not** poorly functional with respect to the function of the channels. Our experimental work using ionic and gating currents recordings combined with noise analysis of the macroscopic currents, show that the Ca_vβ2 mutant with the 8 alanine mutations at hotspot residues for actin binding is fully functional and virtually undistinguishable from the wild-type Ca_vβ with respect to channel activation (Figure 6D), unitary properties (Figure 7B-C), and number of channels at the plasma membrane (Figure 6E). Besides reducing its affinity for actin filaments and the macroscopic ionic currents caused by a concomitant decrease in the number of functionally active channels assembled at the plasma membrane (Figure 6 and 7), the Ca_vβ2 hotspot mutant displayed no other functional alteration; its cellular localization, expression levels, association capability with the Ca_vα1 pore-forming interaction domain and structural integrity were preserved.

Now, to directly address the issue that the Ca_vβ2 hotspot mutant downregulates Ca_v1.2-mediated currents “due to the effect on actin interaction”, we conducted electrophysiological recordings in cells treated with the actin filament disruptor cytochalasin D. Earlier experiments showed that pharmacological disruption of the actin filaments by cytochalasin D inhibits the interaction between actin and Ca_vβ2 in living cells (Stölting et al., 2015). Therefore, we rationalize that if indeed the effect of the actin-association-deficient Ca_vβ2 mutant is mediated by its impaired interaction with actin filaments, then it is expected that subsequent disruption of actin filaments has little further effect on Ca_v-mediated currents.

Time-lapse confocal fluorescence images of tsA201 cells stained for actin filaments demonstrate effective filament disruption under our experimental conditions. Cytochalasin D reduced ionic currents and preserved gating currents (Q_{on}) mediated by Ca_v1.2/Ca_vβ2 WT complex as compared with non-treated cells (p = 0.02 and p = 0.24, respectively). This resembles the effect of the hotspot mutant. Moreover, we found statistically non-significant difference in the average peak current amplitudes for Ca_v1.2 channels complexed with Ca_vβ2 WT and those with Ca_vβ2 8Ala mutant in cytochalasin D-treated cells (p = 0.25). Therefore, disruption of the actin filaments blunts the current reduction mediated by the actin-association-deficient Ca_vβ2 mutant, indicating that the effect of this mutant is a direct consequence of its impaired interaction with actin filaments and not a general effect.

These results are shown in the new Supplementary Figure 16 and described in page 12, line 384-401.

4. The reason for the inclusion of the R-X $\beta 4$ mutant is still unclear to me, it apparently interacts less well with actin, but still produces normal calcium currents, which does not support their hypothesis.

$\text{Ca}_v\beta 4$ was included with the aim to expand our studies to other member of the $\text{Ca}_v\beta$ family and test the robustness of our integrative strategy applied to define the interaction surface between $\text{Ca}_v\beta$ and actin. The use of the R482X originated from the fact that no crystallography data is available for the variable regions of $\text{Ca}_v\beta$ (outside the SH3 and GK conserved domains), which precludes the elucidation of their contribution to the $\text{Ca}_v\beta$ -actin interaction from the Haddock models. This implies that any recombinant $\text{Ca}_v\beta 4$ lacking variable segments, which confer instability under the lower salt conditions used in the actin binding assay (Guzman et al., 2019), was a priori equally suited for the experimental validation. The natural occurring R482X mutant, lacking only the 39 C-terminal amino acids, was the longest version of $\text{Ca}_v\beta 4$ that fulfilled this condition. These points have been already discussed in the section *Integrating $\text{Ca}_v\beta 2$ -derived XL-MS data for $\text{Ca}_v\beta 4$ and actin computational docking*, pages 8, line 264 ff.

$\text{Ca}_v\beta 4$ R482X, the R-X $\beta 4$ mutant, interacts better (not less well) with actin (Figure 5B). Neither R482X, which improves actin binding, nor the actin-association-deficient R482X hotspot mutant alters current amplitudes when coexpressed with $\text{Ca}_v\alpha 1$ ($\text{Ca}_v 1.2$ and $\text{Ca}_v 2.1$). This highlights the functional specificity of $\text{Ca}_v\beta 2$ uncovered in this study and expands the list of its specific functions that are not common to all members of the family (as the modulation of Ca_v current density) (Our Review and Rima). Homologous proteins that share a common function and exhibit a variety of more specific functions are not rare (Capra and Singh, 2008). Moreover, regions determining this functional specificity are commonly found on flexible or disordered segments (Cumberworth et al., 2013; Pancsa and Fuxreiter, 2012). $\text{Ca}_v\beta 2$ and $\text{Ca}_v\beta 4$ used in this work, display different interactors and preferential subcellular compartmentalization (Rima et al., 2016); $\text{Ca}_v\beta 2$ associates with the plasma membrane due to palmitoylation sites at the N-terminal region (Figure 6) and $\text{Ca}_v\beta 4$ (and $\text{Ca}_v\beta 4$ R482X, Figure 8) is mostly located into the nucleus. We now extended this explanation into the text (pages 13 and 16, lines 422-425 and 509-511, respectively).

An added bonus of using $\text{Ca}_v\beta 4$ R482X, which is associated with a rare form of epilepsy but with unclear etiology, is that the availability provided by our study of $\text{Ca}_v\beta 4$ R482X hotspot mutants for acting binding will allow in the future to test the hypothesis whether or not this interaction play a role in the diseased condition. These points have been already discussed in pages XX and XX, lines XX and XX, respectively). Altogether, we believe that the results with $\text{Ca}_v\beta 4$ R482X represent valuable data to be reported.

References for Reviewer 1

- Bangalore, R., G. Mehrke, K. Gingrich, F. Hofmann, and R.S. Kass. 1996. Influence of L-type Ca channel alpha 2/delta-subunit on ionic and gating current in transiently transfected HEK 293 cells. *Am.J.Physiol.* 270:H1521-1528.
- Capra, J.A., and M. Singh. 2008. Characterization and prediction of residues determining protein functional specificity. *Bioinformatics.* 24:1473-1480.
- Cumberworth, A., G. Lamour, M.M. Babu, and J. Gsponer. 2013. Promiscuity as a functional trait: intrinsically disordered regions as central players of interactomes. *The Biochemical journal.* 454:361-369.
- Fuller-Bicer, G.A., G. Varadi, S.E. Koch, M. Ishii, I. Bodi, N. Kadeer, J.N. Muth, G. Mikala, N.N. Petrashevskaya, M.A. Jordan, S.P. Zhang, N. Qin, C.M. Flores, I. Isaacsohn, M. Varadi, Y. Mori, W.K. Jones, and A. Schwartz. 2009. Targeted disruption of the voltage-dependent calcium channel alpha2/delta-1-subunit. *Am J Physiol Heart Circ Physiol.* 297:H117-124.
- Guzman, G.A., R.E. Guzman, N. Jordan, and P. Hidalgo. 2019. A Tripartite Interaction Among the Calcium Channel α 1- and β -Subunits and F-Actin Increases the Readily Releasable Pool of Vesicles and Its Recovery After Depletion. *Frontiers in cellular neuroscience.* 13:125.
- Obermair, G.J., P. Tuluc, and B.E. Flucher. 2008. Auxiliary Ca(2+) channel subunits: lessons learned from muscle. *Current opinion in pharmacology.* 8:311-318.
- Panca, R., and M. Fuxreiter. 2012. Interactions via intrinsically disordered regions: what kind of motifs? *IUBMB life.* 64:513-520.
- Rima, M., M. Daghani, Z. Fajloun, R. M'Rad, J.L. Bruses, M. Ronjat, and M. De Waard. 2016. Protein partners of the calcium channel β -subunit highlight new cellular functions. *The Biochemical journal.* 473:1831-1844.
- Stölting, G., R.C. de Oliveira, R.E. Guzman, E. Miranda-Laferte, R. Conrad, N. Jordan, S. Schmidt, J. Hendriks, T. Gensch, and P. Hidalgo. 2015. Direct interaction of Ca ν β with actin up-regulates L-type calcium currents in HL-1 cardiomyocytes. *The Journal of biological chemistry.* 290:4561-4572.

Reviewer #4 (Remarks to the Author):

My comments have been fully addressed, resulting in new figures in the main text and supplement as well as new data uploads and updated text paragraphs. Figure 1 really helps the reader to follow the experimental steps and design. In general, changes regarding the crosslink section improved the understanding of the data analysis and selection of crosslinks used for modeling/ docking a lot. Thank you for your effort.

I have just a few minor comments about the crosslink experiments:

1. *The SDS-Pages of crosslinker optimization suggest, that the crosslinker concentration with 60 min and 100x molar excess worked the best. One would be maybe afraid that over crosslinking has happened, but with over crosslinking you would probably see background grease or something like this on the gel as well. Did the authors also measure the 100x samples?*

We did not prepare and measure the 100x samples. Although, the cross-linked bands appeared more intense in the 100x samples, the specific cross-linked products between the two proteins (i.e, not observed in the control reactions) were detected only at low molar excess (≤ 5), and not with 100 molar excess. On top of that, and as pointed out by the reviewer, we were also concerned about over crosslinking under the latter condition.

2. *The difference between the DSSO and DSBU data, does probably not come from the space length difference because both spacer length are longer then structurally needed and the SDS gel from the DSBU experiment looks even better as for the DSSO experiment. I rather think measuring the DSBU samples on a 50 cm column with 30min gradient was a not ideal combination. Even with low input amount, crosslink samples benefit from rather longer gradients then shorter ones. Using different instruments adds variability on top of that. Why did the authors not choose one instrument for both crosslinker experiments? The Eclipse would be the better instrument for crosslink samples in this case.*

We fully agree with the reviewer that the best choice would have been the better instrument and longer gradients. During the course of the experiments, we faced problems with the availability of the equipment, which limited our choice.

3. *The quality of the DSBU samples is not very good and it also does not add new information to your dataset, why did the authors include this data (just out of curiosity)? Selecting crosslinks that have been identified in at least 60% of the replica and with two search engines would be enough to argue your selection.*

From the eight intermolecular crosslinks identified with DSBU, one was exclusively detected with this cross-linker (XL ID 14, Supplementary Table 2), which in the subsequent filtering with DisVis was categorized as one of the eleven self-consistent inter-XLs. On the other hand, the identification of common cross-links between the two cross-linkers gave us further confidence in the XL results.

4. *FDR is very critical in crosslinking experiments, although I know that not all search engines report on residue pair level or PPI level, it is essential to choose a search engine that can calculate on those levels and not only on CSM level. The authors have used the crosslinking data additionally to their biological*

findings and have reasoned their selection of crosslinks, even with the low quality of the data, the usage of crosslinking data as support data is valid for this study. In any case I would recommend using software tools that can control FDR for crosslinking data on higher levels than CSM level in the future. Overall, the quality of this publication on the crosslinking part has improved significantly and crosslinking data details are now reported in a transparent manner.

We appreciate this suggestion of the Reviewer as it led us to improve the robustness of our analysis in the future.

Point-by-point response to the Reviewer #1 comments

We sincerely thank the Reviewer for carefully reading our work and for providing valuable comments. The Reviewer's suggestions have greatly improved our manuscript.

Reviewer #1 (Remarks to the Author):

1. I thank the authors for testing the hypothesis whether $\alpha 2\delta$ is involved in the proposed β interaction with actin, with respect to effects on functional effects. The authors should explicitly state at the beginning of the Results for the functional studies that $\alpha 2\delta$ was not included (line 334).

Done. The following sentence has been added:

"To test the effect of $\text{Ca}_v\beta_2$ 8Ala on currents mediated by L-type $\text{Ca}_v1.2$ channels, we performed whole-cell recordings from cells coexpressing the $\text{Ca}_v1.2$ subunit and $\text{Ca}_v\beta_2$ 8Ala (in the absence of $\text{Ca}_v\alpha 2\delta$)"

2. Line 475-479: In this new paragraph in the Discussion, "Data not shown" is not acceptable as a statement. These new data should be shown as a supplemental figure with the description in the Results section.

"A proposed quality control during $\text{Ca}_v1.2$ channel biogenesis is the interaction of the $\text{Ca}_v1.2/\text{Ca}_v\beta$ core complex with the endoplasmic reticulum membrane protein complex and $\text{Ca}_v\alpha 2\delta 1$ accessory subunit¹⁰⁷. In our hands, coexpression of $\text{Ca}_v\alpha 2\delta$ along with the $\text{Ca}_v1.2/\text{Ca}_v\beta 2$ channel core complex marginally altered the biophysical properties of the channel complex and did not prevent the effect of the actin-association-deficient $\text{Ca}_v\beta 2$ mutant (data not shown).

Done. The data has been added to the new Supplementary Figure 17.

3. The remainder of my concerns (points 2-4) have either been addressed or are matters of opinion and the changes to the ms are appropriate.

We acknowledge the Reviewer for the fair revision of our manuscript.